# INTERNSVG: TOWARDS UNIFIED SVG TASKS WITH MULTIMODAL LARGE LANGUAGE MODELS

**Haomin Wang**[1,2*]   **Jinhui Yin**[3,2*]   **Qi Wei**[3,2*]   **Wenguang Zeng**[4]   **Lixin Gu**[2]
**Shenglong Ye**[2]   **Zhangwei Gao**[1,2]   **Yaohui Wang**[2]   **Yanting Zhang**[4]   **Yuanqi Li**[3]
**Yanwen Guo**[3]   **Wenhai Wang**[5]   **Kai Chen**[2]   **Yu Qiao**[2]   **Hongjie Zhang**[2†]
[1] Shanghai Jiao Tong University   [2] Shanghai AI Laboratory   [3] Nanjing University
[4] Donghua University   [5] The Chinese University of Hong Kong
Project Page: `https://hmwang2002.github.io/release/internsvg`
`kiyotakawang@sjtu.edu.cn, nju.zhanghongjie@gmail.com`

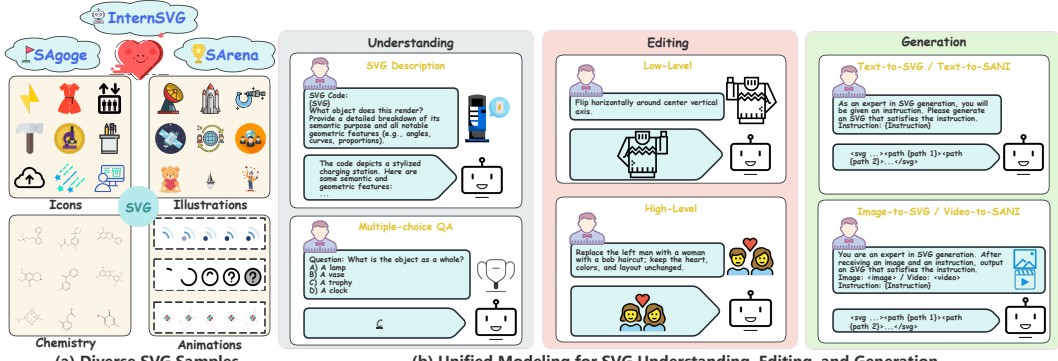

Figure 1: **Overview of our InternSVG family**. **SAgoge** provides large-scale and diverse SVG samples across multiple domains. **SArena** enables comprehensive assessment of existing MLLMs on SVG tasks. **InternSVG** supports unified modeling for SVG understanding, editing, and generation.

## ABSTRACT

General SVG modeling remains challenging due to fragmented datasets, limited transferability of methods across tasks, and the difficulty of handling structural complexity. In response, we leverage the strong transfer and generalization capabilities of multimodal large language models (MLLMs) to achieve unified modeling for SVG understanding, editing, and generation. We present the InternSVG family, an integrated data–benchmark–model suite. At its core is SAgoge, the largest and most comprehensive multimodal dataset for SVG tasks, encompassing both static graphics and dynamic animations. It covers icons, long-sequence illustrations, scientific diagrams, and dynamic animations, supporting tasks of varied difficulty levels and providing deeper hierarchies with richer attributes compared to previous datasets. Based on this resource, we introduce SArena, a companion benchmark with comprehensive task definitions and standardized evaluation that aligns with the domains and difficulty spectrum covered by SAgoge. Building on these foundations, we propose InternSVG, a unified MLLM for SVG understanding, editing, and generation with SVG-specific special tokens, subword-based embedding initialization, and a two-stage training strategy that progresses from short static SVGs to long-sequence illustrations and complex animations. This unified formulation induces positive transfer and improves overall performance. Experiments on SArena and prior benchmark confirm that InternSVG achieves substantial gains and consistently outperforms leading open and proprietary counterparts.

---

[*]Equal Contribution.

[†]Corresponding Author.

# 1 INTRODUCTION

Scalable Vector Graphics (SVG) is an XML-based standard for 2D graphics that offers compact storage, fine-grained editability, and resolution-independent rendering across displays. It can function either as a stand-alone format or as an embedded component within other documents, while also supporting interactivity and animation. Additionally, it interoperates smoothly with CSS, the DOM, and JavaScript in web contexts. These properties have led to broad adoption in web design, scientific visualization, and computer-aided design. Nevertheless, enabling machines to understand, generate, and edit SVGs remains challenging due to the methodological complexity involved, the scarcity of large-scale and high-quality training corpora, and the requirement for models that generalize across tasks rather than specializing in isolation.

Existing resources and methods face three major limitations. First, current datasets provide limited support for unified modeling of SVG tasks. Most datasets and benchmarks target a single task, such as semantic understanding in SGP-Bench (Qiu et al., 2024), instruction-based editing in SVGEditBench (Nishina & Matsui, 2024), or Text-to-SVG and Image-to-SVG generation in ColorSVG-100K (Chen & Pan, 2025), SVG-Stack (Rodriguez et al., 2025), and MMSVG (Yang et al., 2025b). This narrow scope leads to fragmented supervision and evaluation, which in turn limits transfer across understanding, editing, and generation. Second, existing datasets are limited in both scale and diversity. UniSVG (Li et al., 2025) focuses on unified SVG understanding and generation, but it contains only about 525k samples. SVGenius (Chen et al., 2025) provides a comprehensive benchmark spanning understanding, editing, and generation, while it contains only about 2,400 queries, making it suitable for evaluation rather than large-scale training. More broadly, current datasets concentrate on common static images such as icons and illustrations, while paying insufficient attention to SVGs with specific applications in professional domains. Third, existing methods commonly lack transferability and generalization. Optimization-based and differentiable rasterization pipelines scale poorly and lack semantic reasoning (Li et al., 2020; Ma et al., 2022). Although some approaches (Jain et al., 2023; Xing et al., 2024) employ diffusion models to improve visual fidelity, the resulting SVGs often suffer from limited editability and weak detail-aware control. Recently, LLM-based methods (Xing et al., 2025; Rodriguez et al., 2025; Yang et al., 2025b) have achieved significant progress in Text-to-SVG and Image-to-SVG generation, but they struggle to generalize to long sequences and complex SVG content, making it difficult to ensure generation quality. Moreover, these methods largely overlook tasks related to SVG understanding and editing.

To overcome these limitations, we propose the **InternSVG family**, an integrated data–benchmark–model suite for unified SVG modeling. Specifically, we introduce **SAgoge**, a unified multimodal dataset that jointly supports SVG understanding, editing, and generation, covering both static graphics and dynamic animations. SAgoge combines Internet-sourced and synthetic SVG data across icons, illustrations, chemical structural formulas, and animations, totaling about 16 million training samples.

Table 1: Comparison of SVG datasets and benchmarks.

| Name | Understanding | Editing | Generation | Multi-Dom | #Samples |
|---|---|---|---|---|---|
| **Datasets** | | | | | |
| ColorSVG-100K | ✗ | ✗ | ✓ | ✗ | 100k |
| SVG-Stack | ✗ | ✗ | ✓ | ✗ | 2.2M |
| MMSVG | ✗ | ✗ | ✓ | ✓ | 2.0M |
| SVGX | ✓ | ✗ | ✓ | ✗ | 1.0M |
| UniSVG | ✓ | ✗ | ✓ | ✗ | 525k |
| DeepSVG | ✗ | ✗ | ✓ | ✓ | 100k |
| SAgoge (Ours) | ✓ | ✓ | ✓ | ✓ | 16M |
| **Benchmarks** | | | | | |
| SGP-Bench | ✓ | ✗ | ✗ | ✗ | 4.3k |
| SVGEditBench | ✗ | ✓ | ✗ | ✗ | 1.3k |
| VGBench | ✓ | ✗ | ✓ | ✗ | 10.1k |
| SVGenius | ✓ | ✓ | ✓ | ✗ | 2.3k |
| SArena (Ours) | ✓ | ✓ | ✓ | ✓ | 31k |

Comparison of SAgoge and other datasets is shown in Table 1. To the best of our knowledge, SAgoge is the largest multimodal SVG dataset to date, with the broadest task coverage and the most comprehensive range of difficulty. It includes two understanding tasks, ten editing tasks, and four generation tasks, covering semantic understanding, icon editing and generation, long-sequence illustration generation, animation generation, and the generation of scientific diagrams. To enable rigorous and comparable assessment, we propose **SArena**, a companion benchmark that standardizes tasks and metrics for understanding, editing, and generation, with sub-benchmarks covering icons, illustrations, chemical structural formulas, and animations. Together, SAgoge and SArena provide the scale, diversity, full task coverage, and standardized measurement needed to move beyond fragmented evaluations and enable systematic investigation of general SVG modeling in a unified setting.

Building on SAgoge and the accompanying SArena for systematic evaluation, we introduce **InternSVG**, a unified MLLM for SVG understanding, editing, and generation. InternSVG augments a pretrained vision–language backbone with SVG-specific tokenization, introducing compact special

tokens for tags, attributes, and coordinates to reduce sequence length while retaining geometric and hierarchical structure. These tokens are initialized with a subword-based strategy that anchors them in the pretrained embedding space, stabilizing early training and accelerating convergence. Training adopts a two-stage strategy that progresses from short static SVGs to longer illustrations and complex animations. Through extensive experiments, we demonstrate that unified modeling can effectively improve performance across understanding, editing, and generation tasks. Comprehensive evaluations further show that our InternSVG surpasses both open-source and proprietary models on SArena and previous benchmarks. For example, on the SArena-Icon benchmark, InternSVG surpasses Claude-Sonnet-4, the strongest proprietary baseline on SVG tasks, by about $11\%$ higher acc in understanding tasks, $34\%$ higher PSNR in editing tasks, $56\%$ lower FID in Text-to-SVG tasks, and $22\%$ higher SSIM in Image-to-SVG tasks.

In summary, our contributions are below:

(1) We construct SAgoge, the largest and most comprehensive multimodal SVG dataset to date, encompassing static graphics and animations with over 16 million training samples. To enable rigorous and comparable evaluation, we further establish SArena, a companion benchmark that standardizes tasks and metrics across SVG understanding, editing, and generation.

(2) We propose InternSVG, a unified MLLM for SVG understanding, editing, and generation. It introduces SVG-specific tokenization with subword-initialized special tokens and adopts a two-stage training strategy to support effective cross-task generalization.

(3) We conduct extensive experiments to demonstrate the benefits of unified modeling. The results on SArena and prior benchmarks show that our InternSVG outperforms traditional approaches as well as general-purpose open-source and proprietary models.

## 2 RELATED WORKS

### 2.1 SVG DATASETS AND BENCHMARKS

Most existing SVG datasets and benchmarks are limited in task coverage or data type and remain too small for effective model training, leading to fragmented evaluations and limited insights into generalization across tasks and complexity. SGP-Bench (Qiu et al., 2024) evaluates semantic comprehension and consistency in symbolic graphics programs. SVGEditBench (Nishina & Matsui, 2024) and its extension V2 (Nishina & Matsui, 2025) focus narrowly on instruction-based SVG editing measured by low-level syntactic metrics. On the generative side, SVG-Stack (Rodriguez et al., 2025), SVGX (Xing et al., 2025), MMSVG (Yang et al., 2025b), and ColorSVG-100K (Chen & Pan, 2025) address Text-to-SVG and Image-to-SVG generation, while VGBench Zou et al. (2024) and UniSVG (Li et al., 2025) jointly evaluate understanding and generation. DeepSVG (Carlier et al., 2020) introduces a dataset of 100K SVG icons and explores generation, interpolation, and latent-space animation of static and limited animated graphics, but lacks rich editing instructions and image-conditioned generation. SVGenius (Chen et al., 2025) introduces a comprehensive benchmark covering understanding, editing, and generation with systematic complexity levels and multi-dimensional metrics, but it includes only about 2,400 queries, making it sufficient for evaluation yet inadequate for training. In contrast, our SAgoge is substantially larger and more diverse, encompassing both static graphics and SVG animations. It unifies SVG understanding, editing, and generation, and with approximately 16M task samples, its scale and diversity enable robust model training and comprehensive evaluation across the full spectrum of SVG tasks, which effectively address the coverage and scalability limitations of prior datasets.

### 2.2 SVG MODELING METHODS

Early research on SVG modeling treated vector graphics as sequences of geometric primitives and relied on specialized generative architectures trained on limited domains (Ha & Eck, 2017; Lopes et al., 2019; Frans et al., 2022; Vinker et al., 2022; Hu et al., 2024). Approaches such as DeepSVG (Carlier et al., 2020) used hierarchical VAEs with Transformer decoders to generate icons, while optimization-based methods like DiffVG (Li et al., 2020) and LIVE (Ma et al., 2022) applied differentiable rasterization to align SVG primitives with target images iteratively. Although these techniques captured low-level structure, they lacked semantic understanding, struggled with

complex compositions, and were computationally costly, often producing redundant or unsimplified paths. More recent works have explored diffusion-based pipelines, such as VectorFusion (Jain et al., 2023) and SVGDreamer (Xing et al., 2024), which refine vector renderings via Text-to-Image diffusion combined with SVG constraints. These approaches improve visual fidelity but remain limited to generation and offer only restricted controllability and task diversity. The emergence of large language models (LLMs) (Achiam et al., 2023; Yang et al., 2025a; OpenAI, 2025) has shifted SVG modeling toward code-centric paradigms that leverage the textual nature of SVG. Methods such as StarVector (Rodriguez et al., 2025), OmniSVG (Yang et al., 2025b), LLM4SVG (Xing et al., 2025), and SVGBuilder (Chen & Pan, 2025) integrate visual encoders with text decoders to generate SVGs from textual or visual inputs, achieving notable advances in Text-to-SVG and Image-to-SVG synthesis. However, these approaches are fragmented across isolated tasks, with limited mechanisms to handle the increasing complexity of SVG content. Different from prior approaches, InternSVG is a unified MLLM for SVG understanding, editing, and generation. It is capable of understanding both the semantic and structural information of SVG code, editing graphics based on user instructions, and generating SVGs from textual or visual prompts.

## 3 DATASET AND BENCHMARK

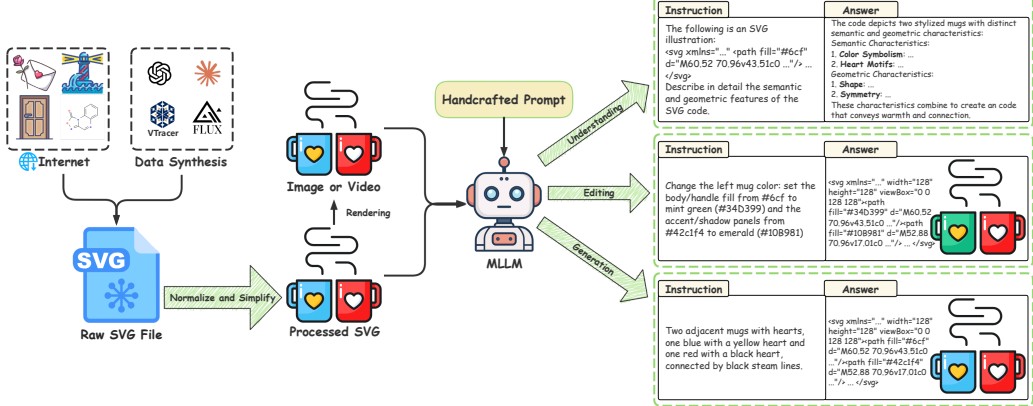

Figure 2: **Overview of the dataset construction pipeline.** Raw SVGs are gathered from the web and a custom synthesis pipeline, then normalized to a $128 \times 128$ canvas and simplified to shorten code. The rendered images or videos, processed SVG code, and handcrafted prompts are fed to an MLLM to synthesize high-quality training samples for understanding, editing, and generation.

### 3.1 SAGOGE DATASET

We introduce SAgoge, a large-scale and comprehensive dataset for SVG tasks with more than 16 million training samples spanning icons, illustrations, chemical structures, and animations. The construction pipeline is illustrated in Figure 2.

#### 3.1.1 TASK DEFINITION

SAgoge is a comprehensive resource that supports unified modeling of SVG understanding, editing, and generation. We define a fine-grained task suite spanning diverse vector-graphic categories to enable consistent training and evaluation. Understanding comprises SVG description and multiple-choice question answering. Editing for icons contains ten subtasks organized by difficulty, including eight low-level operations (color editing, stroke addition, translation, scaling, rotation, flipping, transparency adjustment, and cropping) and two high-level tasks (semantic color editing and style transfer). Generation encompasses Text-to-SVG and Image-to-SVG for icons, illustrations, and chemical structures, as well as Text-to-SANI and Video-to-SANI for animations. Appendix C.3.3 presents examples of tasks in SAgoge.

### 3.1.2 DATASET CREATION PIPELINE

**Data Source.** SAgoge integrates Internet-sourced and synthetic SVGs. Icons and a subset of illustrations and animations are collected from public repositories such as Iconfont, SVGRepo, and OpenClipart. Chemical structures are generated by converting PubChem SDF files to SVG with the Open Babel toolkit. Due to the scarcity of open-source SVGs for illustrations and animations, we build a dedicated data synthesis pipeline to expand coverage in these domains. Implementation details are provided in Appendix C.3 and data statistics of SAgoge are summarized in Table 24.

**Normalize and Simplify.** To improve training efficiency and reduce complexity, all SVGs are normalized to a $128 \times 128$ viewBox. Because collected files are often verbose and LLMs operate under limited context lengths, we further simplify the code by removing nonessential metadata, comments, and redundant declarations while preserving the core geometric primitives, semantic information, and hierarchical group structures. This normalization and simplification shorten token sequences, improve consistency, and make the corpus better suited for scalable training and evaluation.

**Construction of Understanding and Generation Data.** We employ distinct annotation strategies tailored to different data modalities to ensure high-quality. We first render SVGs into raster images or videos, then utilize specific MLLMs with carefully designed prompts. For understanding tasks, we feed rasterized images to MLLMs such as InternVL3 (Zhu et al., 2025) (train set) or GPT-4o (Hurst et al., 2024) (test set) to obtain fine-grained geometric and semantic descriptions. For generation tasks, icons and illustrations are annotated by providing rasterized images to an MLLM for captioning. For chemical data, we directly use the IUPAC names or common chemical names retrieved from PubChem as textual instructions. For animation data, each SVG animation is converted into an MP4 video and annotated by Gemini 2.0 Flash (Google DeepMind, 2024).

**Construction of Editing Data.** The editing dataset consists of triplets (instruction, original SVG, edited SVG) generated across three categories: simple editing, color editing, and style transfer. Simple editing operations are applied via template-based SVG code modifications and verified by Qwen2.5-VL-72B (Bai et al., 2025). Color editing uses code-level replacement from a 147-color palette with MLLM-generated instructions. Style transfer pairs category-consistent SVGs using CLIP (similarity > 0.8) and synthesizes instructions from paired images and captions. Qwen2.5-VL-72B annotates the train set, while GPT-4o is used for the test set with human verification. Prompt templates for dataset construction are provided in Appendix C.4.1.

## 3.2 SARENA BENCHMARK

To enable systematic evaluation across SVG understanding, editing, and generation, we introduce SArena, a benchmark that aligns with the domains and difficulty spectrum covered by SAgoge and provides standardized tasks and metrics. SArena includes 4 sub-benchmarks, *i.e.*, icons, illustrations, chemical structures, and animation. To ensure reliability and fairness of evaluation, all SVG samples used in SArena are carefully curated through a combination of automated preprocessing and manual screening, with low-quality, corrupted, and semantically ambiguous files removed. Data statistics of SArena are shown in Appendix C.2.

### 3.2.1 EVALUATION METRICS

**Understanding.** We evaluate code-level comprehension with four-option multiple-choice QA. The model receives only the SVG code and must infer the answer from its semantics and structure, and performance is reported as accuracy.

**Editing.** The model edits the original SVG code to follow textual instructions. Performance is evaluated on rendered outputs using DINO score (Oquab et al., 2023), Structural Similarity Index (SSIM) (Wang et al., 2004), Learned Perceptual Image Patch Similarity (LPIPS) (Zhang et al., 2018), and Peak Signal-to-Noise Ratio (PSNR).

**Text-to-SVG.** The model synthesizes SVGs from textual instructions. Quality is evaluated with FID (Theis et al., 2015) and FID-CLIP (Wu et al., 2023) for distributional and semantic fidelity, CLIP-T2I (Radford et al., 2021; Hessel et al., 2021) for text–SVG alignment and CLIP-I2I for visual similarity.

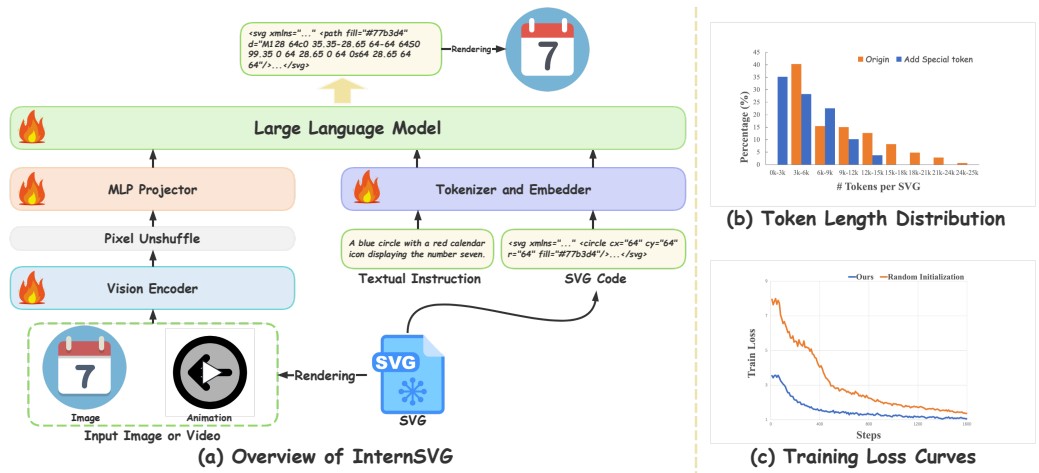

Figure 3: (a) Overall architecuture of InternSVG. (b) Distribution of the number of tokens per SVG before and after adding customized special tokens in the tokenizer. (c) Comparison of training loss curves between subword-based embedding initialization and random initialization.

**Image-to-SVG.** The model receives a reference image and textual instructions and generates the corresponding SVG. Evaluation mirrors the editing task and uses DINO score, SSIM, LPIPS, and PSNR to measure visual similarity between the rendered SVG and the ground-truth image.

**Text-to-SANI.** The model generates SVG animations from textual instructions. Evaluation uses FVD (Unterthiner et al., 2018) for distributional consistency and ViCLIP-based CLIPScores (Wang et al., 2023), with CLIP-T2V quantifying text–animation alignment and CLIP-V2V measuring visual similarity to reference videos.

**Video-to-SANI.** In this task the model generates SVG animations conditioned on an input video and textual instructions, enabling cross-modal animation generation. Evaluation follows the Image-to-SVG protocol and uses DINO, SSIM, LPIPS, and PSNR computed between rendered SVG frames and reference frames on sampled timestamps, with scores averaged to produce the final result.

In addition to the task-specific metrics, we also account for cases where the model generates syntactically incorrect or incomplete SVGs that cannot be rendered. Any unrenderable SVG is replaced with a pure black image or video as a penalty during metric computation, ensuring that the evaluation fairly reflects both the quality of valid outputs and the model's robustness in generation.

# 4 METHOD

## 4.1 MODEL ARCHITECTURE

As illustrated in Figure 3, InternSVG follows the "ViT–MLP–LLM" paradigm (Liu et al., 2023), using InternViT-300M (Chen et al., 2024) as the vision encoder and Qwen2.5-7B (Qwen Team, 2025) as the language model. We further design SVG-specific special tokens and introduce a tailored embedding initialization strategy to incorporate SVG content effectively.

**Special tokens.** We design 55 tag tokens and 42 attribute tokens based on SVG grammar. Tag tokens cover core structural elements such as `svg`, `path`, and `circle`, as well as animation elements like `animate`, `animateMotion`, and `animateTransform`. Attribute tokens include common geometric attributes (e.g., `viewBox`, `cx`, `cy`, `d`) and animation-related attributes (e.g., `dur`, `from`, `to`, `repeatCount`). To represent numerical values, we add integer tokens from `-128` to `128` and 100 decimal tokens from `.0` to `.99`. This fine-grained number representation allows accurate modeling of geometry while keeping the tokenization compact. As shown in Figure 3(b), these special tokens substantially reduce sequence length compared with the original tokenizer, improving representation efficiency and alleviating computational burden.

**Embedding initialization.** Instead of randomly initializing the SVG-specific special tokens, we adopt a subword-based strategy to ensure semantic coherence and stable training. Each token is decomposed into subwords using the pretrained tokenizer, and the embeddings of these subwords are averaged to form the token embedding. Formally, let a new special token $t_{new}$ be decomposed into $n$ subwords $\{s_1, s_2, \ldots, s_n\}$, with corresponding embeddings $\{\mathbf{e}_{s_1}, \mathbf{e}_{s_2}, \ldots, \mathbf{e}_{s_n}\}$. The initialized embedding $\mathbf{e}_{t_{new}}$ is computed as:

$$\mathbf{e}_{t_{new}} = \frac{1}{n} \sum_{i=1}^{n} \mathbf{e}_{s_i} \tag{1}$$

This strategy preserves the semantic prior from the original vocabulary and accelerates adaptation to SVG tokens. As shown in Figure 3(c), it reduces initial loss and accelerates convergence, demonstrating its effectiveness in stabilizing early training and enhancing overall efficiency.

## 4.2 TWO-STAGE TRAINING STRATEGY

The design of our two-stage training strategy is motivated by the inherent imbalance in SVG corpora. Icons are easy to collect in large quantities and have simpler structures, while illustrations are more difficult to obtain, less abundant, and usually longer and more complex. This imbalance in scale and complexity makes unified training challenging. To address this issue, we adopt a curriculum-style approach that progresses from simple to complex samples.

Initially, training is conducted only on the Icon and Chemistry datasets, which contain shorter and simpler SVGs. This stage covers SVG description, all editing tasks, and two generative tasks, namely Text-to-SVG and Image-to-SVG, and allows the model to build basic representation and generation abilities. Once the model has reached preliminary convergence, training is expanded to all datasets and tasks, including longer and more diverse SVGs. To ensure balanced learning, the Icon and Chemistry data are resampled to match the proportions of the other datasets. This progressive strategy enables the model to acquire SVG knowledge gradually and stably while reducing imbalance caused by heterogeneous dataset scales and complexities.

Table 2: Comparison of SVG understanding, editing, and generation on SArena-Icon. We use uppercase bold initials to denote submetrics with **O**verall, **C**olor, **G**eometry, **Q**uantity, and **S**emantics.

| Model | Understanding | | | | | Editing | | | | Text-to-SVG | | | | | Image-to-SVG | | | | |
|---|---|---|---|---|---|---|---|---|---|---|---|---|---|---|---|---|---|---|---|
| | O | C | G | Q | S | DINO↑ | SSIM↑ | LPIPS↓ | PSNR↑ | FID↓ | FID-C↓ | CLIP-T2I↑ | CLIP-I2I↑ | Tokens | DINO↑ | SSIM↑ | LPIPS↓ | PSNR↑ | Tokens |
| Traditional SVG methods | | | | | | | | | | | | | | | | | | | |
| IconShop | – | – | – | – | – | – | – | – | – | 32.288 | 17.919 | 20.894 | 70.922 | 1.5k | – | – | – | – | – |
| VectorFusion | – | – | – | – | – | – | – | – | – | 16.594 | 17.394 | 22.992 | 70.308 | 33k | – | – | – | – | – |
| SVGDreamer | – | – | – | – | – | – | – | – | – | 26.612 | 33.312 | 20.329 | 69.975 | 132k | – | – | – | – | – |
| DiffVG | – | – | – | – | – | – | – | – | – | – | – | – | – | – | 0.869 | 0.927 | 0.097 | 23.614 | 17k |
| LIVE | – | – | – | – | – | – | – | – | – | – | – | – | – | – | 0.973 | 0.986 | 0.024 | 35.419 | 18k |
| VTracer | – | – | – | – | – | – | – | – | – | – | – | – | – | – | 0.966 | 0.875 | 0.054 | 21.748 | 4.4k |
| Large language models | | | | | | | | | | | | | | | | | | | |
| Qwen2.5-VL-7B | 52.8 | 69.3 | 50.4 | 34.9 | 56.4 | 0.909 | 0.728 | 0.192 | 25.402 | 24.781 | 15.454 | 21.538 | 71.384 | 249 | 0.781 | 0.506 | 0.378 | 6.534 | 281 |
| InternVL3-8B | 59.5 | 79.1 | 59.3 | 38.2 | 61.3 | 0.921 | 0.761 | 0.170 | 29.615 | 23.061 | 14.303 | 21.897 | 71.450 | 269 | 0.812 | 0.557 | 0.361 | 7.220 | 256 |
| Llama-4-Maverick | 64.7 | 87.5 | 62.0 | 47.2 | 62.3 | 0.966 | 0.870 | 0.109 | 46.944 | 14.931 | 6.526 | 23.570 | 75.816 | 265 | 0.863 | 0.596 | 0.329 | 8.027 | 255 |
| Qwen2.5-VL-72B | 63.4 | 82.4 | 65.1 | 44.6 | 61.6 | 0.961 | 0.849 | 0.124 | 41.006 | 15.948 | 9.875 | 22.946 | 73.681 | 275 | 0.837 | 0.584 | 0.346 | 7.834 | 372 |
| InternVL3-78B | 65.3 | 86.4 | 71.0 | 48.8 | 54.9 | 0.958 | 0.848 | 0.116 | 40.533 | 17.580 | 10.596 | 22.805 | 73.123 | 252 | 0.850 | 0.584 | 0.339 | 7.802 | 234 |
| GPT-4o | 71.0 | 88.2 | 78.5 | 47.5 | 69.6 | 0.968 | 0.887 | 0.088 | 55.255 | 15.178 | 6.763 | 24.617 | 77.742 | 246 | 0.874 | 0.616 | 0.316 | 8.435 | 231 |
| Gemini-2.5-Flash | 73.0 | 90.1 | 81.9 | 53.0 | 67.2 | 0.942 | 0.815 | 0.113 | 54.200 | 16.720 | 5.208 | 24.658 | 78.218 | 451 | 0.876 | 0.587 | 0.316 | 8.324 | 533 |
| Claude-Sonnet-4 | 77.1 | 91.5 | 82.4 | 53.8 | 80.6 | 0.979 | 0.915 | 0.071 | 57.595 | 15.840 | 4.291 | **25.421** | 80.579 | 444 | 0.915 | 0.665 | 0.276 | 9.855 | 541 |
| StarVector 8B | – | – | – | – | – | – | – | – | – | – | – | – | – | – | 0.871 | 0.623 | 0.206 | 13.595 | 951 |
| LLM4SVG 7B | – | – | – | – | – | – | – | – | – | 21.939 | 8.611 | 19.458 | 70.726 | 705 | 0.748 | 0.472 | 0.409 | 5.375 | 485 |
| OmniSVG 3B | – | – | – | – | – | – | – | – | – | 28.292 | 11.318 | 21.679 | 74.831 | 1.7k | 0.894 | 0.756 | 0.186 | 12.669 | 2.4k |
| InternSVG 8B | **85.1** | **93.0** | **85.8** | **61.9** | **99.7** | **0.989** | **0.952** | **0.036** | **77.331** | **8.715** | **1.876** | 23.916 | **80.911** | 1.0k | **0.949** | **0.811** | **0.127** | **18.226** | 1.3k |
| | +8.0 | +1.5 | +2.8 | +8.1 | +19.1 | +0.010 | +0.037 | +0.035 | +19.736 | +6.216 | +2.415 | -1.505 | +0.332 | | +0.034 | +0.055 | +0.059 | +4.631 | |

# 5 EXPERIMENTS

## 5.1 EXPERIMENTAL SETUP

InternSVG builds on a pretrained InternVL3-8B model and adopts a two-stage training strategy. In stage one, we train on the Icon and Chemistry datasets that contain shorter and simpler SVGs. Icon provides 1.0M samples for understanding, 1.0M for editing, and 5.6M for generation, and Chemistry provides 2.0M generation samples. In stage two, we expand to all domains and tasks, using 350K understanding, 500K editing, and 1.1M generation samples from Icon, 1.0M generation samples from Illustration, 1.0M from Chemistry, and 120K from Animation. Optimization uses AdamW

Table 3: Comparison of SVG generation results on SArena-Illustration.

| Model | Text-to-SVG | | | | | Image-to-SVG | | | | |
|---|---|---|---|---|---|---|---|---|---|---|
| | FID ↓ | FID-C ↓ | CLIP-T2I ↑ | CLIP-I2I ↑ | Tokens | DINO ↑ | SSIM ↑ | LPIPS ↓ | PSNR ↑ | Tokens |
| **Traditional SVG methods** | | | | | | | | | | |
| VectorFusion | 28.198 | 15.661 | 22.638 | 71.220 | 33k | – | – | – | – | – |
| SVGDreamer | 32.410 | 23.427 | 21.026 | 69.775 | 132k | – | – | – | – | – |
| DiffVG | – | – | – | – | – | 0.870 | 0.886 | 0.138 | 21.605 | 17k |
| LIVE | – | – | – | – | – | 0.963 | 0.948 | 0.078 | 29.055 | 18k |
| VTracer | – | – | – | – | – | 0.965 | 0.879 | 0.093 | 21.754 | 10k |
| **Large language models** | | | | | | | | | | |
| Qwen2.5-VL-7B | 37.903 | 28.455 | 18.069 | 61.928 | 756 | 0.739 | 0.513 | 0.413 | 7.732 | 1.2k |
| InternVL3-8B | 36.736 | 25.682 | 18.493 | 61.964 | 493 | 0.772 | 0.569 | 0.397 | 8.542 | 716 |
| Llama-4-Maverick | 30.835 | 14.831 | 21.872 | 67.366 | 551 | 0.839 | 0.644 | 0.340 | 10.469 | 608 |
| Qwen2.5-VL-72B | 29.521 | 18.407 | 20.923 | 65.349 | 527 | 0.808 | 0.628 | 0.363 | 9.900 | 886 |
| InternVL3-78B | 30.457 | 19.195 | 20.577 | 64.826 | 454 | 0.830 | 0.638 | 0.348 | 9.985 | 514 |
| GPT-4o | 28.124 | 14.150 | 23.637 | 70.696 | 473 | 0.850 | 0.663 | 0.327 | 10.723 | 484 |
| Gemini-2.5-Flash | 28.865 | 8.894 | **24.800** | **74.796** | 1.2k | 0.829 | 0.516 | 0.359 | 9.091 | 1.8k |
| Claude-Sonnet-4 | 27.294 | 7.640 | 23.094 | 74.525 | 1.0k | 0.901 | 0.670 | 0.305 | 11.731 | 1.3k |
| LLM4SVG 7B | 48.704 | 29.568 | 15.468 | 62.933 | 1.2k | 0.713 | 0.494 | 0.413 | 6.221 | 476 |
| OmniSVG 3B | 42.756 | 22.885 | 16.861 | 64.815 | 4.5k | 0.797 | 0.656 | 0.330 | 10.433 | 6.7k |
| InternSVG 8B | **22.397** | **5.141** | 21.116 | 74.662 | 8.1k | **0.924** | **0.716** | **0.188** | **14.644** | 7.7k |
| | +4.897 | +2.499 | -3.684 | -0.134 | | +0.023 | +0.046 | +0.117 | +2.913 | |

Table 4: Comparison of SVG generation results on SArena-Chemistry and SArena-Animation.

| Model | Chemistry: Text-to-SVG | | | Chemistry: Image-to-SVG | | | | Animation: Text-to-SANI | | | Animation: Video-to-SANI | | | |
|---|---|---|---|---|---|---|---|---|---|---|---|---|---|---|
| | FID ↓ | FID-C ↓ | CLIP-I2I ↑ | DINO ↑ | SSIM ↑ | LPIPS ↓ | PSNR ↑ | FVD ↓ | CLIP-T2V ↑ | CLIP-V2V ↑ | DINO ↑ | SSIM ↑ | LPIPS ↓ | PSNR ↑ |
| Qwen2.5-VL-7B | 56.248 | 73.698 | 51.814 | 0.769 | 0.468 | 0.274 | 7.501 | 214.379 | 19.118 | 50.649 | 0.787 | 0.716 | 0.273 | 11.758 |
| InternVL3-8B | 33.613 | 61.675 | 56.856 | 0.865 | 0.783 | 0.203 | 13.840 | 310.066 | 17.017 | 43.856 | 0.780 | 0.612 | 0.286 | 9.883 |
| Llama-4-Maverick | 26.844 | 31.924 | 69.643 | 0.908 | 0.798 | 0.173 | 14.977 | 141.470 | 22.304 | 67.615 | 0.841 | 0.754 | 0.246 | 12.858 |
| Qwen2.5-VL-72B | 32.307 | 44.540 | 63.931 | 0.846 | 0.647 | 0.215 | 12.106 | 151.682 | 20.376 | 59.454 | 0.834 | 0.721 | 0.261 | 11.931 |
| InternVL3-78B | 29.216 | 40.080 | 65.969 | 0.911 | 0.813 | 0.177 | 15.375 | 169.159 | 20.263 | 60.896 | 0.828 | 0.704 | 0.264 | 11.336 |
| GPT-4o | 24.505 | 19.297 | 76.599 | 0.920 | 0.791 | 0.174 | 14.673 | 155.393 | 22.808 | 70.608 | 0.860 | 0.743 | 0.250 | 12.260 |
| Gemini-2.5-Flash | 27.708 | 21.777 | 75.897 | 0.934 | 0.817 | 0.155 | 15.539 | 151.983 | 22.239 | 66.554 | 0.847 | 0.701 | 0.257 | 12.015 |
| Claude-Sonnet-4 | 21.252 | 15.240 | 78.308 | 0.957 | 0.871 | **0.132** | 17.554 | 169.484 | **24.070** | **74.179** | 0.867 | **0.760** | 0.240 | 13.189 |
| StarVector 8B | – | – | – | 0.977 | 0.841 | 0.147 | 17.419 | – | – | – | – | – | – | – |
| InternSVG 8B | **9.974** | **0.877** | **93.931** | **0.994** | **0.873** | 0.138 | **17.722** | **99.474** | 22.572 | 73.162 | **0.876** | 0.754 | **0.237** | **14.168** |
| | +11.278 | +14.363 | +15.623 | +0.017 | +0.002 | -0.006 | +0.178 | +41.996 | -1.498 | -1.017 | +0.009 | -0.006 | +0.003 | +0.979 |

with a learning rate of $2 \times 10^{-4}$ in stage one and $1 \times 10^{-4}$ in stage two, running on 96 NVIDIA A800 GPUs with a per-device batch size of 1.

## 5.2 QUANTITATIVE EVALUATIONS

To validate the effectiveness of our SAgoge dataset and InternSVG, we evaluate multiple models on the SArena benchmark. The evaluation covers open-source MLLMs, proprietary systems, traditional SVG generation methods, and LLM-based approaches, including LLM4SVG (Xing et al., 2025), StarVector (Rodriguez et al., 2025), and OmniSVG (Yang et al., 2025b). The complete list of evaluated models with their parameter scales is given in Appendix B.1.

As shown in Table 2, we systematically evaluate various models on SArena-Icon across understanding, editing, and generation tasks. Our InternSVG outperforms the second-best model by 8 points in Overall accuracy. In editing, InternSVG achieves the best scores across DINO, SSIM, LPIPS, and PSNR, yielding higher visual fidelity and geometric consistency. For Text-to-SVG, it achieves the best FID, FID-C, and CLIP-I2I among all methods, while its CLIP-T2I is comparable to that of Claude-Sonnet-4. In Image-to-SVG, optimization-based methods remain strong on similarity, yet InternSVG achieves comparable visual similarity while producing much more compact code at about 1.3k tokens per SVG, roughly one fourteenth of LIVE's token count, and it unifies understanding, editing, and generation within a single model.

As shown in Table 3, on SArena-Illustration, our method achieves the best FID and FID-C in Text-to-SVG, with CLIPScores comparable to Gemini-2.5-Flash, the best-performing proprietary system, and outperforms both general MLLMs and LLM-based methods in Image-to-SVG. On SArena-Chemistry, where common or IUPAC names are used as prompts, existing MLLMs perform poorly on Text-to-SVG due to limited training on scientific graphics, while InternSVG achieves the best re-

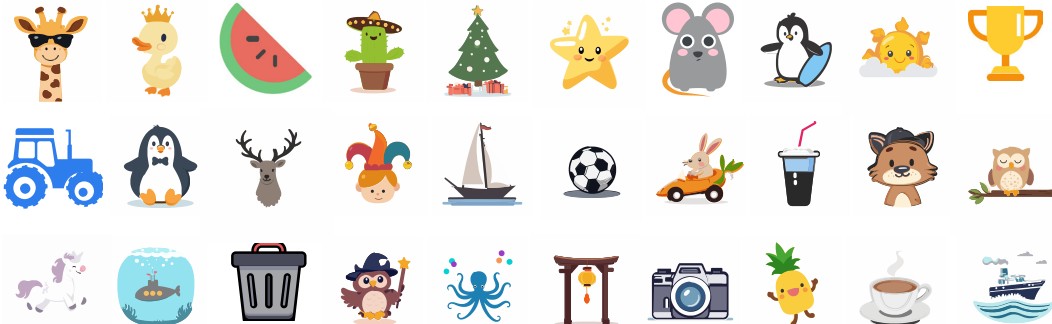

Figure 4: **Visualization of SVG samples generated by InternSVG.**

sults on both tasks with a large margin in Text-to-SVG (Table 4). On SArena-Animation, InternSVG follows a protocol of sampling 8 frames from video input and surpasses all open-source MLLMs, reaching performance close to proprietary models on both Text-to-SANI and Video-to-SANI. Complete evaluation results are provided in Appendix B.2.

To fully validate the effectiveness of SAgoge and InternSVG, we further evaluated the performance of InternSVG on established benchmarks, including SGP-Bench (Qiu et al., 2024), SVG-Stack (Rodriguez et al., 2025), and UniSVG (Li et al., 2025). Notably, InternSVG achieves the best score on SGP-Bench, surpassing proprietary models like Claude-Sonnet-4, and dominates UniSVG with a Final Score of 0.826. On SVG-Stack, InternSVG achieves superior Dino, LPIPS, and MSE scores with comparable SSIM, using only 1.2k tokens compared to StarVector's 3.7k–5.3k. The results demonstrate that the SAgoge dataset and InternSVG framework provide significant improvements across diverse SVG tasks, proving their efficacy extends well beyond the specific data distribution of SArena. Additional results are reported in Appendix B.3, B.4, and B.5.

### 5.3 QUALITATIVE VISUALIZATION

Figure 4 presents qualitative examples of SVGs generated by InternSVG. The results show that our InternSVG can produce diverse and visually appealing graphics with clear structures and semantically rich content. More qualitative visualization results of InternSVG and comparisons with other models can be found in Appendix B.9.

### 5.4 ABLATION STUDIES

Table 5: Ablation results on Generation (G), Understanding (U), and Editing (E) under different task combinations.

| Tasks | U Overall | Text-to-SVG | | | | Image-to-SVG | | | | Editing | | | |
|---|---|---|---|---|---|---|---|---|---|---|---|---|---|
| | | FID ↓ | FID-C ↓ | CLIP-T2I ↑ | CLIP-I2I ↑ | DINO ↑ | SSIM ↑ | LPIPS ↓ | PSNR ↑ | DINO ↑ | SSIM ↑ | LPIPS ↓ | PSNR ↑ |
| G | – | 15.548 | 7.848 | 21.073 | 73.016 | 0.807 | 0.547 | 0.387 | 6.860 | – | – | – | – |
| U | 62.9 | – | – | – | – | – | – | – | – | – | – | – | – |
| E | – | – | – | – | – | – | – | – | – | 0.943 | 0.811 | 0.125 | 42.131 |
| G+U | 75.1 | 12.881 | 6.750 | 21.070 | 73.310 | 0.813 | 0.563 | 0.382 | 7.042 | – | – | – | – |
| G+E | – | 13.842 | 6.810 | 21.099 | 73.279 | 0.808 | 0.553 | 0.383 | 6.897 | 0.969 | 0.881 | 0.090 | 49.503 |
| G+U+E | 75.4 | 12.387 | 6.611 | 21.198 | 73.630 | 0.814 | 0.568 | 0.379 | 7.095 | 0.977 | 0.906 | 0.075 | 54.618 |

**Benefits of Unified SVG Modeling.** To validate the effectiveness of unified SVG modeling in improving model performance, we sample 100K SVGs from the Icon dataset and construct a multi-task training set. The set includes 100K samples for multiple-choice QA and 100K samples for SVG description in the understanding task, 100K samples for semantic color editing in the editing task, and 100K samples each for text-to-SVG and image-to-SVG in the generation task. As shown in Table 5, we compare single-task, two-task, and three-task training configurations. The results show that multi-task joint training leads to consistent performance improvements and achieves the best results across the evaluation metrics of all tasks. Overall, three-task joint training yields significant gains in most metrics, which confirms that unified SVG modeling facilitates cross-task knowledge

transfer, enables the learning of richer structural and semantic representations, and improves the generalization ability of the model.

Table 6: Ablation results on the effects of the two-stage training strategy.

| Strategy | Illustration: Text-to-SVG | | | | Illustration: Image-to-SVG | | | | Animation: Text-to-SANI | | | Animation: Video-to-SANI | | | |
| | FID ↓ | FID-C ↓ | CLIP-T2I ↑ | CLIP-I2I ↑ | DINO ↑ | SSIM ↑ | LPIPS ↓ | PSNR ↑ | FVD ↓ | CLIP-T2V ↑ | CLIP-V2V ↑ | DINO ↑ | SSIM ↑ | LPIPS ↓ | PSNR ↑ |
|---|---|---|---|---|---|---|---|---|---|---|---|---|---|---|---|
| One-stage | 68.644 | 25.671 | 19.217 | 66.400 | 0.830 | 0.503 | 0.278 | 10.345 | 101.433 | 22.291 | 69.993 | 0.867 | 0.735 | 0.245 | 13.420 |
| Two-stage | 22.397 | 5.141 | 21.116 | 74.662 | 0.924 | 0.716 | 0.188 | 14.644 | 99.474 | 22.572 | 73.162 | 0.876 | 0.754 | 0.237 | 14.168 |

**Comparison of One-stage and Two-stage Training.** To verify the effectiveness of the two-stage training strategy, we merge the data from both stages into a single-stage training scheme as a baseline. As demonstrated in Table 6, the two-stage model achieves substantial advantages on both illustration and animation generation. Especially for the long-sequenced illustration generation task, the two-stage training strategy exhibits remarkable improvements, with FID-C reduced from 25.67 to 5.14 and DINO increased from 0.830 to 0.924. More comparisons can be found in Appendix B.7.

**Benefits of SVG-specific special tokens and subword-based embedding initialization.** To assess the contribution of these two components, we sampled 50% of the icon, illustration, and chemistry training data used in stage 2 and trained three model variants: Model Raw (no special tokens), Model T (special tokens with random initialization), and Model T+E (special tokens with subword-based initialization). As shown in Table 7, Model T+E consistently delivers the best performance across both benchmarks, demonstrating that combining special tokens with subword-based embedding initialization provides the largest overall gains. Furthermore, models that incorporate special tokens (Model T and Model T+E) exhibit substantially higher success rates on illustration generation, which involves longer SVG sequences. This trend aligns with the token-compression effect shown in Figure 3(b): introducing SVG-specific tokens reduces sequence length, stabilizes training, and ultimately improves task reliability.

Table 7: Ablation studies on the effects of SVG-specific special tokens and subword-based embedding initialization. SR denotes Success Rate.

| Model | U Overall | Text-to-SVG | | | | | Image-to-SVG | | | | |
| | | SR | FID ↓ | FID-C ↓ | CLIP-T2I ↑ | CLIP-I2I ↑ | SR | DINO ↑ | SSIM ↑ | LPIPS ↓ | PSNR ↑ |
|---|---|---|---|---|---|---|---|---|---|---|---|
| | | | | | SArena-Icon | | | | | | |
| Raw | 79.3 | 92.90% | 15.788 | 5.112 | 22.767 | 76.884 | 84.30% | 0.809 | 0.493 | 0.381 | 6.571 |
| T | 79.7 | 97.62% | 11.922 | 4.578 | 23.275 | 77.374 | 89.59% | 0.919 | 0.741 | 0.166 | 14.342 |
| T + E | **80.8** | **98.42%** | **11.599** | **4.500** | **23.348** | **77.541** | **95.64%** | **0.937** | **0.795** | **0.139** | **15.837** |
| | | | | | SArena-Illustration | | | | | | |
| Raw | – | 57.22% | 68.301 | 26.108 | 18.870 | 66.222 | 50.08% | 0.735 | 0.331 | 0.412 | 6.053 |
| T | – | 69.02% | 53.080 | 22.358 | 19.174 | 66.469 | 57.37% | 0.807 | 0.448 | 0.307 | 9.424 |
| T + E | – | **78.81%** | **42.817** | **17.438** | **19.194** | **67.379** | **75.31%** | **0.862** | **0.578** | **0.257** | **11.810** |

# 6    CONCLUSIONS

In this work, we introduce the InternSVG family, a unified data-benchmark-model suite for scalable vector graphics. It integrates the large-scale and diverse SAgoge, the comprehensive benchmark SArena, and the unified MLLM InternSVG. SAgoge encompasses a broad spectrum of SVG tasks with varied difficulty levels, ranging from static icons and scientific diagrams to long-sequence illustrations and dynamic animations. SArena provides a standardized evaluation framework that assesses both mainstream MLLMs and traditional SVG generation approaches, enabling consistent comparison across tasks and domains. Through extensive experiments, we demonstrate that this unified formulation enhances SVG understanding, editing, and generation, yielding consistent improvements across domains and tasks. Our study highlights the importance of unified SVG modeling, and we hope it can serve as an insight for future research on vector-graphic reasoning and multimodal intelligence.

## REPRODUCIBILITY STATEMENT

All the results presented in this paper can be reproduced with the resources provided. We describe the model architecture and training settings in Section 4 and Section 5. The implementation of SVG-specific tokens is provided in Appendix A. Details of the dataset, including construction and preprocessing, are provided in Section 3.1, Appendix C.3, and Appendix C.4.

## ETHICS STATEMENT

Our work does not involve any sensitive personal data. All dataset elements were collected from open-source websites and publicly available sources, and we have taken care to exclude any content that could be discriminatory or violate copyright terms. We do not anticipate any adverse societal impact from our methods or datasets.

## ACKNOWLEDGEMENTS

This work was supported by the Shanghai Artificial Intelligence Laboratory, the Shanghai Committee of Science and Technology (No. 22YF1461500), and the National Natural Science Foundation of China (No. 62206046).

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

## A   IMPLEMENTATION DETAILS OF INTERNSVG

In this section, we provide the detailed implementation of the SVG-specific tokens used in InternSVG. All SVGs undergo normalization to a canonical $128 \times 128$ coordinate system. This standardization reduces spurious coordinate drift and decreases computational burden for learning absolute magnitudes, thereby mitigating coordinate hallucination and enhancing cross-task consistency.

We extend the base vocabulary with SVG-specific tokens that compress lengthy character sequences. As demonstrated in Table 8, the extended vocabulary incorporates 55 tag tokens covering structural and animation elements, including opening and closing forms such as `<svg`, `</svg>`, `<path`, `<animate`, and `<animateTransform`. Additionally, we include 42 attribute tokens encompassing geometry, styling, and animation controls with embedded quotes, such as `viewBox="`, `width="`, `x="`, and `fill="`.

For compact numerical representation, we introduce 247 integer tokens spanning $-128$ to $128$, plus 100 two-decimal and 10 one-decimal fractional tokens. This fine-grained numeric inventory enables precise geometric specification while maintaining short sequences, yielding substantial reductions in sequence length and computational requirements.

New token parameters undergo initialization, preserving the base tokenizer's semantic knowledge. Each added token's embedding equals the average of its constituent subword embeddings when segmented by the original tokenizer. This procedure applies uniformly across all token types. Following vocabulary expansion, training proceeds end-to-end without parameter freezing. The subword-derived initialization stabilizes optimization, accelerates convergence, and aligns new symbols with adjacent embedding regions, improving grammatical decoding and numerical fidelity across all tasks.

## B   RESULTS OF SARENA

### B.1   EVALUATED MODELS

**Proprietary Models.** We include several leading proprietary MLLMs in our evaluation, namely GPT-4o (Hurst et al., 2024), Gemini-2.5-Flash (Comanici et al., 2025), Claude-Sonnet-3.7 (Anthropic, 2024), Claude-Sonnet-4 (Anthropic, 2025), and Grok 3 (xAI, 2025).

**Open-Source Models.** We further benchmark a diverse set of open-source MLLMs to provide a comprehensive comparison, including Llama-3.1-8B/70B/405B, Llama-3.2-11B/90B (Dubey et al., 2024), Llama-4-Scout (Meta AI, 2025b), Llama-4-Maverick (Meta AI, 2025a), Qwen2.5-VL-7B/32B/72B (Bai et al., 2025), Keye-VL-8B (Team et al., 2025c), GLM-4.1V-9B (Hong et al., 2025), GLM-4.5V (Zeng et al., 2025), Gemma-3-12B/27B (Team et al., 2025a), Kimi-VL-A3B (Team et al., 2025b), Step3-321B (StepFun Team, 2025), InternVL3 series (Zhu et al., 2025), and InternVL3.5 series (Wang et al., 2025).

**Traditional SVG Generation Methods.** We also evaluate a range of traditional SVG generation methods, including Vectorfusion (Jain et al., 2023), IconShop (Wu et al., 2023), SVGDreamer (Xing et al., 2024), DiffVG (Li et al., 2020), LIVE (Ma et al., 2022), and VTracer (Vision Cortex, 2023).

**LLM-based SVG Generation Methods.** To thoroughly validate the effectiveness of InternSVG, we compare it against representative LLM-based SVG generation approaches on the SArena benchmark. Specifically, we include LLM4SVG (Xing et al., 2025), StarVector (Rodriguez et al., 2025), and OmniSVG (Yang et al., 2025b), which exemplify current efforts in leveraging large language models for SVG generation.

### B.2   COMPLETE EVALUATION RESULTS ON SARENA

Table 9, 10, 11, 12, 13, 14, and 15 present the complete evaluation results on the SArena benchmark across all four domains and tasks.

As shown in Tables 9, 10, and 11, InternSVG achieves substantial improvements on both Understanding and Editing tasks. For the Understanding task, our model obtains the best results across

Table 8: SVG-specific token vocabulary in InternSVG.

(a) Tag tokens

| Category | Tokens |
|---|---|
| Root | `<svg, </svg>, <defs, </defs>, <use, </use> ,/>` |
| Grouping | `<g, </g>` |
| Shapes | `<path, </path>, <rect, </rect>, <circle, </circle>, <ellipse, </ellipse>, <line, </line>, <polyline, </polyline>, <polygon, </polygon>` |
| Text | `<text, </text>, <tspan, </tspan>, <textPath, </textPath>` |
| Gradients | `<linearGradient, </linearGradient>, <radialGradient, </radialGradient>, <stop, </stop>` |
| Clipping | `<clipPath, </clipPath>, <mask, </mask>` |
| Filters | `<filter, </filter>, <feGaussianBlur, </feGaussianBlur>, <feColorMatrix, </feColorMatrix>, <feComposite, </feComposite>, <feBlend, </feBlend>` |
| Animation | `<animate, </animate>, <animateMotion, </animateMotion>, <animateTransform, </animateTransform>` |

(b) Attribute tokens

| Category | Tokens |
|---|---|
| Geometry | `width=", height=", viewBox=", x=", y=", x1=", y1=", x2=", y2=", cx=", cy=", r=", rx=", ry=", d=", points="` |
| Styling | `fill=", stroke=", stroke-width=", stroke-linecap=", stroke-linejoin=", stroke-miterlimit=", fill-rule=", opacity="` |
| Transform | `transform="` |
| Text | `font-size=", font-family=", text-anchor="` |
| Gradients | `gradientUnits=", gradientTransform=", offset=", stop-color="` |
| Animation | `begin=", dur=", repeatCount=", from=", to=", rotate=", path="` |
| Identifiers | `id=", class=", clip-path="` |

Table 9: Comparison of SVG understanding and editing performance on SArena-Icon.

| Model | Understanding | | | | | Editing | | | | |
|---|---|---|---|---|---|---|---|---|---|---|
| | Overall | Color | Geometry | Quantity | Semantic | DINO ↑ | SSIM ↑ | LPIPS ↓ | PSNR ↑ | Tokens |
| Qwen2.5-VL-7B | 52.8 | 69.3 | 50.4 | 34.9 | 56.4 | 0.909 | 0.728 | 0.192 | 25.402 | 1.0k |
| InternVL3-8B | 59.5 | 79.1 | 59.3 | 38.2 | 61.3 | 0.921 | 0.761 | 0.170 | 29.615 | 1.2k |
| Gemma-3-27B | 59.5 | 82.2 | 67.6 | 43.6 | 44.7 | 0.942 | 0.815 | 0.113 | 54.200 | 1.3k |
| Qwen2.5-VL-32B | 65.5 | 82.8 | 65.5 | 47.7 | 66.1 | 0.933 | 0.782 | 0.148 | 37.737 | 1.0k |
| Llama-4-Scout | 57.5 | 82.4 | 57.0 | 41.6 | 49.0 | 0.949 | 0.825 | 0.138 | 34.070 | 1.3k |
| Llama-4-Maverick | 64.7 | 87.5 | 62.0 | 47.2 | 62.3 | 0.966 | 0.870 | 0.109 | 46.944 | 1.3k |
| Qwen2.5-VL-72B | 63.4 | 82.4 | 65.1 | 44.6 | 61.6 | 0.961 | 0.849 | 0.124 | 41.006 | 1.2k |
| InternVL3-78B | 65.3 | 86.4 | 71.0 | 48.8 | 54.9 | 0.958 | 0.848 | 0.116 | 40.533 | 1.2k |
| GPT-4o | 71.0 | 88.2 | 78.5 | 47.5 | 69.6 | 0.968 | 0.887 | 0.088 | 55.255 | 1.2k |
| Gemini-2.5-Flash | 73.0 | 90.1 | 81.9 | 53.0 | 67.2 | 0.942 | 0.815 | 0.113 | 54.200 | 1.3k |
| Claude-Sonnet-4 | 77.1 | 91.5 | 82.4 | 53.8 | 80.6 | 0.979 | 0.915 | 0.071 | 57.595 | 1.3k |
| InternSVG 8B | **85.1** | **93.0** | **85.8** | **61.9** | **99.7** | **0.989** | **0.952** | **0.036** | **77.331** | 1.4k |

all metrics, surpassing the second-best Claude-4-Sonnet by 8 points in Overall accuracy. Especially on the Semantic subtask, the score reaches 99.7, indicating that InternSVG demonstrates strong capability in comprehending the semantics encoded in SVG code. The simple-level editing tasks primarily assess proficiency in SVG syntax, including the use of nested groups with the transform

attribute to carry out translation, rotation, and flipping operations. After training on SAgoge, our InternSVG effectively acquires these syntactic skills and achieves perfect scores in several subtasks. On the hard-level editing tasks, InternSVG attains the best performance in semantic-level color editing, consistent with the Understanding results and further confirming its strong semantic comprehension. However, in the more challenging style transfer task, the model still lags slightly behind Claude-Sonnet-4, leaving room for improvement.

Table 10: Comparison of SVG editing performance across 8 simple subtasks on SArena-Icon.

| Model | Low-level Color Editing | | | | Cropping | | | | Flipping | | | | Rotation | | | |
|---|---|---|---|---|---|---|---|---|---|---|---|---|---|---|---|---|
| | DINO ↑ | SSIM ↑ | LPIPS ↓ | PSNR ↑ | DINO ↑ | SSIM ↑ | LPIPS ↓ | PSNR ↑ | DINO ↑ | SSIM ↑ | LPIPS ↓ | PSNR ↑ | DINO ↑ | SSIM ↑ | LPIPS ↓ | PSNR ↑ |
| Qwen2.5-VL-7B | 0.958 | 0.892 | 0.061 | 73.123 | 0.870 | 0.673 | 0.270 | 10.087 | 0.852 | 0.636 | 0.313 | 9.683 | 0.919 | 0.803 | 0.152 | 47.833 |
| InternVL3-8B | 0.963 | 0.903 | 0.055 | 75.568 | 0.884 | 0.705 | 0.257 | 10.271 | 0.842 | 0.704 | 0.259 | 23.198 | 0.979 | 0.818 | 0.157 | 48.211 |
| InternVL3.5-8B | 0.999 | 0.992 | 0.007 | 88.473 | 0.881 | 0.761 | 0.195 | 11.376 | 0.905 | 0.704 | 0.241 | 13.358 | 0.886 | 0.697 | 0.246 | 21.118 |
| Gemma-3-27B | 1.000 | 1.000 | 0.000 | 99.057 | 0.885 | 0.619 | 0.297 | 14.116 | 0.995 | 0.982 | 0.008 | 96.554 | 0.991 | 0.945 | 0.041 | 85.314 |
| InternVL3.5-30B | 0.999 | 0.995 | 0.005 | 91.706 | 0.889 | 0.732 | 0.235 | 10.902 | 0.916 | 0.769 | 0.195 | 23.892 | 0.869 | 0.708 | 0.262 | 18.751 |
| Qwen2.5-VL-32B | 0.967 | 0.914 | 0.044 | 88.400 | 0.903 | 0.657 | 0.306 | 9.062 | 0.919 | 0.807 | 0.154 | 35.634 | 0.986 | 0.959 | 0.024 | 90.586 |
| Llama-4-Scout | 0.969 | 0.925 | 0.049 | 87.067 | 0.879 | 0.652 | 0.283 | 9.134 | 0.901 | 0.755 | 0.206 | 21.027 | 0.974 | 0.926 | 0.051 | 80.043 |
| Llama-4-Maverick | 0.998 | 0.996 | 0.006 | 94.874 | 0.903 | 0.677 | 0.301 | 9.404 | 0.955 | 0.914 | 0.074 | 76.565 | 0.989 | 0.967 | 0.024 | 88.142 |
| Qwen2.5-VL-72B | 0.995 | 0.986 | 0.008 | 97.542 | 0.909 | 0.668 | 0.307 | 9.174 | 0.948 | 0.874 | 0.090 | 52.671 | 0.992 | 0.949 | 0.045 | 82.266 |
| InternVL3-78B | 0.995 | 0.987 | 0.008 | 96.985 | 0.909 | 0.682 | 0.299 | 9.626 | 0.936 | 0.833 | 0.129 | 32.765 | 0.994 | 0.974 | 0.017 | 92.534 |
| InternVL3.5-241B | 0.983 | 0.956 | 0.021 | 91.262 | 0.904 | 0.763 | 0.225 | 11.763 | 0.896 | 0.754 | 0.165 | 30.961 | 0.901 | 0.783 | 0.188 | 39.965 |
| GPT-4o | 0.995 | 0.987 | 0.007 | 98.406 | 0.913 | 0.688 | 0.300 | 9.556 | 0.994 | 0.976 | 0.017 | 87.340 | 0.995 | 0.986 | 0.010 | 94.845 |
| Gemini-2.5-Flash | 1.000 | 1.000 | 9.761 | 99.057 | 0.885 | 0.619 | 0.297 | 14.116 | 0.995 | 0.982 | 0.008 | 96.554 | 0.991 | 0.945 | 0.041 | 85.314 |
| Claude-Sonnet-4 | 1.000 | 1.000 | 0.000 | 100.000 | 0.928 | 0.696 | 0.291 | 9.626 | 0.944 | 0.943 | 0.055 | 73.786 | 0.999 | 0.994 | 0.006 | 96.676 |
| InternSVG 8B | **1.000** | **1.000** | **0.000** | **100.000** | **1.000** | **1.000** | **0.000** | **100.000** | **0.996** | **0.987** | **0.005** | **98.672** | **1.000** | **1.000** | **0.000** | **99.692** |
| Model | Scaling | | | | Adding Stroke | | | | Translation | | | | Transparency | | | |
| | DINO ↑ | SSIM ↑ | LPIPS ↓ | PSNR ↑ | DINO ↑ | SSIM ↑ | LPIPS ↓ | PSNR ↑ | DINO ↑ | SSIM ↑ | LPIPS ↓ | PSNR ↑ | DINO ↑ | SSIM ↑ | LPIPS ↓ | PSNR ↑ |
| Qwen2.5-VL-7B | 0.902 | 0.653 | 0.262 | 12.466 | 0.917 | 0.728 | 0.180 | 25.767 | 0.908 | 0.634 | 0.295 | 13.257 | 0.966 | 0.889 | 0.073 | 50.893 |
| InternVL3-8B | 0.923 | 0.684 | 0.231 | 12.403 | 0.933 | 0.791 | 0.150 | 35.333 | 0.916 | 0.708 | 0.222 | 27.231 | 0.982 | 0.954 | 0.026 | 67.912 |
| InternVL3.5-8B | 0.932 | 0.710 | 0.234 | 16.638 | 0.936 | 0.721 | 0.162 | 20.350 | 0.917 | 0.660 | 0.276 | 12.508 | 0.989 | 0.967 | 0.024 | 59.713 |
| Gemma-3-27B | 0.943 | 0.846 | 0.100 | 67.280 | 0.968 | 0.857 | 0.116 | 40.216 | 0.962 | 0.896 | 0.045 | 82.705 | 0.883 | 0.687 | 0.141 | 63.444 |
| InternVL3.5-30B | 0.930 | 0.693 | 0.236 | 14.118 | 0.949 | 0.769 | 0.135 | 27.933 | 0.947 | 0.746 | 0.222 | 32.944 | 0.992 | 0.968 | 0.024 | 63.038 |
| Qwen2.5-VL-32B | 0.917 | 0.673 | 0.236 | 19.639 | 0.932 | 0.739 | 0.139 | 33.796 | 0.934 | 0.748 | 0.191 | 31.632 | 0.980 | 0.949 | 0.029 | 80.879 |
| Llama-4-Scout | 0.925 | 0.705 | 0.226 | 18.068 | 0.960 | 0.840 | 0.104 | 38.360 | 0.926 | 0.686 | 0.251 | 18.387 | 0.983 | 0.957 | 0.028 | 66.797 |
| Llama-4-Maverick | 0.927 | 0.776 | 0.194 | 23.361 | 0.970 | 0.886 | 0.073 | 52.249 | 0.956 | 0.741 | 0.226 | 31.710 | 0.996 | 0.991 | 0.006 | 94.987 |
| Qwen2.5-VL-72B | 0.901 | 0.678 | 0.267 | 11.492 | 0.965 | 0.875 | 0.105 | 44.055 | 0.951 | 0.704 | 0.256 | 18.695 | 0.995 | 0.992 | 0.010 | 72.101 |
| InternVL3-78B | 0.931 | 0.695 | 0.238 | 12.792 | 0.947 | 0.790 | 0.145 | 37.317 | 0.957 | 0.831 | 0.134 | 46.221 | 0.992 | 0.984 | 0.015 | 68.573 |
| InternVL3.5-241B | 0.919 | 0.661 | 0.245 | 11.857 | 0.948 | 0.762 | 0.136 | 27.335 | 0.928 | 0.750 | 0.160 | 25.850 | 0.956 | 0.882 | 0.059 | 64.399 |
| GPT-4o | 0.947 | 0.811 | 0.163 | 45.845 | 0.966 | 0.864 | 0.093 | 48.913 | 0.982 | 0.928 | 0.060 | 72.016 | 0.990 | 0.977 | 0.014 | 85.619 |
| Gemini-2.5-Flash | 0.943 | 0.846 | 0.100 | 67.280 | 0.968 | 0.857 | 0.116 | 40.216 | 0.962 | 0.896 | 0.045 | 82.705 | 0.883 | 0.687 | 0.141 | 63.444 |
| Claude-Sonnet-4 | 0.953 | 0.833 | 0.138 | 50.330 | 0.982 | 0.907 | 0.055 | 51.913 | 0.999 | 0.997 | 0.002 | 87.758 | 0.999 | 1.000 | 0.000 | 97.535 |
| InternSVG 8B | **0.999** | **1.000** | **0.000** | **98.655** | **1.000** | **1.000** | **0.000** | **99.488** | **1.000** | **1.000** | **0.000** | **100.000** | **1.000** | **1.000** | **0.000** | **99.968** |

Table 11: Comparison of SVG editing performance across 2 hard subtasks on SArena-Icon.

| Model | Semantic-level Color Editing | | | | Style Transfer | | | |
|---|---|---|---|---|---|---|---|---|
| | DINO ↑ | SSIM ↑ | LPIPS ↓ | PSNR ↑ | DINO ↑ | SSIM ↑ | LPIPS ↓ | PSNR ↑ |
| Qwen2.5-VL-7B | 0.919 | 0.768 | 0.166 | 23.902 | 0.889 | 0.658 | 0.193 | 11.940 |
| InternVL3-8B | 0.903 | 0.728 | 0.184 | 22.071 | 0.917 | 0.728 | 0.158 | 13.457 |
| Gemma-3-27B | 0.981 | 0.920 | 0.072 | 53.068 | 0.869 | 0.591 | 0.210 | 12.174 |
| Qwen2.5-VL-32B | 0.926 | 0.769 | 0.158 | 28.290 | 0.910 | 0.723 | 0.162 | 14.283 |
| Llama-4-Scout | 0.964 | 0.860 | 0.120 | 27.852 | 0.963 | 0.848 | 0.119 | 15.417 |
| Llama-4-Maverick | 0.975 | 0.891 | 0.099 | 41.222 | 0.969 | 0.855 | 0.105 | 16.765 |
| Qwen2.5-VL-72B | 0.975 | 0.888 | 0.100 | 42.759 | 0.957 | 0.836 | 0.113 | 16.771 |
| InternVL3-78B | 0.955 | 0.857 | 0.105 | 27.033 | 0.912 | 0.705 | 0.175 | 13.429 |
| GPT-4o | 0.972 | 0.912 | 0.073 | 54.651 | 0.952 | 0.819 | 0.117 | 18.173 |
| Gemini-2.5-Flash | 0.981 | 0.920 | 0.072 | 53.068 | 0.869 | 0.591 | 0.210 | 12.174 |
| Claude-Sonnet-4 | 0.991 | 0.944 | 0.050 | 56.741 | **0.976** | **0.867** | **0.097** | **18.374** |
| InternSVG 8B | **0.996** | **0.959** | **0.041** | **69.875** | 0.952 | 0.808 | 0.139 | 18.100 |

The complete results for both Text-to-SVG and Image-to-SVG tasks on icons and illustrations are reported in Table 12 and Table 13. InternSVG consistently outperforms prior SVG generation approaches across all major metrics. Compared with large-scale open-source models such as Llama-4-Maverick and GLM-4.5V, it achieves lower FID, FID-C, and higher CLIP-I2I scores, with only a slight gap on CLIP-T2I. For Image-to-SVG, InternSVG achieves the best performance across all metrics, highlighting its strength in visual fidelity and semantic consistency.

Table 12: Comparison of SVG generation performance on SArena-Icon (Text-to-SVG and Image-to-SVG).

| Model | Text-to-SVG | | | | | | Image-to-SVG | | | | | |
|---|---|---|---|---|---|---|---|---|---|---|---|---|
| | FID ↓ | FID-C ↓ | CLIP-T2I ↑ | CLIP-I2I ↑ | SR | Tokens | DINO ↑ | SSIM ↑ | LPIPS ↓ | PSNR ↑ | SR | Tokens |
| Traditional SVG methods | | | | | | | | | | | | |
| IconShop | 32.288 | 17.919 | 20.894 | 70.922 | 86.51% | 1.5k | – | – | – | – | – | – |
| VectorFusion | 16.594 | 17.394 | 22.992 | 70.308 | 100.0% | 33k | – | – | – | – | – | – |
| SVGDreamer | 26.612 | 33.312 | 20.329 | 69.975 | 100.0% | 132k | – | – | – | – | – | – |
| DiffVG | – | – | – | – | – | – | 0.869 | 0.927 | 0.097 | 23.614 | 100.0% | 17k |
| LIVE | – | – | – | – | – | – | 0.973 | 0.986 | 0.024 | 35.419 | 100.0% | 18k |
| VTracer | – | – | – | – | – | – | 0.966 | 0.875 | 0.054 | 21.748 | 100.0% | 4.4k |
| Large language models | | | | | | | | | | | | |
| Llama-3.1-8B | 19.428 | 11.247 | 21.863 | 71.859 | 97.54% | 280 | – | – | – | – | – | – |
| Qwen2.5-VL-7B | 24.781 | 15.454 | 21.538 | 71.384 | 90.72% | 249 | 0.781 | 0.506 | 0.378 | 6.534 | 86.35% | 281 |
| Keye-VL-8B | 21.961 | 14.393 | 21.557 | 71.167 | 93.71% | 227 | 0.801 | 0.531 | 0.368 | 6.939 | 90.95% | 286 |
| GLM-4.1V-9B | 22.684 | 10.447 | 22.562 | 73.197 | 94.11% | 269 | 0.820 | 0.539 | 0.345 | 7.329 | 84.12% | 289 |
| InternVL3-8B | 23.061 | 14.303 | 21.897 | 71.450 | 93.96% | 269 | 0.812 | 0.557 | 0.361 | 7.220 | 94.76% | 256 |
| InternVL3.5-8B | 17.357 | 7.128 | 21.888 | 75.005 | 91.72% | 1.0k | 0.852 | 0.618 | 0.291 | 8.737 | 91.49% | 692 |
| Llama-3.2-11B | 28.156 | 14.345 | 21.711 | 71.485 | 86.18% | 261 | 0.759 | 0.467 | 0.389 | 5.908 | 81.46% | 216 |
| Gemma-3-12B | 17.137 | 10.409 | 22.023 | 71.622 | 98.40% | 290 | 0.821 | 0.576 | 0.352 | 7.632 | 95.29% | 360 |
| InternVL3-14B | 18.996 | 13.224 | 22.066 | 71.493 | 98.20% | 227 | 0.825 | 0.562 | 0.359 | 7.343 | 96.77% | 216 |
| InternVL3.5-14B | 15.897 | 5.985 | 22.351 | 75.912 | 92.82% | 1.0k | 0.855 | 0.607 | 0.308 | 8.455 | 93.00% | 832 |
| Kimi-VL-A3B | 30.807 | 16.996 | 21.439 | 70.536 | 86.18% | 228 | 0.798 | 0.562 | 0.362 | 7.179 | 92.20% | 245 |
| InternVL3.5-20B | 16.779 | 5.595 | 23.017 | 77.455 | 90.74% | 802 | 0.908 | 0.706 | 0.196 | 12.747 | 90.25% | 1.0k |
| Gemma-3-27B | 15.145 | 9.303 | 22.526 | 73.277 | 99.37% | 249 | 0.826 | 0.595 | 0.354 | 7.833 | 99.50% | 267 |
| InternVL3.5-30B | 16.307 | 5.843 | 22.483 | 76.397 | 92.08% | 961 | 0.883 | 0.653 | 0.270 | 9.636 | 96.37% | 783 |
| Qwen2.5-VL-32B | 20.043 | 10.393 | 22.783 | 73.228 | 93.98% | 317 | 0.836 | 0.562 | 0.357 | 7.503 | 96.92% | 309 |
| InternVL3-38B | 18.014 | 11.042 | 22.795 | 73.077 | 97.21% | 251 | 0.829 | 0.549 | 0.351 | 7.305 | 93.40% | 230 |
| InternVL3.5-38B | 14.558 | 5.218 | 22.546 | 76.487 | 94.46% | 1.0k | 0.861 | 0.611 | 0.320 | 8.393 | 96.24% | 1.0k |
| Grok-3 | 21.967 | 8.694 | 24.122 | 76.797 | 84.17% | 346 | – | – | – | – | – | – |
| Llama-3.1-70B | 18.032 | 8.300 | 22.747 | 73.876 | 96.08% | 255 | – | – | – | – | – | – |
| Llama-3.1-405B | 16.794 | 8.390 | 22.822 | 73.920 | 96.87% | 236 | – | – | – | – | – | – |
| DeepSeek-V3 | 24.990 | 8.803 | 23.790 | 76.470 | 82.27% | 251 | – | – | – | – | – | – |
| GPT-4o | 15.178 | 6.763 | 24.617 | 77.742 | 99.47% | 246 | 0.874 | 0.616 | 0.316 | 8.435 | 98.02% | 231 |
| Gemini-2.5-Flash | 16.720 | 5.208 | 24.658 | 78.218 | 97.34% | 451 | 0.876 | 0.587 | 0.316 | 8.324 | 90.50% | 533 |
| Claude-Sonnet-3.7 | 14.383 | 3.499 | 25.294 | 80.786 | 99.15% | 417 | 0.909 | 0.647 | 0.290 | 9.259 | 98.34% | 389 |
| Claude-Sonnet-4 | 15.840 | 4.291 | **25.421** | 80.579 | 99.57% | 444 | 0.915 | 0.665 | 0.276 | 9.855 | 99.63% | 541 |
| Llama-3.2-90B | 19.309 | 8.550 | 22.841 | 74.006 | 94.69% | 249 | 0.757 | 0.437 | 0.377 | 5.777 | 67.75% | 192 |
| Llama-4-Scout | 17.908 | 9.382 | 22.849 | 73.563 | 96.03% | 256 | 0.844 | 0.582 | 0.346 | 7.736 | 97.61% | 246 |
| Llama-4-Maverick | 14.931 | 6.526 | 23.570 | 75.816 | 98.12% | 265 | 0.863 | 0.596 | 0.329 | 8.027 | 97.75% | 255 |
| GLM-4.5V | 16.641 | 5.093 | 24.450 | 78.349 | 97.32% | 372 | 0.872 | 0.627 | 0.315 | 8.666 | 98.24% | 322 |
| Step3-321B | 20.061 | 9.706 | 23.053 | 74.184 | 86.73% | 308 | 0.834 | 0.555 | 0.340 | 7.516 | 86.53% | 301 |
| Qwen2.5-VL-72B | 15.948 | 9.875 | 22.946 | 73.681 | 98.50% | 275 | 0.837 | 0.584 | 0.346 | 7.834 | 99.04% | 372 |
| InternVL3-78B | 17.580 | 10.596 | 22.805 | 73.123 | 98.05% | 252 | 0.850 | 0.584 | 0.339 | 7.802 | 98.95% | 234 |
| InternVL3.5-241B | 11.265 | 4.426 | 22.534 | 76.807 | 96.99% | 991 | 0.882 | 0.642 | 0.291 | 9.187 | 98.15% | 914 |
| StarVector 8B | – | – | – | – | – | – | 0.871 | 0.623 | 0.206 | 13.595 | 72.51% | 951 |
| LLM4SVG 7B | 21.939 | 8.611 | 19.458 | 70.726 | 90.95% | 705 | 0.748 | 0.472 | 0.409 | 5.375 | 95.53% | 485 |
| OmniSVG 3B | 28.292 | 11.318 | 21.679 | 74.831 | 99.68% | 1.7k | 0.894 | 0.756 | 0.186 | 12.669 | 99.97% | 2.4k |
| InternSVG 8B | **8.715** | **1.876** | 23.916 | **80.911** | 97.24% | 1.0k | **0.949** | **0.811** | **0.127** | **18.226** | 94.45% | 1.3k |

Table 14 reports the complete evaluation results on SArena-Chemistry for both Text-to-SVG and Image-to-SVG tasks. In this benchmark, common names and IUPAC names of chemical compounds are used as instructions to guide models to generate structural formulas. The results indicate that most existing MLLMs perform poorly in Text-to-SVG generation due to their limited training exposure to scientific graphics. For example, Qwen2.5-VL-72B obtains FID scores around 32 and relatively low CLIP-I2I scores. In contrast, InternSVG achieves the lowest FID (9.974) and FID-C (0.877) and the highest CLIP-I2I (93.931), surpassing all baselines by a large margin. On Image-to-SVG, InternSVG also outperforms both open-source and proprietary models, achieving the best results across DINO (0.994), SSIM (0.873), and PSNR (17.722). These results highlight the strong capability of InternSVG in handling domain-specific scientific graphics.

For animation generation, InternSVG achieves strong performance on both Text-to-SANI and Video-to-SANI tasks according to Table 15. In the Text-to-SANI task, it attains the lowest FVD of 99.474 and ranks second on CLIP-T2V and CLIP-V2V, only slightly behind Claude-Sonnet-4. In the Video-to-SANI task, InternSVG demonstrates performance comparable to or even surpassing that

Table 13: Comparison of SVG generation performance on SArena-Illustration (Text-to-SVG and Image-to-SVG).

| Model | Text-to-SVG | | | | | | Image-to-SVG | | | | | |
|---|---|---|---|---|---|---|---|---|---|---|---|---|
| | FID ↓ | FID-C ↓ | CLIP-T2I ↑ | CLIP-I2I ↑ | SR | Tokens | DINO ↑ | SSIM ↑ | LPIPS ↓ | PSNR ↑ | SR | Tokens |
| **Traditional SVG methods** | | | | | | | | | | | | |
| VectorFusion | 28.198 | 15.661 | 22.638 | 71.220 | 100.0% | 33k | – | – | – | – | – | – |
| SVGDreamer | 32.410 | 23.427 | 21.026 | 69.775 | 100.0% | 132k | – | – | – | – | – | – |
| DiffVG | – | – | – | – | – | – | 0.870 | 0.886 | 0.138 | 21.605 | 100.0% | 17k |
| LIVE | – | – | – | – | – | – | 0.963 | 0.948 | 0.078 | 29.055 | 100.0% | 18k |
| VTracer | – | – | – | – | – | – | 0.965 | 0.879 | 0.093 | 21.754 | 100.0% | 10k |
| **Large language models** | | | | | | | | | | | | |
| Qwen2.5-VL-7B | 37.903 | 28.455 | 18.069 | 61.928 | 88.81% | 756 | 0.739 | 0.513 | 0.413 | 7.732 | 83.46% | 1.2k |
| InternVL3-8B | 36.736 | 25.682 | 18.493 | 61.964 | 91.45% | 493 | 0.772 | 0.569 | 0.397 | 8.542 | 92.70% | 716 |
| InternVL3.5-8B | 70.837 | 35.776 | 18.095 | 63.357 | 51.52% | 3.6k | 0.721 | 0.306 | 0.410 | 5.283 | 48.43% | 2.5k |
| InternVL3.5-14B | 65.967 | 34.912 | 18.131 | 63.496 | 51.97% | 3.5k | 0.722 | 0.296 | 0.414 | 5.130 | 47.53% | 2.8k |
| Gemma-3-27B | 27.838 | 13.766 | 21.486 | 67.255 | 98.80% | 613 | 0.824 | 0.617 | 0.379 | 9.920 | 98.45% | 764 |
| InternVL3.5-30B | 68.438 | 33.285 | 18.354 | 63.910 | 54.57% | 3.8k | 0.739 | 0.331 | 0.404 | 5.778 | 52.37% | 3.0k |
| Qwen2.5-VL-32B | 32.115 | 17.804 | 19.773 | 64.555 | 94.70% | 779 | 0.816 | 0.591 | 0.382 | 9.297 | 95.25% | 828 |
| InternVL3.5-38B | 42.172 | 21.556 | 18.221 | 65.511 | 78.21% | 4.3k | 0.755 | 0.393 | 0.400 | 6.540 | 64.07% | 3.8k |
| Llama-4-Scout | 35.489 | 18.647 | 20.299 | 64.182 | 91.80% | 524 | 0.807 | 0.599 | 0.360 | 9.549 | 92.80% | 574 |
| Llama-4-Maverick | 30.835 | 14.831 | 21.872 | 67.366 | 97.30% | 551 | 0.839 | 0.644 | 0.340 | 10.469 | 98.20% | 608 |
| Qwen2.5-VL-72B | 29.521 | 18.407 | 20.923 | 65.349 | 98.60% | 527 | 0.808 | 0.628 | 0.363 | 9.900 | 98.45% | 886 |
| InternVL3-78B | 30.457 | 19.195 | 20.577 | 64.826 | 97.40% | 454 | 0.830 | 0.638 | 0.348 | 9.985 | 99.35% | 514 |
| InternVL3.5-241B | 43.339 | 23.061 | 18.191 | 65.689 | 75.86% | 2.9k | 0.792 | 0.480 | 0.378 | 8.093 | 77.26% | 3.1k |
| GPT-4o | 28.124 | 14.150 | 23.637 | 70.696 | 99.75% | 473 | 0.850 | 0.663 | 0.327 | 10.723 | 99.35% | 484 |
| Gemini-2.5-Flash | 28.865 | 8.894 | **24.800** | **74.796** | 96.50% | 1.2k | 0.829 | 0.516 | 0.359 | 9.091 | 77.41% | 1.8k |
| Claude-Sonnet-4 | 27.294 | 7.640 | 23.094 | 74.525 | 98.85% | 1.0k | 0.901 | 0.670 | 0.305 | 11.731 | 99.25% | 1.3k |
| StarVector 8B | – | – | – | – | – | – | 0.650 | 0.070 | 0.447 | 1.990 | 8.050% | 2.6k |
| LLM4SVG 7B | 48.704 | 29.568 | 15.468 | 62.933 | 79.36% | 1.2k | 0.713 | 0.494 | 0.413 | 6.221 | 96.60% | 476 |
| OmniSVG 3B | 42.756 | 22.885 | 16.861 | 64.815 | 99.75% | 4.5k | 0.797 | 0.656 | 0.330 | 10.433 | 100.0% | 6.7k |
| InternSVG 8B | **22.397** | **5.141** | 21.116 | 74.662 | 100.0% | 8.1k | **0.924** | **0.716** | **0.188** | **14.644** | 90.01% | 7.7k |

Table 14: Comparison of SVG generation performance on SArena-Chemistry (Text-to-SVG and Image-to-SVG).

| Model | Text-to-SVG | | | | | Image-to-SVG | | | | | |
|---|---|---|---|---|---|---|---|---|---|---|---|
| | FID ↓ | FID-C ↓ | CLIP-I2I ↑ | SR | Tokens | DINO ↑ | SSIM ↑ | LPIPS ↓ | PSNR ↑ | SR | Tokens |
| Qwen2.5-VL-7B | 56.248 | 73.698 | 51.814 | 65.60% | 907 | 0.769 | 0.468 | 0.274 | 7.501 | 60.97% | 996 |
| InternVL3-8B | 33.613 | 61.675 | 56.856 | 90.81% | 910 | 0.865 | 0.783 | 0.203 | 13.840 | 96.27% | 805 |
| Gemma-3-27B | 29.937 | 49.967 | 60.776 | 92.07% | 776 | 0.887 | 0.823 | 0.190 | 14.959 | 98.63% | 683 |
| Qwen2.5-VL-32B | 53.047 | 56.431 | 58.428 | 68.30% | 1.2k | 0.821 | 0.570 | 0.225 | 10.005 | 67.03% | 900 |
| Llama-4-Scout | 33.781 | 46.584 | 62.522 | 87.95% | 849 | 0.866 | 0.734 | 0.205 | 12.984 | 89.08% | 624 |
| Llama-4-Maverick | 26.844 | 31.924 | 69.643 | 93.97% | 747 | 0.908 | 0.798 | 0.173 | 14.977 | 93.27% | 687 |
| Qwen2.5-VL-72B | 32.307 | 44.540 | 63.931 | 90.34% | 620 | 0.846 | 0.647 | 0.215 | 12.106 | 76.76% | 716 |
| InternVL3-78B | 29.216 | 40.080 | 65.969 | 92.07% | 698 | 0.911 | 0.813 | 0.177 | 15.375 | 96.24% | 545 |
| GPT-4o | 24.505 | 19.297 | 76.599 | 97.37% | 640 | 0.920 | 0.791 | 0.174 | 14.673 | 92.21% | 533 |
| Gemini-2.5-Flash | 27.708 | 21.777 | 75.897 | 92.07% | 1.4k | 0.934 | 0.817 | 0.155 | 15.539 | 94.44% | 1.1k |
| Claude-Sonnet-4 | 21.252 | 15.240 | 78.308 | 97.90% | 1.2k | 0.957 | 0.871 | **0.132** | 17.554 | 99.37% | 956 |
| StarVector 8B | – | – | – | – | – | 0.977 | 0.841 | 0.147 | 17.419 | 96.70% | 1.2k |
| InternSVG 8B | **9.974** | **0.877** | **93.931** | **99.93%** | 981 | **0.994** | **0.873** | 0.138 | **17.722** | **100.0%** | 931 |

of Claude-Sonnet-4. Its SSIM score is only lower by 0.006, while it achieves higher results on all other metrics.

## B.3 RESULTS ON SGP-BENCH

To further validate the effectiveness of SAgoge in enhancing model capabilities for SVG modeling, we conduct comparative experiments on SGP-Bench (Qiu et al., 2024), a benchmark specifically designed to evaluate semantic and structural understanding of symbolic graphic programs (*e.g.*, SVGs). As shown in Table 16, our InternSVG achieves an overall accuracy of 72.3%, substantially outperforming existing open-source models. In particular, it surpasses the second-best open-source MLLM, GLM-4.5V, by 6.2 percentage points. When compared with proprietary systems, InternSVG also demonstrates advantages, achieving 1.3 percentage points higher overall accuracy

Table 15: Comparison of SVG generation performance on SArena-Animation (Text-to-SANI and Video-to-SANI).

| Model | Text-to-SANI | | | | | Video-to-SANI | | | | | |
|---|---|---|---|---|---|---|---|---|---|---|---|
| | FVD ↓ | CLIP-T2V ↑ | CLIP-V2V ↑ | SR | Tokens | DINO ↑ | SSIM ↑ | LPIPS ↓ | PSNR ↑ | SR | Tokens |
| Qwen2.5-VL-7B | 214.379 | 19.118 | 50.649 | 94.25% | 296 | 0.787 | 0.716 | 0.273 | 11.758 | 95.63% | 423 |
| InternVL3-8B | 310.066 | 17.017 | 43.856 | 74.01% | 433 | 0.780 | 0.612 | 0.286 | 9.883 | 82.54% | 415 |
| Gemma-3-27B | 159.119 | 21.105 | 59.309 | 98.41% | 533 | 0.824 | 0.733 | 0.265 | 12.290 | 98.41% | 516 |
| Qwen2.5-VL-32B | 128.299 | 20.535 | 59.188 | 98.21% | 537 | 0.823 | 0.696 | 0.273 | 11.417 | 96.83% | 505 |
| Llama-4-Scout | 167.932 | 21.014 | 62.929 | 100.0% | 505 | 0.831 | 0.742 | 0.259 | 12.427 | 99.80% | 426 |
| Llama-4-Maverick | 141.470 | 22.304 | 67.615 | 100.0% | 563 | 0.841 | 0.754 | 0.246 | 12.858 | 100.0% | 447 |
| Qwen2.5-VL-72B | 151.682 | 20.376 | 59.454 | 97.62% | 433 | 0.834 | 0.721 | 0.261 | 11.931 | 98.61% | 402 |
| InternVL3-78B | 169.159 | 20.263 | 60.896 | 97.42% | 409 | 0.828 | 0.704 | 0.264 | 11.336 | 97.82% | 385 |
| GPT-4o | 155.393 | 22.808 | 70.608 | 99.80% | 404 | 0.860 | 0.743 | 0.250 | 12.260 | 99.80% | 400 |
| Gemini-2.5-Flash | 151.983 | 22.239 | 66.554 | 93.85% | 986 | 0.847 | 0.701 | 0.257 | 12.015 | 91.87% | 917 |
| Claude-Sonnet-4 | 169.484 | **24.070** | **74.179** | 100.0% | 907 | 0.867 | **0.760** | 0.240 | 13.189 | 99.80% | 866 |
| InternSVG 8B | **99.474** | 22.572 | 73.162 | 99.01% | 812 | **0.876** | 0.754 | **0.237** | **14.168** | 97.82% | 888 |

Table 16: Comparison of SVG understanding performance on SGP-Bench.

| Model | Semantics ↑ | Count ↑ | Color ↑ | Shape ↑ | Reasoning ↑ | Overall ↑ |
|---|---|---|---|---|---|---|
| Gemma-1.1-2B | 32.1 | 33.3 | 25.0 | 35.6 | 28.7 | 31.7 |
| InternLM2.5-7B | 27.3 | 31.7 | 59.8 | 51.5 | 28.2 | 42.1 |
| Keye-VL-8B | 41.4 | 47.5 | 71.4 | 54.9 | 40.6 | 52.2 |
| GLM-4.1V-9B | 41.6 | 55.6 | 79.1 | 61.5 | 40.0 | 57.1 |
| InternVL3-8B | 33.7 | 46.5 | 69.8 | 59.1 | 36.1 | 50.6 |
| Gemma-3-12B | 24.8 | 30.8 | 47.2 | 25.7 | 22.8 | 30.5 |
| DeepSeek-Coder-V2-16B | 30.9 | 37.9 | 63.7 | 54.8 | 26.8 | 45.1 |
| InternVL3-14B | 38.2 | 52.9 | 74.4 | 54.1 | 41.7 | 52.9 |
| Kimi-VL-A3B-2506 | 31.1 | 41.5 | 67.0 | 47.4 | 32.4 | 44.9 |
| Gemma-3-27B | 36.7 | 51.4 | 76.3 | 62.1 | 39.4 | 54.7 |
| Qwen2.5-VL-32B | 40.0 | 55.7 | 76.3 | 61.2 | 43.9 | 56.5 |
| InternVL3-38B | 40.8 | 58.7 | 82.2 | 63.6 | 43.9 | 59.1 |
| GPT-4o | 45.9 | 56.8 | 87.3 | 75.2 | 50.4 | 64.8 |
| Gemini-2.5-Flash | 53.8 | 57.8 | 88.1 | 75.6 | 55.5 | 67.6 |
| Claude-Sonnet-4 | **55.9** | 67.6 | **89.5** | 79.0 | **58.9** | 71.5 |
| GLM-4.5V | 47.3 | 63.7 | 87.3 | 72.3 | 55.8 | 66.1 |
| Qwen2.5-VL-72B | 40.2 | 55.1 | 80.1 | 62.0 | 41.1 | 57.1 |
| InternVL3-78B | 41.0 | 59.1 | 84.0 | 65.2 | 47.0 | 60.3 |
| Step3-321B-A38B | 35.9 | 54.0 | 82.8 | 63.2 | 38.6 | 56.5 |
| InternSVG 8B | 54.6 | **70.7** | 85.5 | **82.4** | 57.5 | **72.3** |

than Claude-Sonnet-4, while also leading on the Count and Shape subtasks. The strong performance on SGP-Bench demonstrates the effectiveness of SAgoge in enhancing SVG modeling.

### B.4 RESULTS ON SVG-STACK

We evaluate the performance of InternSVG on SVG-Stack (Rodriguez et al., 2025), a benchmark that focuses on the Image-to-SVG task. As shown in Table 17, InternSVG achieves the best performance on Dino, LPIPS, and MSE, while using significantly fewer output tokens compared to StarVector (1.2k vs. 3.7k–5.3k), and obtains comparable SSIM scores.

### B.5 RESULTS ON UNISVG

UniSVG (Li et al., 2025) is a comprehensive benchmark designed to evaluate both SVG understanding and generation capabilities. On UniSVG, InternSVG achieves the best performance with

Table 17: Comparison of Image-to-SVG results on SVG-Stack.

| Method | Dino ↑ | LPIPS ↓ | SSIM ↑ | MSE ↓ | Tokens |
|---|---|---|---|---|---|
| GPT-4V | 0.852 | 0.317 | 0.711 | 0.195 | 443 |
| StarVector 1B | 0.926 | 0.149 | 0.840 | 0.078 | 3.7k |
| StarVector 8B | 0.966 | 0.058 | **0.947** | 0.026 | 5.3k |
| InternSVG 8B | **0.968** | **0.052** | 0.916 | **0.023** | 1.2k |

a Final Score of 0.826, substantially outperforming all baselines. In particular, it attains a 0.939 ISVGEN score on the Image-to-SVG task, which shows a 0.166 improvement over the second-best method, and also obtains the highest TSVGEN score of 0.741 on the Text-to-SVG task. For the SVG understanding task, InternSVG achieves the highest accuracy of 0.625 on the Hard set, while demonstrating comparable performance on other metrics. Together with the strong results on SGP-Bench and SVG-Stack, these findings indicate that the benefits of InternSVG are not limited to SArena or any related data source. Instead, they demonstrate that the SAgoge dataset and the InternSVG framework provide robust and broadly generalizable improvements across diverse SVG understanding and generation tasks.

Table 18: Comparison of model performance on UniSVG.

| Model Name | Final Score | SVGEN | | | | | | | | SVGUN | | | |
|---|---|---|---|---|---|---|---|---|---|---|---|---|---|
| | | ISVGEN | | | | TSVGEN | | | | Accuracy | | Bert | |
| | | Low-Level | | High-Level | Score | Low-Level | | High-level | Score | Easy-Acc ↑ | Hard-Acc↑ | Bert ↑ | SBert ↑ |
| | | SSIM ↑ | LPIPS ↓ | CLIP Score ↑ | | SSIM ↑ | LPIPS ↓ | CLIP Score ↑ | | | | | |
| LLaMA-3.2-3B | 0.567 | 0.563 | 0.674 | 0.690 | 0.592 | 0.491 | 0.616 | 0.772 | 0.638 | 0.347 | 0.201 | -0.291 | 0.455 |
| Qwen2.5-VL-7B | 0.614 | 0.564 | 0.614 | 0.738 | 0.633 | 0.538 | 0.619 | 0.764 | 0.642 | 0.543 | 0.571 | 0.082 | 0.596 |
| GPT-4V | 0.650 | 0.557 | 0.582 | 0.740 | 0.638 | 0.620 | 0.531 | 0.816 | 0.712 | 0.893 | 0.448 | 0.211 | 0.520 |
| Claude 3.7 | 0.722 | 0.622 | 0.473 | 0.855 | 0.743 | **0.622** | 0.503 | 0.852 | 0.735 | 0.863 | 0.624 | 0.322 | 0.709 |
| LLaMA-3.2-3B-SFT | 0.732 | 0.722 | 0.378 | 0.843 | 0.775 | 0.617 | 0.511 | 0.812 | 0.709 | **0.990** | 0.560 | 0.511 | 0.735 |
| Qwen2.5-VL-7B-SFT | 0.752 | 0.725 | 0.368 | 0.836 | 0.773 | 0.621 | **0.479** | 0.853 | 0.740 | 0.983 | 0.604 | **0.574** | **0.827** |
| InternSVG 8B | **0.826** | **0.890** | **0.100** | **0.970** | **0.939** | 0.620 | 0.490 | **0.860** | **0.741** | 0.943 | **0.625** | 0.564 | 0.821 |

## B.6 QUANTITATIVE EVALUATION OF SVG CODE QUALITY

Table 19: Comparison of generated SVG code quality on SArena-Icon.

| Model | Path | Rect | Circle | Polygon | Line | Ellipse | Polyline | Time |
|---|---|---|---|---|---|---|---|---|
| GT | 2.98 | 0.35 | 0.29 | 0.17 | 0.10 | 0.06 | 0.02 | – |
| SVGDreamer | 256.00 | 0.00 | 0.00 | 0.00 | 0.00 | 0.00 | 0.00 | 289s |
| VectorFusion | 64.00 | 0.00 | 0.00 | 0.00 | 0.00 | 0.00 | 0.00 | 71s |
| IconShop | 6.19 | 0.00 | 0.00 | 0.00 | 0.00 | 0.00 | 0.00 | 5.7s |
| LLM4SVG | 2.81 | 0.03 | 0.17 | 0.00 | 0.00 | 0.01 | 0.00 | 9.7s |
| OminiSVG | 4.30 | 0.00 | 0.00 | 0.00 | 0.00 | 0.00 | 0.00 | 12.3s |
| InternSVG 8B | 4.48 | 0.25 | 0.20 | 0.17 | 0.07 | 0.02 | 0.01 | 6.9s |

To complement visual metrics, we conduct a quantitative evaluation of the structural quality of generated SVG code. We utilize the SArena-Icon benchmark, which contains 6,013 real human-designed icons, to ensure that the analysis reflects the structural properties expected in real design scenarios. Inspired by the complexity metrics proposed in Svgenius (Chen et al., 2025), we measure path count via XML parsing, assess the conciseness of command structures, and compare the average inference time across models on Text-to-SVG tasks. We also measure the average counts of SVG primitives to evaluate each baseline's structural expressiveness more comprehensively.

As shown in Table 19, InternSVG demonstrates strong performance in structural expressiveness, editability, and inference efficiency. InternSVG natively supports the full set of SVG primitives, avoiding the heavy Bézier-curve approximations used by optimization-based methods such as SVG-Dreamer (Xing et al., 2024) and VectorFusion (Jain et al., 2023). This primitive-level representation has also been highlighted as essential in recent work such as StarVector (Rodriguez et al., 2025), enabling InternSVG to model diverse geometric structures more naturally and accurately. Furthermore, InternSVG produces highly concise outputs, with an average of 4.48 paths, compared with

Table 20: Comparison of one-stage and two-stage training strategy across **SArena-Icon**, **SArena-Chemistry**, **SArena-Illustration**, and **SArena-Animation**.

(a) **Icon & Chemistry**: Editing, Text-to-SVG and Image-to-SVG.

| Model | Icon: Editing | | | | Icon: Text-to-SVG | | | | Icon: Image-to-SVG | | | | Chemistry: Text-to-SVG | | | Chemistry: Image-to-SVG | | | |
|---|---|---|---|---|---|---|---|---|---|---|---|---|---|---|---|---|---|---|---|
| | DINO↑ | SSIM↑ | LPIPS↓ | PSNR↑ | FID↓ | FID-C↓ | CLIP-T2I↑ | CLIP-I2I↑ | DINO↑ | SSIM↑ | LPIPS↓ | PSNR↑ | FID↓ | FID-C↓ | CLIP-I2I↑ | DINO↑ | SSIM↑ | LPIPS↓ | PSNR↑ |
| One-stage | 0.972 | 0.904 | 0.069 | 65.481 | 10.854 | 2.245 | **24.206** | **81.615** | **0.950** | 0.810 | **0.124** | **18.284** | 14.605 | 1.199 | 92.181 | 0.993 | 0.871 | 0.139 | 17.688 |
| Two-stage | **0.989** | **0.952** | **0.036** | **77.331** | **8.715** | **1.876** | 23.916 | 80.911 | 0.949 | **0.811** | 0.127 | 18.226 | **9.974** | **0.877** | **93.931** | **0.994** | **0.873** | **0.138** | **17.722** |

(b) **Illustration & Animation**: Illustration (Text-to-SVG / Image-to-SVG) and Animation (Text-to-SANI / Video-to-SANI).

| Model | Illustration: Text-to-SVG | | | | Illustration: Image-to-SVG | | | | Animation: Text-to-SANI | | | Animation: Video-to-SANI | | | |
|---|---|---|---|---|---|---|---|---|---|---|---|---|---|---|---|
| | FID↓ | FID-C↓ | CLIP-T2I↑ | CLIP-I2I↑ | DINO↑ | SSIM↑ | LPIPS↓ | PSNR↑ | FVD↓ | CLIP-T2V↑ | CLIP-V2V↑ | DINO↑ | SSIM↑ | LPIPS↓ | PSNR↑ |
| One-stage | 68.644 | 25.671 | 19.217 | 66.400 | 0.830 | 0.503 | 0.278 | 10.345 | 101.433 | 22.291 | 69.993 | 0.867 | 0.735 | 0.245 | 13.420 |
| Two-stage | **22.397** | **5.141** | **21.116** | **74.662** | **0.924** | **0.716** | **0.188** | **14.644** | **99.474** | **22.572** | **73.162** | **0.876** | **0.754** | **0.237** | **14.168** |

256.00 for SVGDreamer and 64.00 for VectorFusion, whose fragmented paths are often difficult to modify. The cleaner, semantically coherent structures generated by InternSVG align more closely with practical design workflows. In addition, optimization-based methods require tens to hundreds of seconds per sample (*e.g.*, 289 s for SVGDreamer and 71 s for VectorFusion), whereas InternSVG generates results in only 6.9 seconds on average, benefiting from its explicit structural encoding. This yields substantially faster inference without compromising structural quality.

## B.7 Two-Stage vs. Single-Stage Training

In this section, we provide a comprehensive analysis of the one-stage and two-stage training strategies. As shown in Table 20, for Icon generation, the one-stage model performs relatively well because Icon samples dominate the training data in the single-stage setting. Nevertheless, the two-stage model achieves comparable results and remains superior on FID, FID-C, and SSIM. For Icon editing as well as Chemistry, Illustration, and Animation generation, the two-stage model consistently outperforms the one-stage baseline. The experimental results further validate that the proposed two-stage training strategy is effective in alleviating data imbalance, and the progressive training scheme from easier to more challenging tasks leads to substantial improvements in SVG modeling performance.

## B.8 Cross-Domain Generalization Analysis

To rigorously quantify the cross-domain generalization strength of InternSVG, we conducted a zero-shot cross-domain transfer experiment. Specifically, we trained a variant of InternSVG 8B only on the Icon Image-to-SVG data used in stage 2, and then directly evaluated it on the SArena-Chemistry Image-to-SVG task, without using any Chemistry data during training. The results show that this icon-only variant still achieves substantial improvements over the original InternVL3-8B across all metrics. As shown in Table 21, the DINO Score increased from 0.865 to 0.930 (7.5%↑), SSIM increased from 0.783 to 0.824 (5.2%↑), LPIPS decreased from 0.203 to 0.156 (23.2%↓), and PSNR increased from 13.84 to 16.296 (17.8%↑). The comprehensive improvements across all metrics strongly validate that our method and data possess exceptional generalization capabilities in cross-domain transfer scenarios.

Table 21: Zero-shot cross-domain transfer performance on SArena-Chemistry.

| Model | DINO↑ | SSIM↑ | LPIPS↓ | PSNR↑ | SR |
|---|---|---|---|---|---|
| InternVL3-8B | 0.865 | 0.783 | 0.203 | 13.840 | 96.27% |
| InternSVG 8B w/ icon | **0.930** | **0.824** | **0.156** | **16.296** | **98.61%** |

## B.9 Additional Qualitative Visualization

In this section, we present additional qualitative visualizations of InternSVG on SVG understanding, editing, and generation tasks. We further compare the generated SVGs with those produced by baseline methods to assess visual quality.

As illustrated in Figure 5 and 7, for the Text-to-SVG generation tasks on icons and illustrations, VectorFusion and SVGDreamer produce visually appealing results but exhibit poor instruction-following ability and often include a large number of irrelevant elements, limiting their practical applicability. Existing general-purpose MLLMs typically suffer from structural instability and incomplete semantic representation in SVG generation, making it difficult to balance visual quality and semantic alignment. In contrast, LLM4SVG and OmniSVG frequently fail to generate valid outputs or produce results that are entirely inconsistent with the textual description, highlighting the limited generalization ability of current LLM-based approaches for SVG generation. By comparison, InternSVG consistently generates high-quality and semantically aligned SVGs across diverse scenarios, demonstrating the advantages of its unified modeling paradigm for understanding, editing, and generation. Figure 6 and 8 compare the performance of InternSVG with other approaches on the Image-to-SVG task. The results demonstrate that InternSVG not only substantially outperforms general-purpose MLLMs and existing LLM-based SVG generation models, but also exceeds LIVE in performance while requiring significantly fewer tokens. In terms of detail preservation, InternSVG demonstrates superior capability compared with existing approaches.

As shown in Figure 9 and 10, for the generation of chemical structural formulas, existing general-purpose MLLMs generally lack relevant training data and therefore fail to produce correct chemical structures in either task. In addition, compared with StarVector, our method demonstrates superior control over fine-grained details of chemical bonds.

Figure 11 and 12 present a comparison between InternSVG and general-purpose MLLMs on SArena-Animation. Existing approaches show limited ability in handling vector animations, often struggling with visual quality and motion control. In contrast, InternSVG demonstrates superior capability, producing animations with higher fidelity and more consistent temporal dynamics.

Figure 13, 14, and 15 present visualizations of InternSVG outputs on SVG understanding and editing tasks.

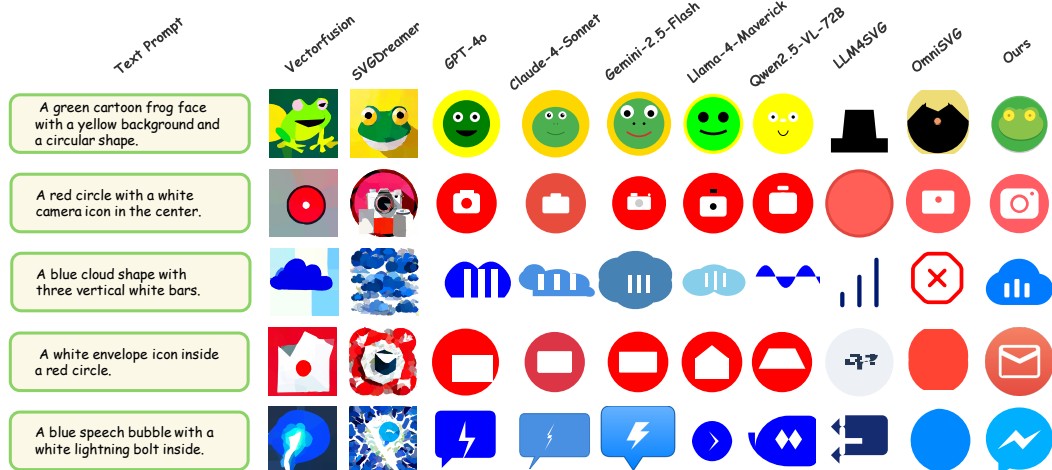

Figure 5: **Qualitative comparison of Text-to-SVG performance between baselines and InternSVG on SArena-Icon.** Red cross icons denote cases where the model failed to generate a valid SVG output.

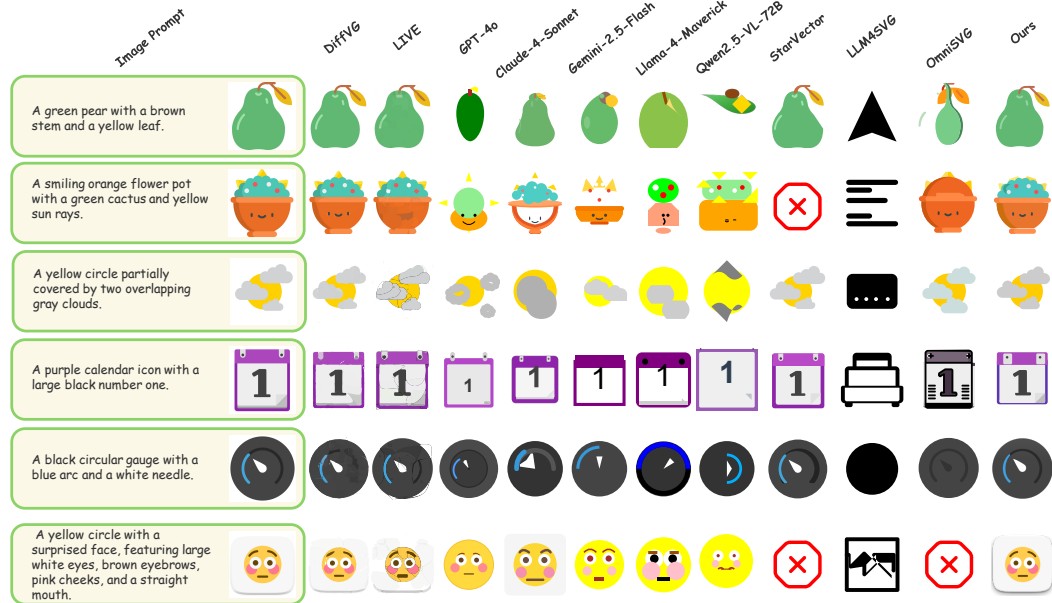

Figure 6: **Qualitative comparison of Image-to-SVG performance between SOTA methods and InternSVG on SArena-Icon.** Red cross icons denote cases where the model failed to generate a valid SVG output.

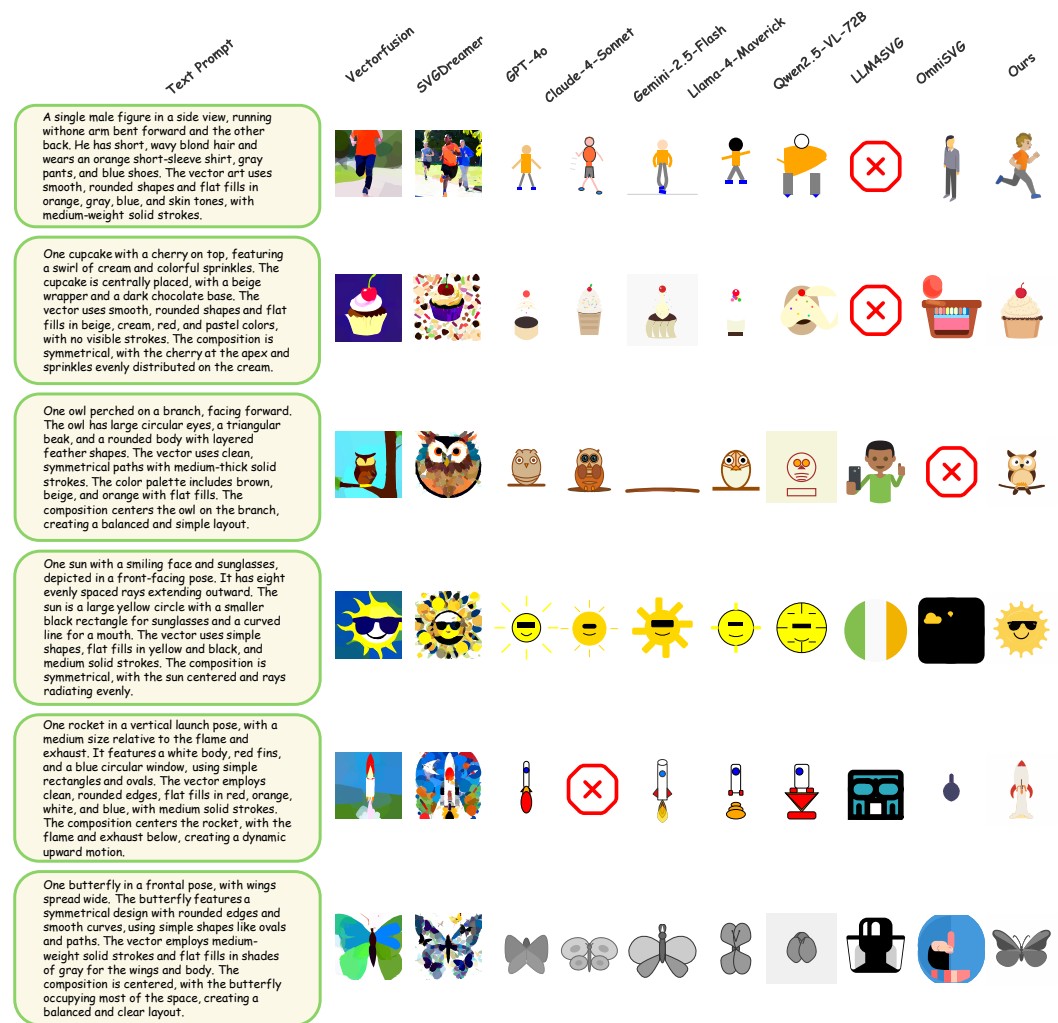

Figure 7: **Qualitative comparison of Text-to-SVG performance between SOTA methods and InternSVG on SArena-Illustration.** Red cross icons denote cases where the model failed to generate a valid SVG output.

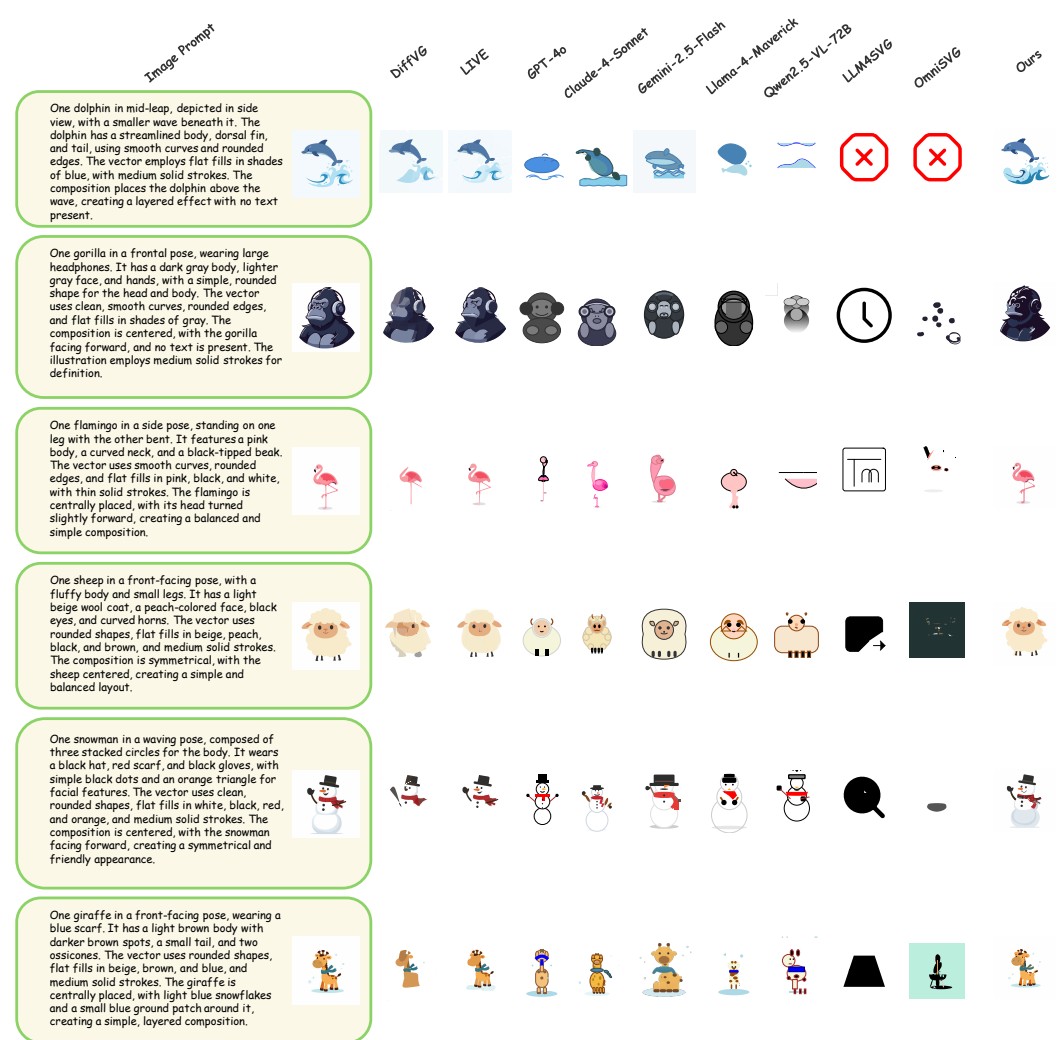

Figure 8: **Qualitative comparison of Image-to-SVG performance between baselines and InternSVG on SArena-Illustration.** Red cross icons denote cases where the model failed to generate a valid SVG output.

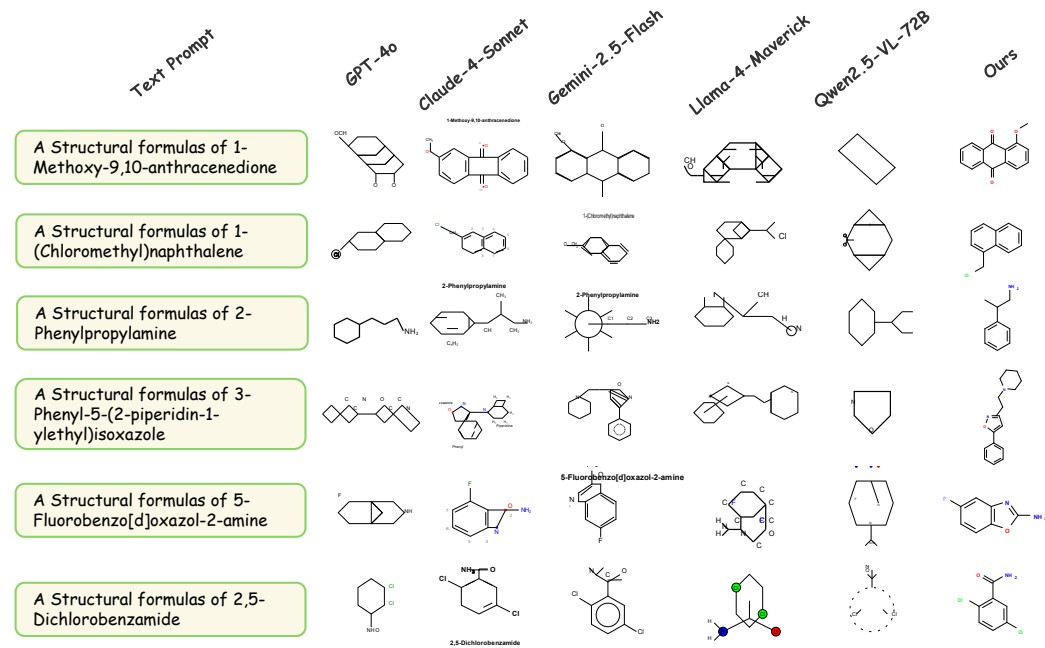

Figure 9: **Qualitative comparison of Text-to-SVG performance between baselines and InternSVG on SArena-Chemistry.**

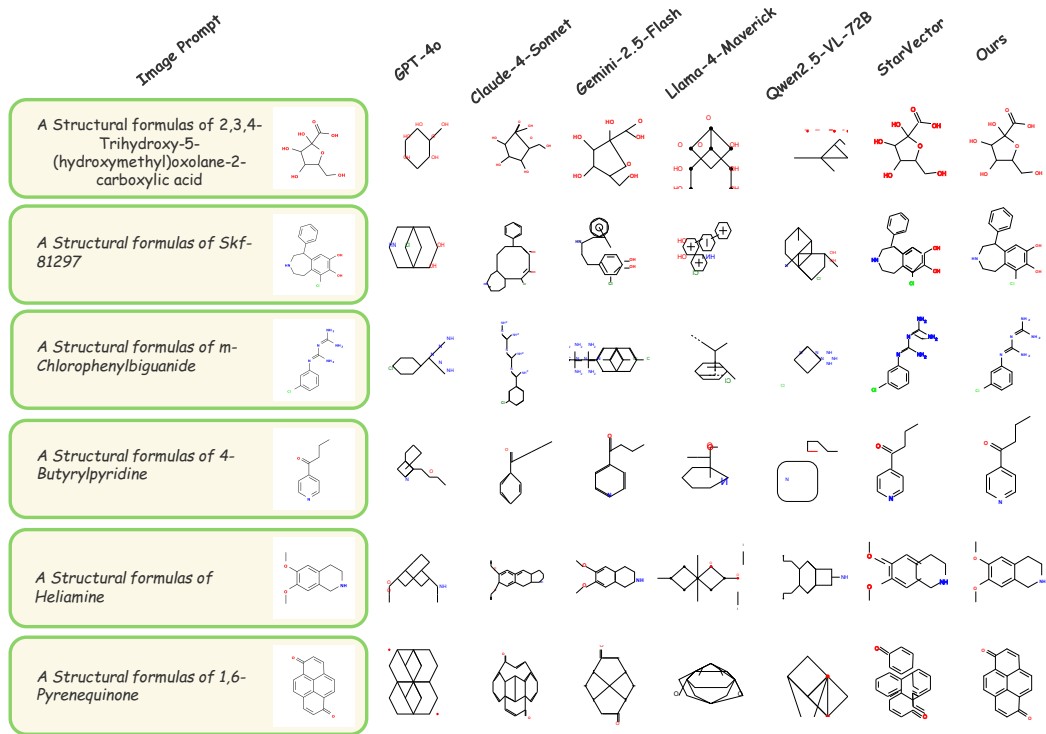

Figure 10: **Qualitative comparison of Image-to-SVG performance between baselines and InternSVG on SArena-Chemistry.**

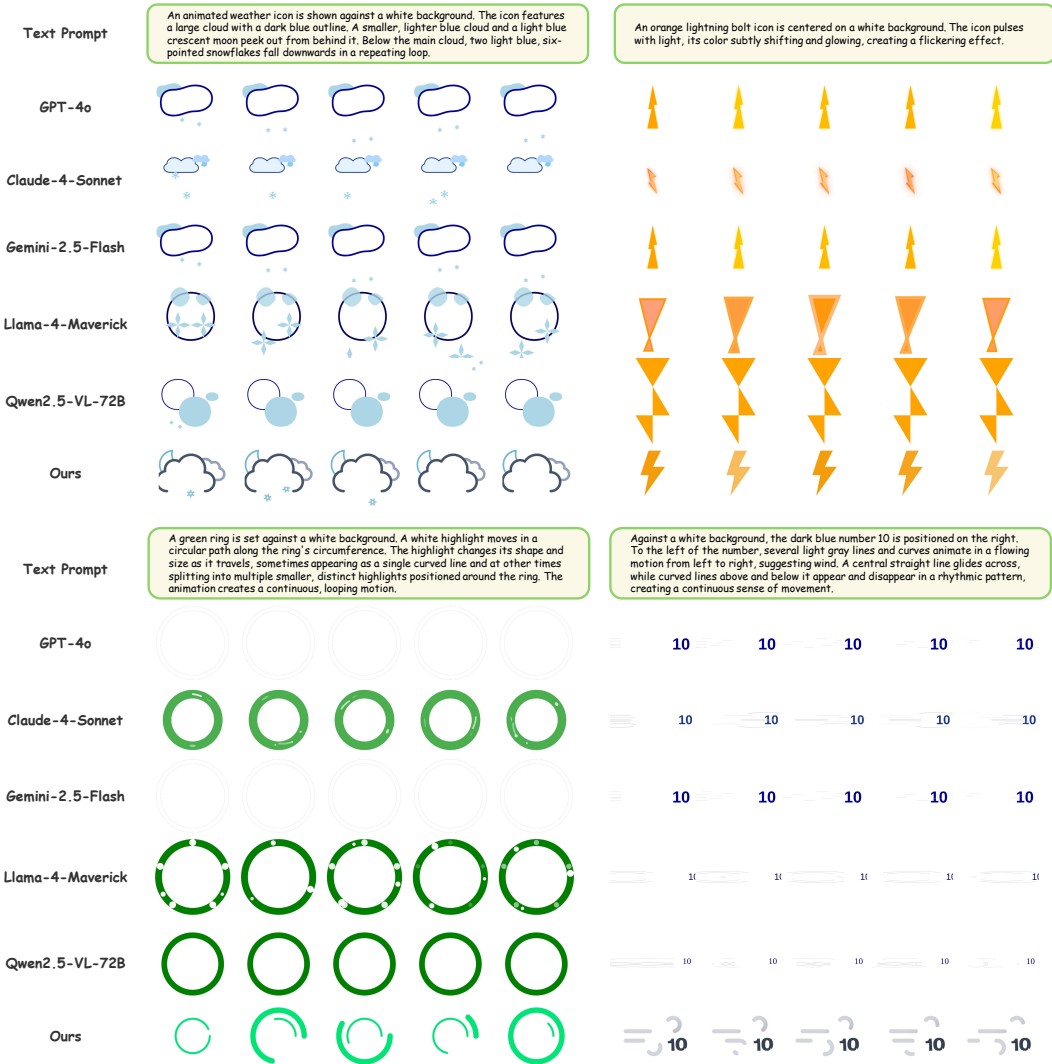

Figure 11: **Qualitative comparison of Text-to-SANI performance between baselines and InternSVG on SArena-Animation.**

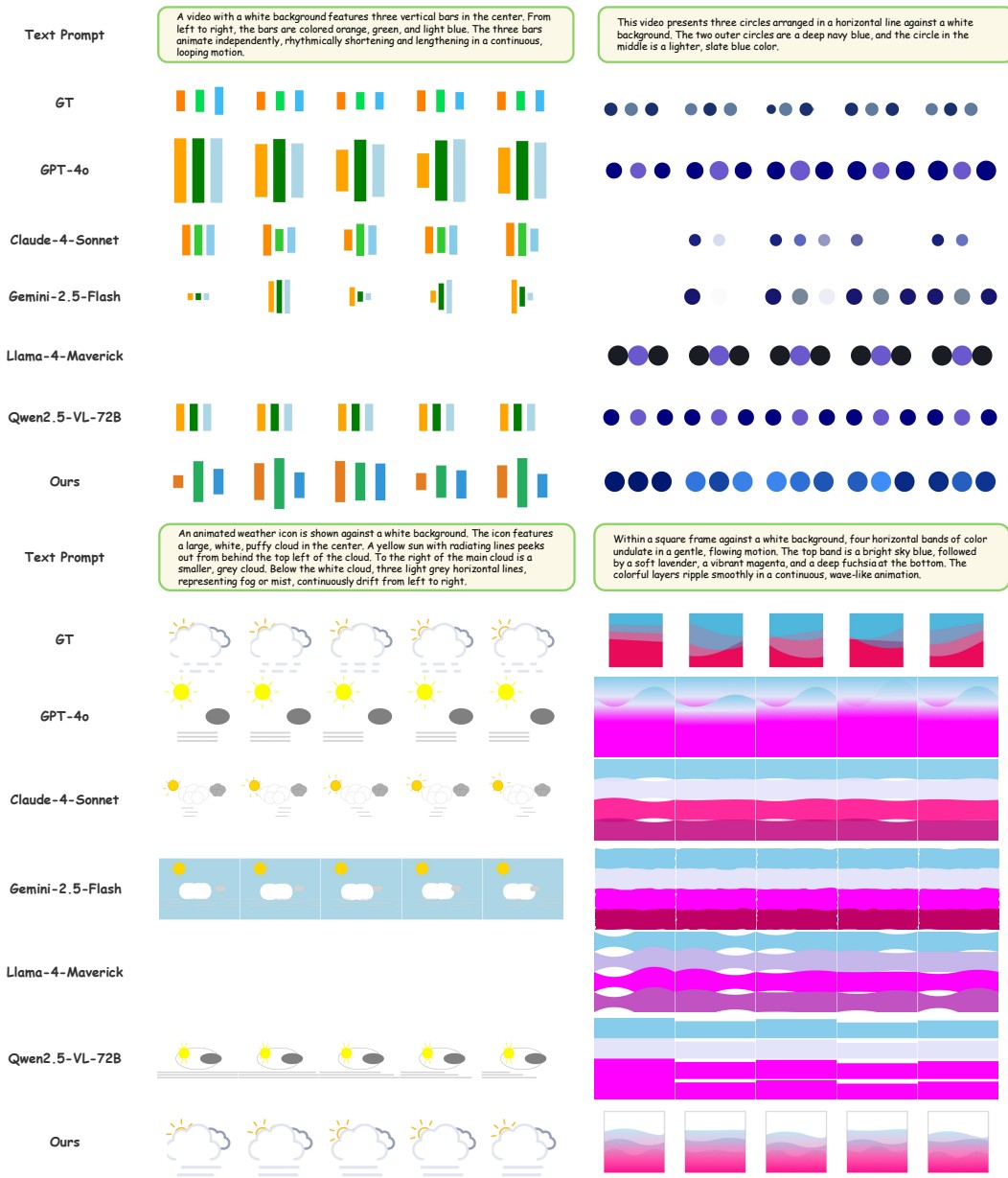

Figure 12: **Qualitative comparison of Video-to-SANI performance between baselines and InternSVG on SArena-Animation.**

**Question**

The SVG source is given below:
<svg xmlns="http://www.w3.org/2000/svg" width="128" height="128" viewBox="0 0 128 128"><path fill="#FFBDBC" d="M82.77 65.73c-7.36 5.62-13.95.03-11.16-8.31.29-.88.49-1.8.48-2.73 0-7.46-8.54-13.33-8.54-13.33s.39 9.37-5.45 12.6c-5.84 3.24-4.33 11.53-2.33 14.78 3.71 7.94-7 14.22-12.82 1.77l-3.59 8.97a22.6 22.6 0 0 0-.11 16.5 22.58 22.58 0 0 0 21.07 14.46h6.3c8.28 0 15.9-4.53 19.85-11.81l1.06-1.96c3.31-6.11 3.62-13.41.84-19.79z"/> </svg>
Identify the main object depicted, and list its geometric shapes (e.g., rectangles, circles, lines) as well as semantic cues (e.g., what the object is, its context, implied function).

**Response**

The code depicts a sleeveless shirt with the following properties:

**Geometric Properties:**
- **Shape**: The shirt has a simple, elongated shape with a rounded neckline and armholes.
- **Symmetry**: It is vertically symmetrical, with both sides mirroring each other.
- **Lines**: There are two horizontal lines across the lower part of the shirt, creating a banded appearance.

**Semantic Properties:**
- **Color**: The shirt is light blue with a darker blue outline.
- **Style**: It resembles a tank top or sleeveless athletic shirt, suggesting casual or sportswear.
- **Design**: The horizontal lines may indicate a design feature or a seam.

Overall, the shirt has a straightforward and functional design, typical of casual or athletic wear.

**Question**

The following is an SVG illustration:
<svg xmlns="http://www.w3.org/2000/svg" width="128" height="128" viewBox="0 0 128 128"><path d="M45.56 16c0-6.57 5.27-12 11.89-12 5.51 0 10.08 3.76 11.46 8.81 1.34-.52 2.79-.81 4.32-.81 6.62 0 11.89 5.43 11.89 12v41.72c.95-1.04 1.94-2.09 2.97-3.1 3.68-3.61 8.7-7.62 14.65-7.62 6.64 0 11.62 5.68 11.24 12.02-.09 1.47-.58 2.7-.94 3.5l-13.36 29.67c-.5 1.11-.87 2.26-1.31 3.65-.1.3-.2.6-.3.92C94.39 116.1 83.87 124 71.79 124H41.67C26.34 124 14 111.41 14 96V44c0-6.57 5.27-12 11.89-12 1.37 0 2.67.23 3.89.66V30c0-6.57 5.27-12 11.89-12 1.36 0 2.67.23 3.89.66zm8 0c0-2.26 1.79-4 3.89-4 2.09 0 3.89 1.74 3.89 4v55c0 2.21 1.79 4 4 4s4-1.79 4-4V24c0-2.26 1.79-4 3.89-4 2.09 0 3.89 1.74 3.89 4v52c0 1.6.95 3.05 2.42 3.68s3.18.32 4.34-.79c1.5-1.43 3.1-3.23 4.64-4.97l.17-.2c1.65-1.86 3.29-3.71 5-5.38 3.61-3.54 6.58-5.34 9.05-5.34 1.84 0 3.37 1.6 3.25 3.54 0 .01 0 .06-.03.15-3.79 12.44-11.48 23.23-15.5 35.6C87.82 110.42 80.32 116 71.79 116H41.67C30.86 116 22 107.1 22 96V44c0-2.26 1.79-4 3.89-4s3.89 1.74 3.89 4v27c0 2.21 1.79 4 4 4s4-1.79 4-4V30c0-2.26 1.79-4 3.89-4s3.89 1.74 3.89 4v38c0 2.21 1.79 4 4 4s4-1.79 4-4z"/><path fill="#fa1228" d="M57.45 12c-2.1 0-3.89 1.74-3.89 4v52c0 2.21-1.79 4-4 4s-4-1.79-4-4V30c0-2.26-1.79-4-3.89-4s-3.89 1.74-3.89 4v41c0 2.21-1.79 4-4 4s-4-1.79-4-4V44c0-2.26-1.79-4-3.89-4S22 41.74 22 44v52c0 11.1 8.86 20 19.67 20h30.12c8.53 0 16.03-5.58 18.67-13.71 4.02-12.37 11.71-23.16 15.5-35.6.03-.09.03-.14.03-.15.12-1.94-1.41-3.54-3.25-3.54-2.47 0-5.44 1.8-9.05 5.34-1.71 1.67-3.35 3.52-5 5.38l-.17.2c-1.54 1.74-3.14 3.54-4.64 4.97a4 4 0 0 1-4.34.79A4 4 0 0 1 77.12 76V24c0-2.26-1.8-4-3.89-4-2.1 0-3.89 1.74-3.89 4v47c0 2.21-1.79 4-4 4s-4-1.79-4-4V16c0-2.26-1.8-4-3.89-4"/></svg>
Describe in detail the semantic and geometric features of the SVG code.

**Response**

The code depicts a sleeveless shirt with the following properties:

**Geometric Properties:**
- **Shape**: The shirt has a simple, elongated shape with a rounded neckline and armholes.
- **Symmetry**: It is vertically symmetrical, with both sides mirroring each other.
- **Lines**: There are two horizontal lines across the lower part of the shirt, creating a banded appearance.

**Semantic Properties:**
- **Color**: The shirt is light blue with a darker blue outline.
- **Style**: It resembles a tank top or sleeveless athletic shirt, suggesting casual or sportswear.
- **Design**: The horizontal lines may indicate a design feature or a seam.

Overall, the shirt has a straightforward and functional design, typical of casual or athletic wear.

Figure 13: **Visualization of InternSVG outputs on SVG understanding.**

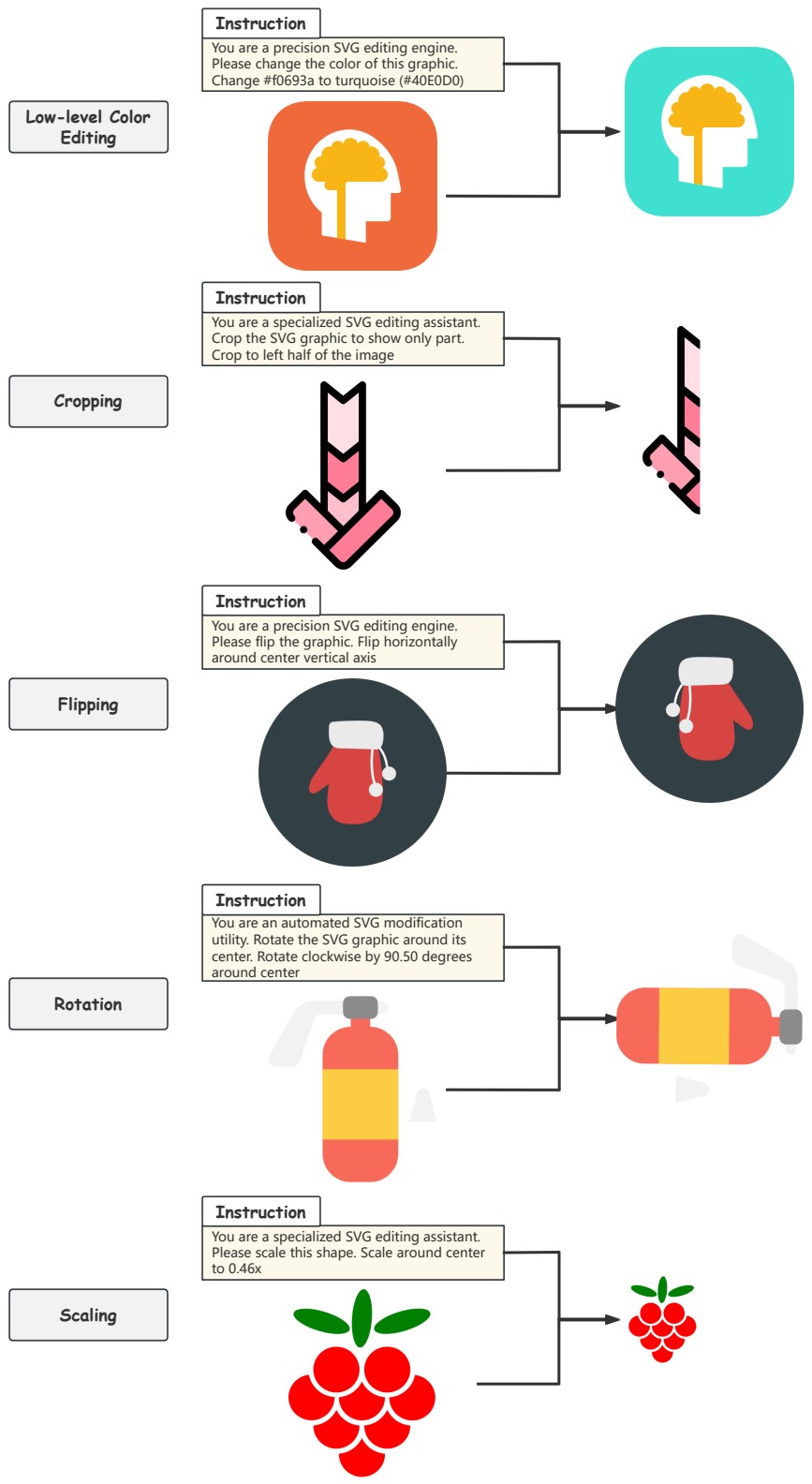

Figure 14: **Visualization of InternSVG outputs on SVG editing.**

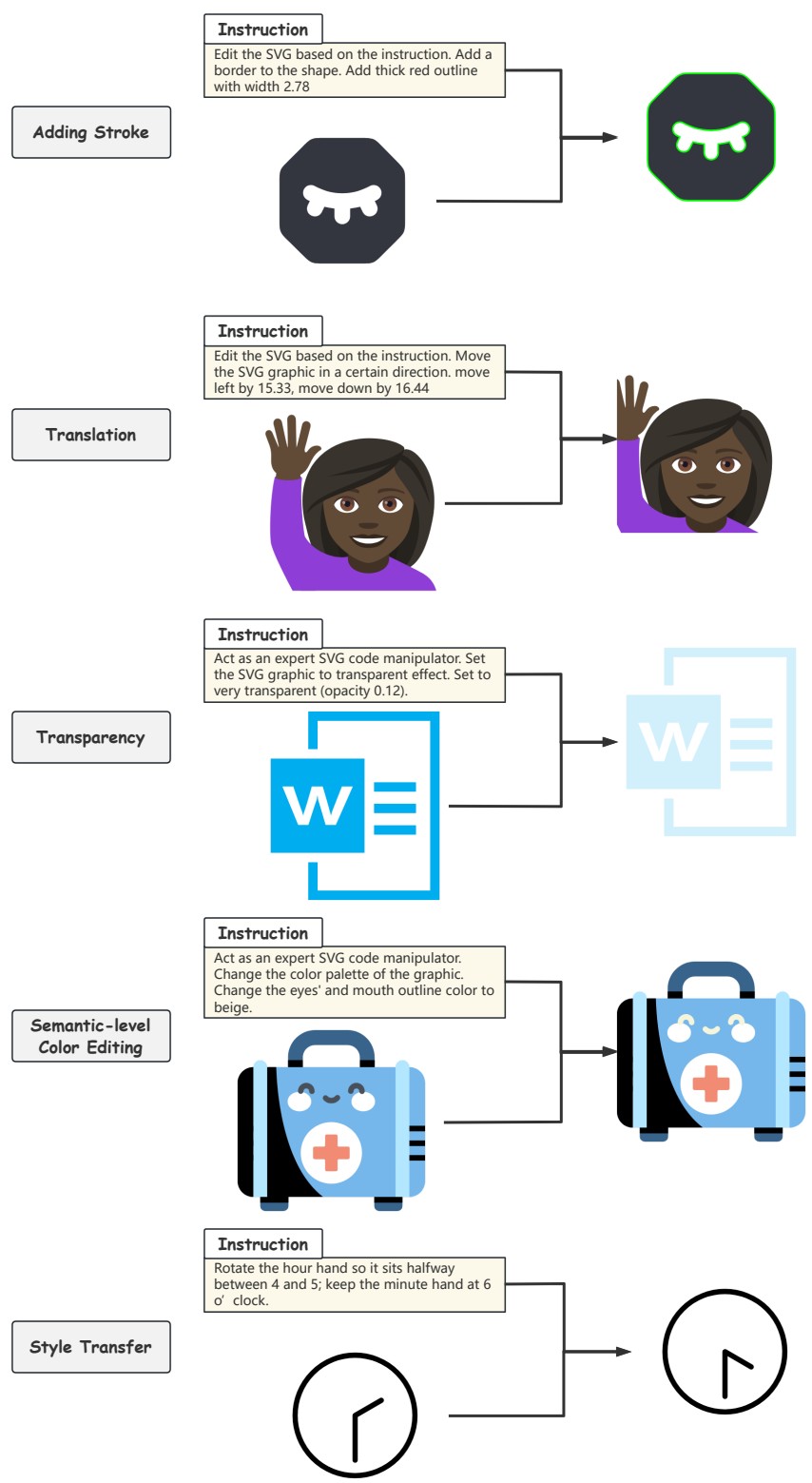

Figure 15: **Visualization of InternSVG outputs on SVG editing.**

## B.10 USER STUDY AND EVALUATION DETAILS

We conducted a user study to comprehensively evaluate the quality of SAgoge's synthetic annotations and the practical efficacy of InternSVG's generation capabilities. A total of 42 valid responses were collected from participants with academic or professional backgrounds relevant to our study, including 14.29% researchers, 57.14% PhD students, and 28.57% UI designers. The cohort primarily came from computer-related fields and also included professional UI designers. These qualitative evaluations complement the quantitative metrics presented in the main experiments, providing further validation of data reliability and model performance.

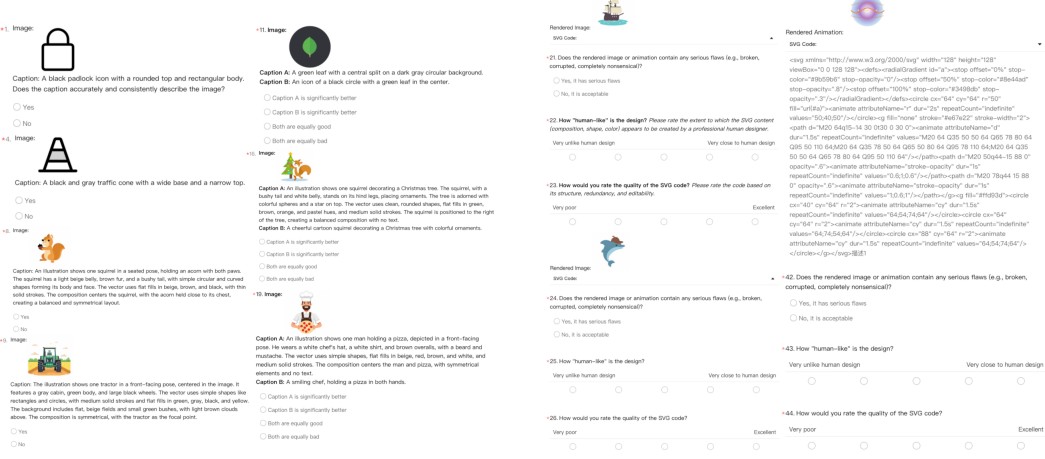

(a) Consistency check and Pairwise Preference Test    (b) Synthetic Data Quality Evaluation

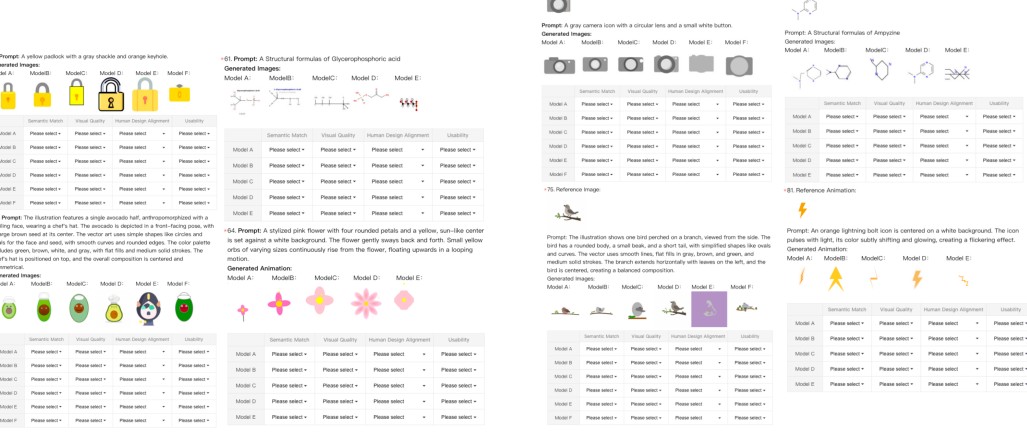

(c) Text-to-SVG / Text-to-SANI Comparison    (d) Image-to-SVG / Video-to-SANI Comparison

Figure 16: **Screenshots of user study.** (a) Participants judge whether a caption matches the image, and compare two captions to decide which one better describes the content. (b) Participants rate the quality and human-likeness of synthetic SVGs. (c) & (d) Participants compare the outputs of six different models for Text-to-SVG / Text-to-SANI and Image-to-SVG / Video-to-SANI tasks.

First, we assessed the quality of our synthetic annotations to ensure they serve as reliable supervision for model training. Specifically, we sampled 100 model-generated instructions from our dataset, comprising 40 icons, 40 illustrations, and 20 animations. We designed two types of evaluation tasks for these samples (see Figure 16a): a consistency check and a pairwise preference test.

In the consistency check, participants judged whether the model-generated instruction is semantically consistent with the rendered image or video. In the pairwise preference test, participants were shown a human-authored instruction and a model-generated instruction for the same image (with

sources blinded) and selected among "A is better", "B is better", "Both are good", or "Both are bad". As summarized in Table 22, the results show significantly high semantic consistency (95.04% overall). In the comparative assessment, participants prefer model-generated instructions in 45.43% of cases, while human-authored instructions are preferred in only 15.67% of cases, and 36.90% of samples are considered equally good. Overall, more than 82% of the samples indicate that model-generated annotations are comparable to human annotations in quality.

To more directly assess whether there is a noticeable gap in quality or "human-likeness" compared to real human-authored SVGs, we additionally conducted a user study on the synthesized illustration and animation data (see Figure 16b). For each participant, we randomly sampled five illustration SVGs and five animation SVGs, presenting both the SVG code and the rendered images. Participants evaluated: (i) whether the image contains any visible defects; (ii) the likelihood that the content is created by a human designer (Human-Likeness Score); and (iii) the quality and readability of the SVG code (Code Quality Score). Both scoring metrics (ii) and (iii) were evaluated on a 1–5 scale, with higher scores indicating better quality. The results, shown in Table 23, indicate that for illustrations, 96.19% of samples are judged to contain no defects, with a Human-Likeness Score of 4.20 and a Code Quality Score of 4.31. For animations, 93.81% of samples are considered defect-free, with a Human-Likeness Score of 4.07 and a Code Quality Score of 4.23. These findings demonstrate that synthesized data is generally of high quality and closely aligned with human design conventions, validating the reliability of our synthetic pipeline.

Table 22: Human evaluation results on synthetic data annotation quality (N=42, Total Samples=504).

| Category | Total Samples | Semantic Consistency | | Caption Preference (Model vs. Human) | | | |
|---|---|---|---|---|---|---|---|
| | | Yes | No | Model Better | Human Better | Both Good | Both Bad |
| Icon | 210 | 97.62% | 2.38% | 43.81% | 18.10% | 36.19% | 1.90% |
| Illustration | 210 | 92.38% | 7.62% | 46.67% | 15.24% | 35.71% | 2.38% |
| Animation | 84 | 95.24% | 4.76% | 46.43% | 10.71% | 41.67% | 1.19% |
| **Overall** | **504** | **95.04%** | **4.96%** | **45.43%** | **15.67%** | **36.90%** | **1.98%** |

Table 23: Human evaluation of synthetic SVG quality.

| Category | Defect-Free | Defective | Human-Likeness Score | Code Quality Score |
|---|---|---|---|---|
| Illustration | 96.19% | 3.81% | 4.20 | 4.31 |
| Animation | 93.81% | 6.19% | 4.07 | 4.23 |

Second, to verify whether the perceptual and semantic quality of SVGs generated by our model aligns with human preferences, we conducted a systematic human evaluation. For each participant, we randomly sampled 5 icons, 5 illustrations, 3 chemical structures, and 3 animations from the Text-to-SVG / Text-to-SANI and Image-to-SVG / Video-to-SANI evaluation results for scoring.

In each task group (see Figure 16c and Figure 16d), we presented the participants with the input text description or image, along with the SVG outputs generated by six models (GPT-4o, Gemini-2.5-Flash, Claude-Sonnet-4, Qwen2.5-VL-72B, OmniSVG, InternSVG) in randomized order. Participants rated each output on a scale of 1–5 across multiple dimensions, including Semantic Match / Visual Similarity, Visual Quality, Human Design Alignment, and Usability.

(1) **Semantic Match / Visual Similarity:** Whether the output accurately fulfills the input description or faithfully reconstructs the image content;

(2) **Visual Quality:** Whether the graphic structure is clear, concise, and stable, and whether there are obvious defects;

(3) **Human Design Alignment:** Whether the SVG conforms to common human design intuitions (simplicity, structural clarity, compositional balance, color harmony, etc.);

(4) **Usability:** Whether the generated SVG can be directly used in practical design and editing workflows.

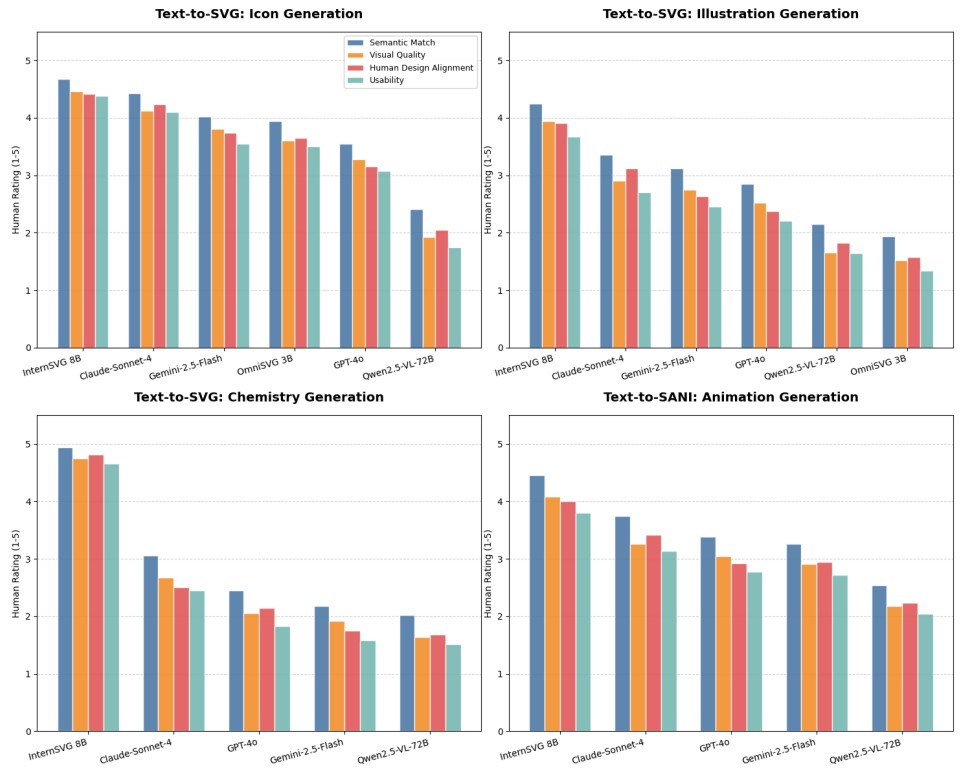

(a) Human evaluation results for **Text-to-SVG / Text-to-SANI**.

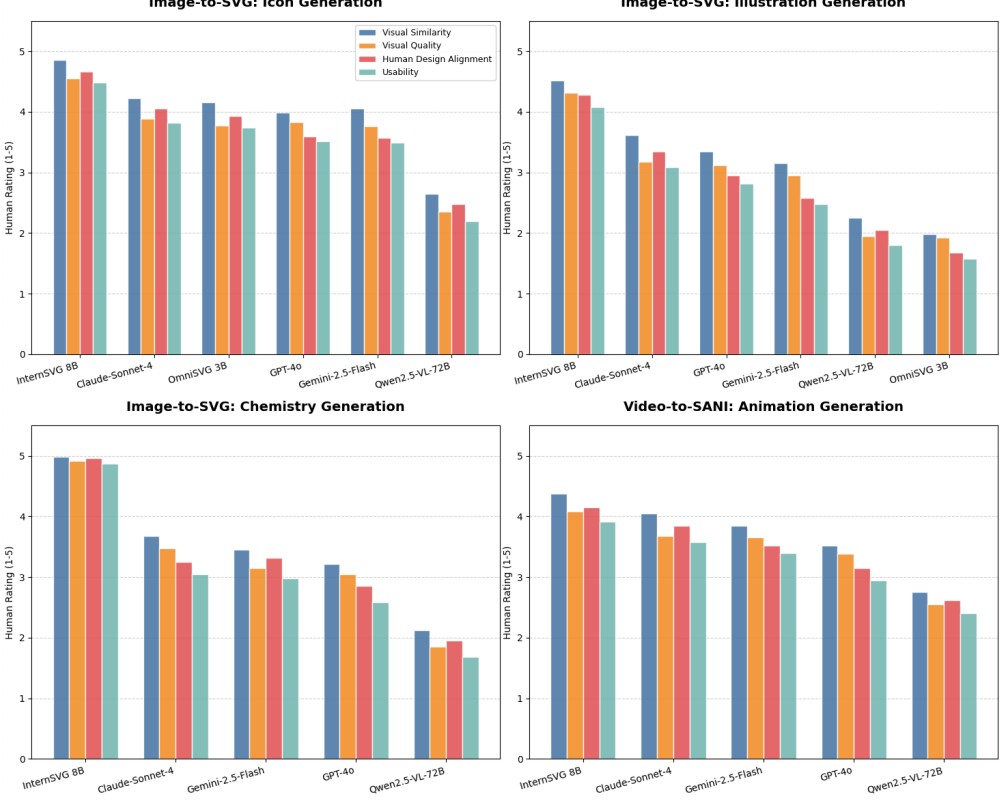

(b) Human evaluation results for **Image-to-SVG / Video-to-SANI**.

Figure 17: Human evaluation results for synthetic SVG generation tasks.

The user study results, visualized in Figure 17a and Figure 17b, demonstrate that InternSVG significantly outperforms other baseline models in both adherence to user input instructions and image reconstruction fidelity. The generated SVGs exhibit higher quality, clearer and more stable graphic structures, fewer defects, and better alignment with human design preferences and usability. Moreover, our quantitative evaluation metrics are consistent with the user study results. For example, in Image-to-SVG for icons, the user evaluation results show InternSVG (4.64), Claude-Sonnet-4 (4.00), OmniSVG-3B (3.90), Gemini-2.5-Flash (3.73), GPT-4o (3.72), and Qwen2.5-VL-72B (2.42), presenting a trend that is largely consistent with our quantitative metric results.

## C  DETAILS OF SAGOGE AND SARENA

### C.1  STATISTICS OF SAGOGE

Table 24 presents the statistics of SAgoge in its four domains. The Icon subset contains 2.8M SVGs and 11M samples, with an average sequence length of 846 tokens, which reflects relatively simple structures. The Illustration subset is smaller in scale, with 600k SVGs and 1.6M samples, but it has much longer sequences with an average of 8.7k tokens that indicate high structural complexity. The Animation subset consists of 61K SVGs and 122K samples, and it provides dynamic elements that capture temporal changes. The Chemistry subset provides 1.7M SVGs and 3.4M samples, and its average length of 1.7k tokens shows the rich geometric and semantic details of chemical diagrams.

Table 24: **Data statistics of SAgoge.** #SVGs denotes the number of vector graphic files, #Samples refers to the number of training samples, and Avg. Tokens indicates the average token length of the SVG code.

| Dataset | #SVGs | #Samples | Avg. Tokens |
|---|---|---|---|
| Icon | 2.8M | 11M | 846 |
| Illustration | 600K | 1.6M | 8673 |
| Animation | 61K | 122K | 847 |
| Chemistry | 1.7M | 3.4M | 1752 |

### C.2  STATISTICS OF SARENA

Table 25 shows the data statistics of SArena. SArena encompasses multiple domains (Icon, Illustration, Animation, Chemistry) and a diverse set of tasks, including multiple-choice QA, editing, text-to-SVG, image-to-SVG, and video-to-SVG. Here, Avg. Tokens is the average token length of the training sample.

Table 25: **Data statistics of SArena.**

| Benchmark | Tasks | #Samples | Avg. Tokens |
|---|---|---|---|
| Icon | Mutiple-choice QA | 6012 | 1197 |
| | Editing | 2000 | 3175 |
| | Text-to-SVG | 6013 | 1118 |
| | Image-to-SVG | 6013 | 1169 |
| Illustration | Text-to-SVG | 2001 | 8181 |
| | Image-to-SVG | 2001 | 8291 |
| Animation | Text-to-SANI | 504 | 1677 |
| | Video-to-SANI | 504 | 1719 |
| Chemistry | Text-to-SVG | 3003 | 1010 |
| | Image-to-SVG | 3003 | 1065 |

### C.3  DATA SYNTHESIS PIPELINE

#### C.3.1  ILLUSTRATION DATA SYNTHESIS PIPELINE

As high-quality open-source SVG resources for illustrations are scarce, we establish a dedicated synthesis pipeline to expand our dataset. The overall process is depicted in Figure 18.

We begin by employing GPT-4o to automatically generate diverse textual prompts covering a wide range of objects, styles, and scene descriptions. These prompts are crafted to encourage the generation of vector-like visual features, ensuring compatibility with subsequent vectorization. The

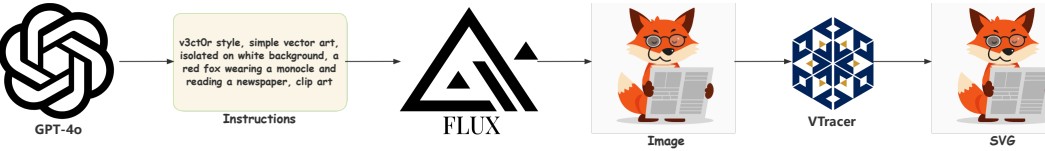

Figure 18: **Illustration data synthesis pipeline.** Textual prompts generated by GPT-4o are processed by a FLUX-based model and converted to SVGs via VTracer.

prompts are then processed by a FLUX-based generative model equipped with a vector-style LoRA adapter[1]. This step ensures that the rendered images are not only semantically consistent with the prompts but also stylistically aligned with the characteristics of SVG, such as simplified shapes, flat colors, and clean edges. Finally, we apply VTracer to convert the generated images into SVG format, which performs vectorization by detecting paths, curves, and fills. This automated workflow bridges the gap between raster generation and vector representation, enabling us to systematically produce a large-scale, diverse, and high-quality dataset of SVG illustrations.

### C.3.2 ANIMATION DATA SYNTHESIS PIPELINE

High-quality open-source SVG animations are extremely scarce, making it infeasible to rely solely on existing resources to achieve the desired scale and diversity. To overcome this limitation, we exploit the powerful code-generation capability of Claude-Sonnet-4 to synthesize SVG animations compliant with the SMIL standard. We design generation protocols with explicit constraints, such as fixed canvas size and mandatory animation primitives, which ensure structural consistency, renderability, and suitability for large-scale training. This pipeline enables systematic expansion of the animation dataset and produces SVG animations with richer semantic content and greater geometric diversity.

---

[1]https://huggingface.co/renderartist/simplevectorflux

### C.3.3 EXAMPLES OF SAGOGE

```
<svg xmlns="http://www.w3.org/2000/svg"
width="128" height="128" viewBox="0 0
128 128"><path fill="#0b80e0"
d="M117.33 53.33V42.67c0-5.87-4.8-10.67-
10.66-10.67H16c-5.87 0-10.67 4.8-10.67
10.67v42.66C5.33 91.2 10.13 96 16
96h90.67c5.86 0 10.66-4.8 10.66-
10.67V74.67c2.94 0 5.34-2.4 5.34-
5.34V58.67c0-2.94-2.4-5.34-5.34-5.34m-
5.33 32c0 2.94-2.4 5.34-5.33 5.34H16c-2.93
0-5.33-2.4-5.33-5.34V42.67c0-2.94 2.4-5.34
5.33-5.34h90.67c2.93 0 5.33 2.4 5.33
5.34zm-96 0h48V42.67H16z"/></svg>
```

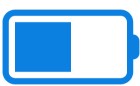

**Question: What is the shape of the main part of the object?**
A) Circle
B) Rectangle
C) Triangle
D) Square

**Category: Geometry**

```
<svg xmlns="http://www.w3.org/2000/svg"
width="128" height="128" viewBox="0 0
128 128"><path fill="#d5e0e1" d="M.5
64a63.5 63.5 0 1 0 127 0 63.5 63.5 0 1 0-127
0"/><path fill="#2c424f" d="M27.2
52.6a25.1 25.1 0 1 0 50.2 0 25.1 25.1 0 1 0-
50.2 0"/><path fill="#f7f8f9" d="M32.5
52.6a19.8 19.8 0 1 0 39.6 0 19.8 19.8 0 1 0-
39.6 0"/><path fill="#2c424f" d="m69.17
71.26 2.4-2.41 4.88 4.88-2.4 2.41z"/><path
fill="#bf4028" d="m99.85 97.25-2.3 2.3c-
1.35 1.35-3.5 1.25-4.75-.15L74.25 78.55c-
1.15-1.3-1.1-3.25.15-4.5s3.2-1.3
4.5-.15l20.85 18.55c1.35 1.25 1.4 3.45.1
4.8"/><path fill="#f7f8f9" d="M41.35
56.95a6.75 6.75 0 1 0 13.5 0 6.75 6.75 0 1 0-
13.5 0M47.3 41.1a2.95 2.95 0 1 0 5.9 0 2.95
2.95 0 1 0-5.9 0"/></svg>
```

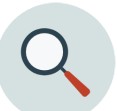

**Question: What color is the handle of the magnifying glass?**
A) Blue
B) Green
C) Yellow
D) Red

**Category: Color**

```
<svg xmlns="http://www.w3.org/2000/svg"
width="128" height="128" viewBox="0 0
128 128"><path d="M64 7.2c-6.4 0-12.8.8-
19.2 3.2H44l9.6 44.8c-2.4 2.4-3.2 5.6-3.2 8.8
0 7.2 5.6 13.6 13.6 13.6q3.6 0 7.2-2.4l17.6
7.2L95.2 68l-17.6-7.2c-.8-4-4-7.2-8-9.6l-7.2-
28.8h.8c23.2 0 41.6 19.2 41.6 41.6 0 23.2-
19.2 41.6-41.6 41.6-23.2 0-41.6-19.2-41.6-
41.6 0-12.8 5.6-24 15.2-32l-6.4-14.8C16
27.6 6.4 44.4 6.4 63.6c0 31.6 25.6 58 57.6
58s57.6-25.6 57.6-58c0-31.2-26.4-56.4-
57.6-56.4"/></svg>
```

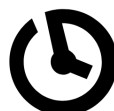

**Question: How many hands does the clock have?**
A) One
B) Two
C) Three
D) Four

**Category: Quantity**

```
<svg xmlns="http://www.w3.org/2000/svg"
width="128" height="128" class="icon"
viewBox="0 0 128 128"><path fill="#3572a5"
d="M0 64a64 64 0 1 0 128 0A64 64 0 1 0 0 64"
class="selected" /><path fill="#fff"
d="M102.86 54.21c1.96 7.85 2.05 13.64 0
19.76-1.97 5.95-4.09 9.82-9.77
9.82H63.85v2.68h19.41v7.44c0 5.64-4.78 8.51-
9.69 9.94-7.41 2.14-13.27 1.81-19.45 0-5.17-
1.52-9.68-4.64-9.68-9.94V75.3c0-5.35 4.24-
10.26 9.59-10.26h19.52c6.5 0 12.39-5.33 12.39-
12.06v-8.69h7.15c5.69 0 8.37 4.26 9.77
9.91zM74.78 91.43c-2.02 0-3.67 1.66-3.67 3.71
0 2.06 1.65 3.74 3.67 3.74 2.03 0 3.67-1.68 3.67-
3.73 0-2.06-1.64-3.72-3.67-3.72"/><path
fill="#fff" d="M54.03 63.04c-6.63 0-12.27 5.56-
12.27 12.01v8.74H35.1c-5.68 0-9-4.03-10.39-
9.82-1.87-7.77-1.79-12.32 0-19.77 1.56-6.49
6.53-9.91 12.21-9.91h26.93v-2.68H44.44v-
7.24c0-5.64 1.4-8.7 9.67-10.16a54.5 54.5 0 0 1
9.24-.8 60.7 60.7 0 0 1 10.17.8c5.3.89 9.74 4.87
9.74 10.16v18.61c0 5.46-4.28 10.06-9.71
10.06zM52.8 36.85c2.02 0 3.66-1.66 3.67-3.71
0-2.06-1.65-3.74-3.67-3.74-2.03 0-3.67 1.68-
3.67 3.74 0 2.05 1.64 3.71 3.67 3.71"/></svg>
```

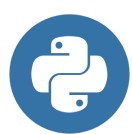

**Question: What is the object as a whole?**
A) A dragon logo
B) A medical symbol
C) The Python programming language logo
D) An abstract art piece

**Category: Semantic**

```
<svg xmlns="http://www.w3.org/2000/svg"
width="128" height="128" fill="none"
viewBox="0 0 128 128"><path
stroke="#33363F" stroke-linecap="round"
stroke-width="2" d="M88 109.33H40"/><path
fill="#33363F" d="M69.33 98.67a5.33 5.33 0 1
1-10.66 0zm-10.66 0V85.33h10.66v13.34z"/>
<path stroke="#33363F" stroke-
linecap="round" stroke-width="2" d="M56
50.67h16M29.33 77.33s-10.66-8-10.66-
21.33V45.33a10.67 10.67 0 0 1 10.66-
10.66h0A10.67 10.67 0 0 1 40
45.33v5.34m58.67 26.66s10.66-8 10.66-
21.33V45.33a10.67 10.67 0 0 0-10.66-
10.66h0A10.67 10.67 0 0 0-10.67-
10.67H50.67A10.67 10.67 0 0 0 40
40v20.59a21.33 21.33 0 0 0 9.5 17.75l11.54
7.69a5.33 5.33 0 0 0 5.92 0l11.54-7.69A21.33
21.33 0 0 0 88 60.58v.01"/></svg>
```

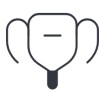

**Question: What is the object as a whole?**
A) A lamp
B) A vase
C) A trophy
D) A clock

**Category: Semantic**

```
<svg xmlns="http://www.w3.org/2000/svg"
width="128" height="128" viewBox="0 0 128
128"><path fill="#5bc2a7" fill-rule="evenodd"
d="M126.67 48.44a6.96 6.96 0 0 0-3.61-2.27
11.8 11.8 0 0 0-3.13-.37l-35.96-.03a3.07 3.07 0
0 1-1.75-.7 3.07 3.07 0 0 1-1.15-1.49L70.65
6.23a11.3 11.3 0 0 0-1.5-3.39A6.3 6.3 0 0 0
63.99 0a6.3 6.3 0 0 0-5.14 2.85 11.3 11.3 0 0 0-
1.5 3.4L46.93 43.58a3.07 3.07 0 0 1-1.14 1.49
3.07 3.07 0 0 1-1.74.71l-35.98.02c-1.39.02-
2.68.11-4.25.77a6.65 6.65 0 0 0-2.48 1.86A6 6 0
0 0 0 52.19c.04 2.13.94 3.43 1.68 4.32.78.91 1.6
1.55 2.51 2.16l28.99 19.06c.52.22 1.43 1.82 1.35
2.89a2 2 0 0 1-.09.64L23.17 118.1a11.8 11.8 0 0
0-.56 3.34c.02 1.06.14 2.43 1.21 3.97 1.03 1.58
3.32 2.57 4.89 2.48 2.96-.17 4.19-1.33 5.66-
2.44l28.35-24.07a2.4 2.4 0 0 1 1.48-.46 2.35
2.35 0 0 1 1.44.43l29.88 24.27c1.47 1.07 2.69
2.18 5.59 2.33h.2c1.49 0 3.58-.9 4.61-2.4 1.09-
1.52 1.22-2.92 1.23-3.95a11.3 11.3 0 0 0-.66-
3.58L93.83 80.95a2 2 0 0 1-.1-.66c-.07-1 .75-
2.43 1.22-2.63l28.85-18.98c.91-.61 1.74-1.26
2.52-2.17.73-.9 1.63-2.19 1.67-4.32a5.94 5.94 0
0 0-1.32-3.75" clip-rule="evenodd"/></svg>
```

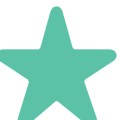

**Question: What is the primary color of the star?**
A) Red
B) Green
C) Blue
D) Yellow

**Category: Color**

```
<svg xmlns="http://www.w3.org/2000/svg"
width="128" height="128" viewBox="0 0
128 128"><g fill="none" fill-
rule="evenodd"><path fill="#07508b"
d="m104.08 71.7-9.42-9.01-29.78-27.36-
12.84 12.19c-1.17 1.6-1.09 3.75.24
5.25l11.74 10.1-.04.02 12.61 12.06c1.49 1.66
1.45 4.11-.11 5.71l-34.01 32.32 15.46 14.76
46.15-43.84c3.54-3.37 3.55-8.84 0-
12.22z"/><path fill="#e57b25" d="m2.64
56.06 9.43 8.99 29.79 27.36 12.83-12.2a4.15
4.15 0 0 0-.24-5.24L42.72 64.85l.04-.03-
12.63-12.05c-1.47-1.65-1.43-4.12.12-
5.7l34.01-32.34L48.8 0 2.65 43.85c-3.53
3.37-3.53 8.84 0 12.21z"/></g></svg>
```

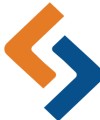

**Question: What is the primary shape of the orange part?**
A) Circle
B) Arrow
C) Triangle
D) Square

**Category: Geometry**

```
<svg xmlns="http://www.w3.org/2000/svg"
width="128" height="128" fill="none"
viewBox="0 0 24 24"><g stroke="#222">
<path d="M3 9.85C3 8.83 3.83 8 4.85 8c.71
0 1.35-.4 1.66-1.02l.82-
1.65c.11-.22.17-.33.23-.42.33-.5.86-.83
1.45-.9.11-.01.23-.01.48-.01h5.02c.25 0 .37 0
.48.01.59.07 1.12.4
1.45 9.06.09.12.2.23.42l.82 1.65c.31.62.95
1.02 1.66 1.02C20.17 8 21 8.83 21
9.85v5.01c0 2 0 3-.46
3.74-.24.38-.56.7-.94.94-.74.46-1.74.46-
3.74.46H8.14c-2 0-3 0-
3.74-.46-.38-.24-.7-.56-.94-.94C3 17.86 3
16.86 3 14.86z"/><circle cx="12" cy="13"
r="3.5"/></g></svg>
```

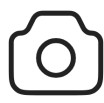

**Question: How many distinct outlines are present in the image?**
A) One
B) Two
C) Three
D) Four

**Category: Quantity**

Figure 19: **Examples of multiple-choice QA tasks.** In this task, we only provide the SVG code, requiring the model to answer the question directly based on the semantic and structural information contained in the SVG code.

**Question**

<svg xmlns="http://www.w3.org/2000/svg" width="128" height="128" viewBox="0 0 128 128"><path fill="#78b9eb" d="M69.76 14.38a43.8 43.8 0 0 0-11.52 0A46 46 0 0 0 18 60v10a14 14 0 0 1 28 0 17.3 17.3 0 0 1 .94-5.68A17.96 17.96 0 0 1 60 52.44a18.4 18.4 0 0 1 8 0 17.96 17.96 0 0 1 13.06 11.88A17.3 17.3 0 0 1 82 70a14 14 0 0 1 28 0V60a46 46 0 0 0-40.24-45.62M50.82 101.18A3.9 3.9 0 0 0 48 100a4.01 4.01 0 0 0-4 4v5.58a12.29 12.29 0 0 0 10.42 12.32A12.01 12.01 0 0 0 68 110V92h2a4.03 4.03 0 0 0-4-4 4.01 4.01 0 0 0-4-4H58a4.03 4.03 0 0 0-4 4 4.01 4.01 0 0 0 4 4h2v18a4 4 0 0 1-4.68 3.94 4.16 4.16 0 0 1-3.32-4.16V104a3.9 3.9 0 0 0-1.18-2.82"/><path fill="#1a6fb0" d="M97.93 26.07a47.97 47.97 0 0 0-26.89-13.5l-5.29-9.54a2.08 2.08 0 0 0-3.5 l-5.29 9.55A47.99 47.99 0 0 0 16 60v10a2 2 0 0 0 4 0 12 12 0 0 1 24 0 2 2 0 0 0 4 0 16.02 16.02 0 0 1 10-14.82V82a6 6 0 0 0 0 12v16a2 2 0 0 1-2.36 1.96 2.16 2.16 0 0 1-1.64-2.18V104a6.03 6.03 0 0 0-6-6 6 0 0 0-6 6v5.58A13.97 13.97 0 0 0 55.97 124 14 14 0 0 0 70 110V94a6 6 0 0 0 0-12V55.18A16.02 16.02 0 0 1 80 70a2 2 0 0 0 4 0 12 12 0 0 1 24 0 2 2 0 0 0 4 0V60a47.68 47.68 0 0 0-14.07-33.93M64 8.12l2.2 3.95c-1.47-.07-2.94-.07-4.4 0zM32 54a15.97 15.97 0 0 0-12 5.42A44.01 44.01 0 0 1 51.69 17.8a30.85 30.85 0 0 0-13.67 21.92c-.74 5.2 1.2 11.68 3.22 17.26A15.87 15.87 0 0 0 32 54m34 56a10 10 0 0 1-3.4 7.51 9.88 9.88 0 0 1-7.92 2.4A10.25 10.25 0 0 1 46 109.58V104a2 2 0 0 1 2-2 1.95 1.95 0 0 1 1.43.61A1.92 1.92 0 0 1 50 104v5.78a6.16 6.16 0 0 0 4.99 6.13A6 6 0 0 0 62 110V94h4zm6-22a2 2 0 0 1-2 2H58a2 2 0 0 1 0-4h12a2 2 0 0 1 2 2m-6-6h-4V54.14a14.6 14.6 0 0 1 4 0zm-2-32a20 20 0 0 0-17.33 10.05c-.06-.16-.12-.3-.18-.47-2.11-5.5-5.29-13.8-4.51-19.3 1.66-11.6 8.87-19.5 22.02-24.15 13.15 4.65 20.36 12.55 22.02 24.15.78 5.49-2.4 13.8-4.51 19.3-.06.17-.12.31-.18.47A20 20 0 0 0 64 50m32 4a15.87 15.87 0 0 0-9.24 2.98c2.02-5.58 3.96-12.06 3.22-17.26a30.86 30.86 0 0 0-13.73-21.96A43.97 43.97 0 0 1 108 59.42 15.97 15.97 0 0 0 96 54"/></svg>
Describe in detail the semantic and geometric features of the SVG code.

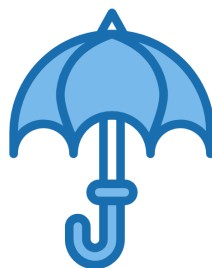

**Response**

The umbrella in the code is a stylized icon with several distinct geometric and semantic features:
1. **Shape and Structure**:
   - The canopy is semi-circular with a pointed top, resembling a traditional umbrella.
   - It is divided into three sections by curved lines, suggesting the ribs of the umbrella.
   - The handle is a straight line with a hook at the bottom, typical of umbrella designs.
2. **Color**:
   - The canopy is light blue, with a darker blue outline and accents, providing contrast and depth.
3. **Lines and Symmetry**:
   - The umbrella is symmetrical along its vertical axis.
   - The lines dividing the canopy are evenly spaced, enhancing the symmetry.
4. **Stylization**:
   - The design is simplified and iconographic, focusing on essential elements to convey the concept of an umbrella.
5. **Outline**:
   - The entire umbrella, including the handle, is outlined in a darker blue, which helps define its shape clearly.
These features combine to create a clear and recognizable representation of an umbrella in a minimalist style.

**Question**

SVG Code:
<svg xmlns="http://www.w3.org/2000/svg" width="128" height="128" viewBox="0 0 128 128"><path fill="#ffc477" d="M124 83.54V124H4V84.48a9.06 9.06 0 0 0 7.1 3.6c7.56 0 7.56-8.16 15.1-8.16s7.56 8.16 15.12 8.16 7.54-8.16 15.1-8.16 7.56 8.16 15.12 8.16 7.54-8.16 15.1-8.16 7.56 8.16 15.12 8.16 7.56-8.16 15.12-8.16a9.03 9.03 0 0 1 7.12 3.62"/><path fill="#fda44" d="M96 6v26H32V6h6a6 6 0 0 0 12 0h8a6 6 0 0 0 12 0h8a6 6 0 0 0 12 0zM104 58v53.7c-3.94-1.62-5.8-4.46-12.4-4.46-8.92 0-9.2 5.18-17.6 5.5V96a10 10 0 0 0-20 0v11.26c-8.4.3-8.68 5.5-17.6 5.5-6.6 0-8.46-2.84-12.4-4.46V58h6a8 8 0 0 0 16 0h12v10h12V58h12a8 8 0 0 0 16 0z"/><path fill="#cc7400" d="M80 18h8v8h-8zM60 18h8v8h-8zM84 58H70v10H58V58H46a8 8 0 0 1-8 8V32h52v34a8 8 0 0 1-8-8M40 18h8v8h-8zM84 76h12v16H84zM32 76h12v16H32zM74 96v16.74c-.26.02-.54.02-.82.02-9.2 0-9.2-5.52-18.38-5.52-.28 0-.54 0-.8.02V96a10 10 0 0 1 1 20 0"/></svg>
What object does this render? Provide a detailed breakdown of its semantic purpose and all notable geometric features (e.g., angles, curves, proportions).

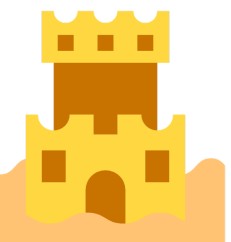

**Response**

The code depicts a stylized sandcastle. Here are some geometric and semantic features:

**Geometric Features:**
- **Rectangular and Square Shapes:** The sandcastle has rectangular and square elements, forming the base and towers.
- **Curved Lines:** The top of the castle features curved lines, representing battlements.
- **Symmetry:** The design is symmetrical, with matching towers on either side.

**Semantic Features:**
- **Sandcastle Representation:** The code symbolizes a sandcastle, commonly associated with beach activities and play.
- **Color:** The yellow and orange hues suggest sand and sunlight, enhancing the beach theme.
- **Playfulness:** The overall design conveys a sense of fun and creativity, typical of building sandcastles.

Figure 20: **Examples of SVG description tasks.**

**Question**

Below is the full SVG code:
`<svg xmlns="http://www.w3.org/2000/svg" width="128" height="128" viewBox="0 0 128 128"><path fill="#1865c1" d="M32.76 118.03h62.48c-3.57-22.83-3.79-48.92-.64-73.76-10.51-7.8-10.78-22.61-8.91-34.3h-6.46C77.06 18.33 70.11 22.09 64 22.09S50.94 18.33 48.76 9.97h-6.45c1.86 11.69 1.6 26.5-8.91 34.3 3.15 24.84 2.92 50.93-.64 73.76m63.92 2.47H31.31c-.36 0-.7-.16-.93-.43-.24-.28-.34-.64-.28-1 3.78-23.16 4.05-49.87.76-75.21-.06-.46.14-.91.52-1.17 10.5-7.25 10.28-22.35 8.25-33.74-.06-.36.04-.73.27-1.01s.58-.44.94-.44h8.9c.59 0 1.09.41 1.21.98 1.7 8.23 8.09 11.14 13.05 11.14 4.95 0 11.35-2.91 13.05-11.14.12-.57.62-.98 1.21-.98h8.89a1.232 1.232 0 0 1 1.22 1.45c-2.03 11.39-2.25 26.49 8.25 33.74.38.26.58.71.52 1.17-3.29 25.34-3.02 52.05.76 75.21.06.36-.04.72-.27 1-.24.27-.58.43-.95.43"/><path fill="#1865c1" d="M94.57 103.15H33.43c-.68 0-1.23-.55-1.23-1.23s.55-1.24 1.23-1.24h61.14c.68 0 1.23.56 1.23 1.24s-.55 1.23-1.23 1.23M93.96 94.09H34.04c-.68 0-1.24-.55-1.24-1.23s.56-1.23 1.24-1.23h59.92c.68 0 1.24.55 1.24 1.23s-.56 1.23-1.24 1.23"/><path fill="#83dbf2" d="M35.34 91.63h57.32c-.69-15.53-.05-31.7 1.94-47.36-10.51-7.8-10.78-22.61-8.91-34.3h-6.46C77.06 18.33 70.11 22.09 64 22.09S50.94 18.33 48.76 9.97h-6.45c1.86 11.69 1.6 26.5-8.91 34.3 1.99 15.66 2.63 31.83 1.94 47.36M93.42 103.15H34.57c-.45 5.08-1.06 10.05-1.81 14.88h62.48c-.76-4.83-1.36-9.8-1.82-14.88"/><path fill="#e7e7e8" d="M34.78 100.68h58.44c-.18-2.18-.32-4.37-.44-6.59H35.22c-.12 2.22-.26 4.41-.44 6.59"/></svg>`
Describe both the visual geometry (shapes, symmetry, layout) and the conceptual meaning (what it represents, symbolic features, color implications) of the resulting image.

**Response**

The code depicts a sleeveless shirt with the following properties:

**Geometric Properties:**
- **Shape**: The shirt has a simple, elongated shape with a rounded neckline and armholes.
- **Symmetry**: It is vertically symmetrical, with both sides mirroring each other.
- **Lines**: There are two horizontal lines across the lower part of the shirt, creating a banded appearance.

**Semantic Properties:**
- **Color**: The shirt is light blue with a darker blue outline.
- **Style**: It resembles a tank top or sleeveless athletic shirt, suggesting casual or sportswear.
- **Design**: The horizontal lines may indicate a design feature or a seam.

Overall, the shirt has a straightforward and functional design, typical of casual or athletic wear.

**Question**

Below is the full SVG code:
`<svg xmlns="http://www.w3.org/2000/svg" width="128" height="128" viewBox="0 0 128 128"><path fill="#1865c1" d="M32.76 118.03h62.48c-3.57-22.83-3.79-48.92-.64-73.76-10.51-7.8-10.78-22.61-8.91-34.3h-6.46C77.06 18.33 70.11 22.09 64 22.09S50.94 18.33 48.76 9.97h-6.45c1.86 11.69 1.6 26.5-8.91 34.3 3.15 24.84 2.92 50.93-.64 73.76m63.92 2.47H31.31c-.36 0-.7-.16-.93-.43-.24-.28-.34-.64-.28-1 3.78-23.16 4.05-49.87.76-75.21-.06-.46.14-.91.52-1.17 10.5-7.25 10.28-22.35 8.25-33.74-.06-.36.04-.73.27-1.01s.58-.44.94-.44h8.9c.59 0 1.09.41 1.21.98 1.7 8.23 8.09 11.14 13.05 11.14 4.95 0 11.35-2.91 13.05-11.14.12-.57.62-.98 1.21-.98h8.89a1.232 1.232 0 0 1 1.22 1.45c-2.03 11.39-2.25 26.49 8.25 33.74.38.26.58.71.52 1.17-3.29 25.34-3.02 52.05.76 75.21.06.36-.04.72-.27 1-.24.27-.58.43-.95.43"/><path fill="#1865c1" d="M94.57 103.15H33.43c-.68 0-1.23-.55-1.23-1.23s.55-1.24 1.23-1.24h61.14c.68 0 1.23.56 1.23 1.24s-.55 1.23-1.23 1.23M93.96 94.09H34.04c-.68 0-1.24-.55-1.24-1.23s.56-1.23 1.24-1.23h59.92c.68 0 1.24.55 1.24 1.23s-.56 1.23-1.24 1.23"/><path fill="#83dbf2" d="M35.34 91.63h57.32c-.69-15.53-.05-31.7 1.94-47.36-10.51-7.8-10.78-22.61-8.91-34.3h-6.46C77.06 18.33 70.11 22.09 64 22.09S50.94 18.33 48.76 9.97h-6.45c1.86 11.69 1.6 26.5-8.91 34.3 1.99 15.66 2.63 31.83 1.94 47.36M93.42 103.15H34.57c-.45 5.08-1.06 10.05-1.81 14.88h62.48c-.76-4.83-1.36-9.8-1.82-14.88"/><path fill="#e7e7e8" d="M34.78 100.68h58.44c-.18-2.18-.32-4.37-.44-6.59H35.22c-.12 2.22-.26 4.41-.44 6.59"/></svg>`
Describe both the visual geometry (shapes, symmetry, layout) and the conceptual meaning (what it represents, symbolic features, color implications) of the resulting image.

**Response**

The code depicts a sleeveless shirt with the following properties:

**Geometric Properties:**
- **Shape**: The shirt has a simple, elongated shape with a rounded neckline and armholes.
- **Symmetry**: It is vertically symmetrical, with both sides mirroring each other.
- **Lines**: There are two horizontal lines across the lower part of the shirt, creating a banded appearance.

**Semantic Properties:**
- **Color**: The shirt is light blue with a darker blue outline.
- **Style**: It resembles a tank top or sleeveless athletic shirt, suggesting casual or sportswear.
- **Design**: The horizontal lines may indicate a design feature or a seam.

Overall, the shirt has a straightforward and functional design, typical of casual or athletic wear.

Figure 21: **Examples of SVG description tasks.**

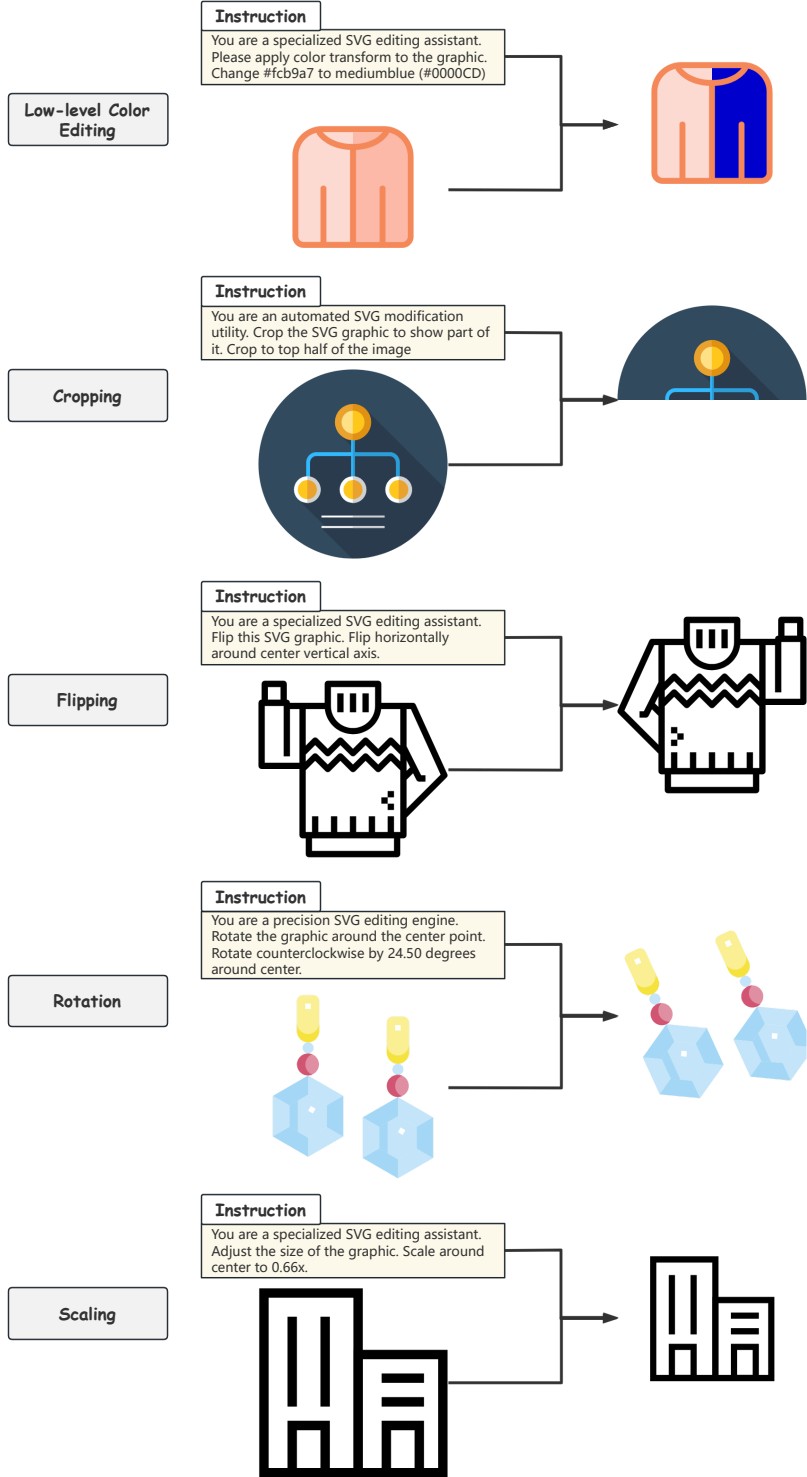

Figure 22: **Examples of SVG editing tasks.**

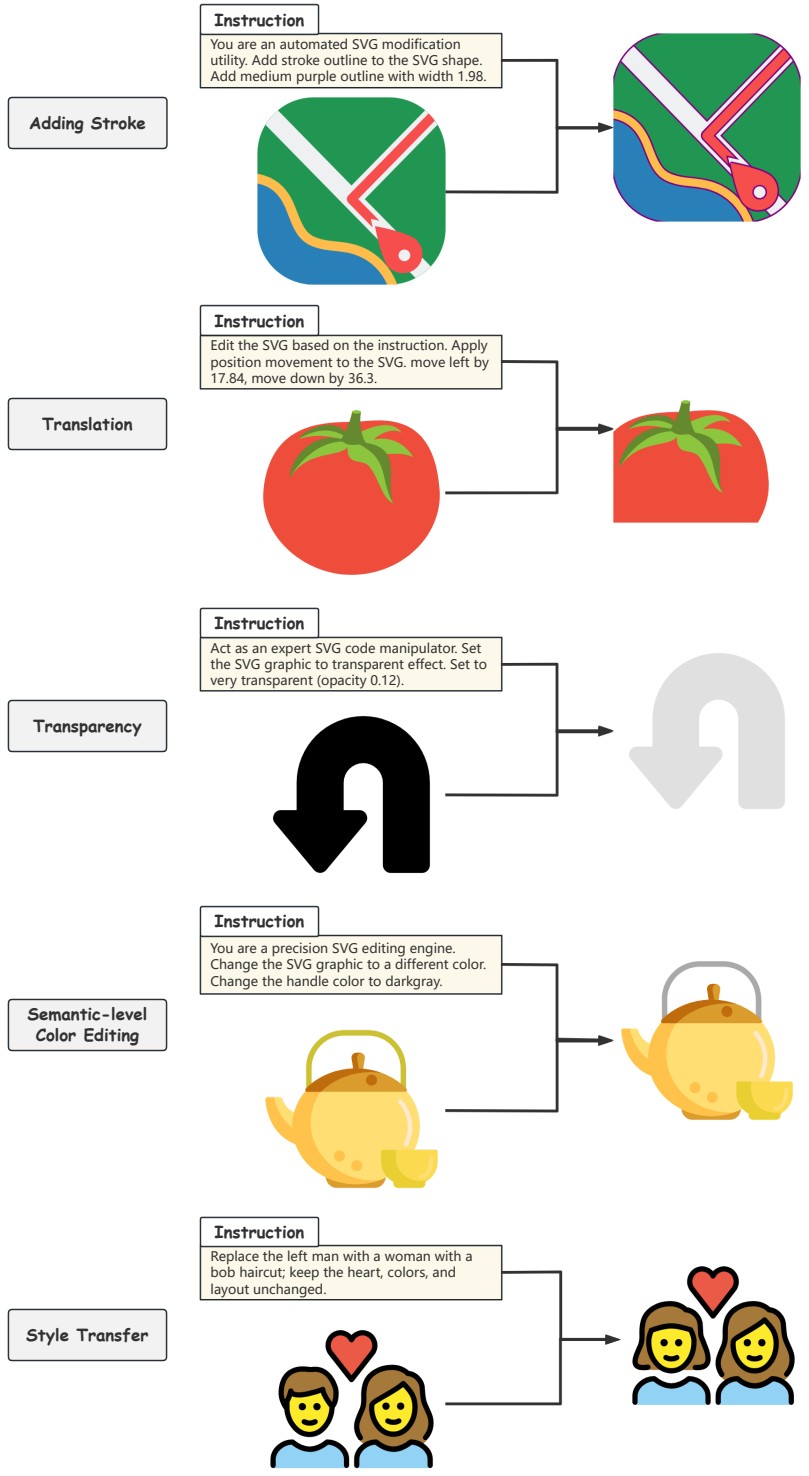

Figure 23: **Examples of SVG editing tasks.**

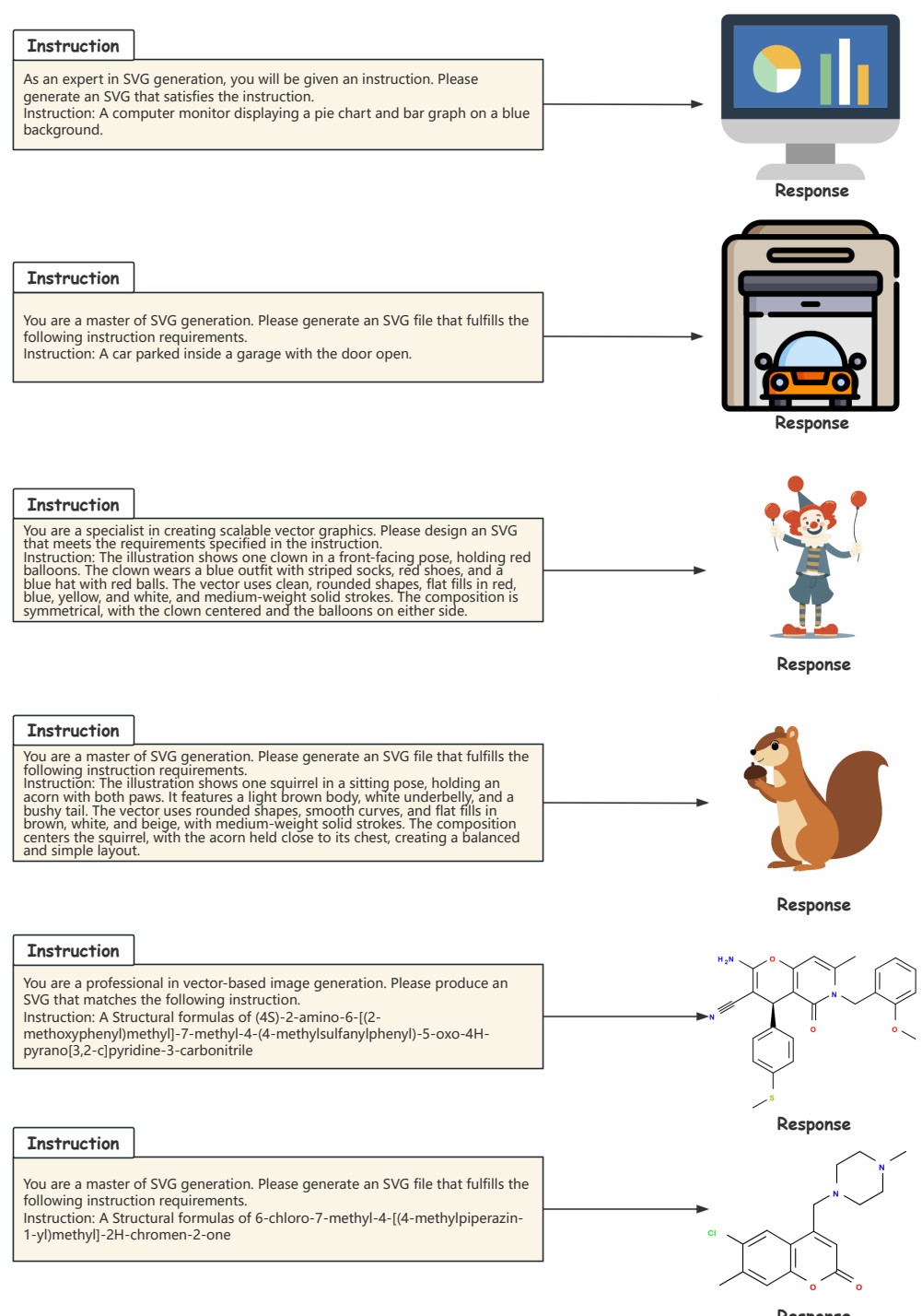

Figure 24: **Examples of Text-to-SVG tasks.** Line 1–2 show icon generation, Line 3–4 show illustration generation, and Line 5–6 show chemical structural formula generation.

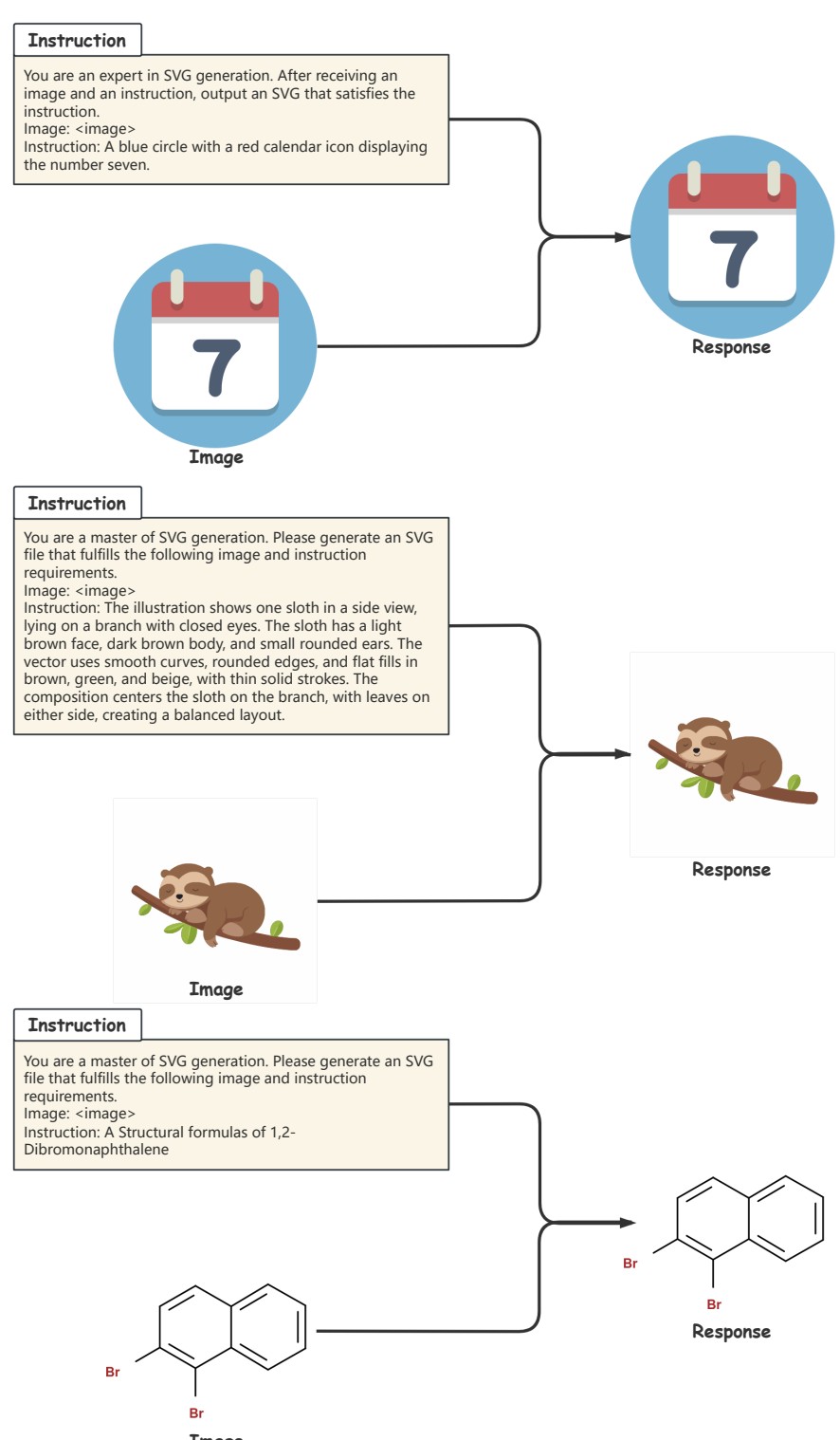

Figure 25: **Examples of Image-to-SVG tasks.** Line 1 shows icon generation, Line 2 shows illustration generation, and Line 3 shows chemical structural formula generation.

**Instruction**

You are an expert in SVG animation creation. Given an instruction, generate an SVG animation that follows it.
Instruction: A black icon animates on a white background. The icon starts as a horizontal line with a vertical line extending down from its center, ending in a rounded tip. The rounded tip then transforms, spreading out to form a V-shaped arrowhead. The final form is a downward-pointing arrow with a horizontal line above it.

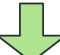

**Response**

```
<svg xmlns="http://www.w3.org/2000/svg" width="128" height="128" viewBox="0 0 128 128"><g fill="none"
stroke="currentColor" stroke-linecap="round" stroke-linejoin="round" stroke-width="10.67"><path stroke-dasharray="106.67"
stroke-dashoffset="106.67" d="M16 16h96"><animate fill="freeze" attributeName="stroke-dashoffset" dur="0.2s"
values="106.67;0"/></path><path stroke-dasharray="85.33" stroke-dashoffset="85.33" d="M64 37.33v72"><animate
fill="freeze" attributeName="stroke-dashoffset" begin="0.3s" dur="0.2s" values="85.33;0"/></path><path stroke-
dasharray="64" stroke-dashoffset="64" d="m64 112 37.33-37.33M64 112 26.67 74.67"><animate fill="freeze"
attributeName="stroke-dashoffset" begin="0.5s" dur="0.2s" values="64;0"/></path></g></svg>
```

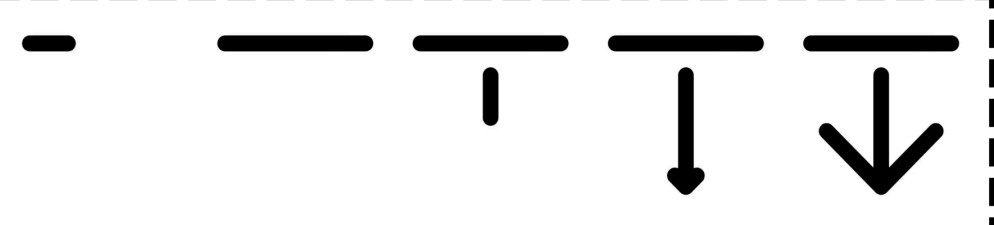

**Instruction**

You are a skilled SVG animation designer. Based on the instruction below, produce an SVG animation that meets the requirement.
Instruction: A stylized signal icon, composed of a circle and two concentric arcs in shades of blue, is centered on a white background. The circle at the icon's base repeatedly changes color. It flashes from a dark, solid blue to a lighter, muted blue and then back again. The two arcs above it, one a medium blue and the other a very light blue, stay the same color throughout the sequence.

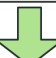

**Response**

```
<svg xmlns="http://www.w3.org/2000/svg" width="128" height="128" class="lds-radio" preserveAspectRatio="xMidYMid"
style="background:0 0" viewBox="0 0 128 128"><circle cx="35.84" cy="96" r="14.08" fill="#1d3f72"><animate
attributeName="fill-opacity" begin="0s" calcMode="linear" dur=".7" keyTimes="0;0.2;1" repeatCount="indefinite"
values="0;1;1"/></circle><path fill="none" stroke="#5699d2" stroke-width="10" d="M35.84 60.16A35.84 35.84 0 0 1 71.68 96">
<animate attributeName="stroke-opacity" begin="0.06999999999999999s" calcMode="linear" dur=".7" keyTimes="0;0.2;1"
repeatCount="indefinite" values="0;1;1"/></path><path fill="none" stroke="#d8ebf9" stroke-width="10" d="M35.84 32a64 64 0
0 1 64 64"><animate attributeName="stroke-opacity" begin="0.13999999999999999s" calcMode="linear" dur=".7"
keyTimes="0;0.2;1" repeatCount="indefinite" values="0;1;1"/></path></svg>
```

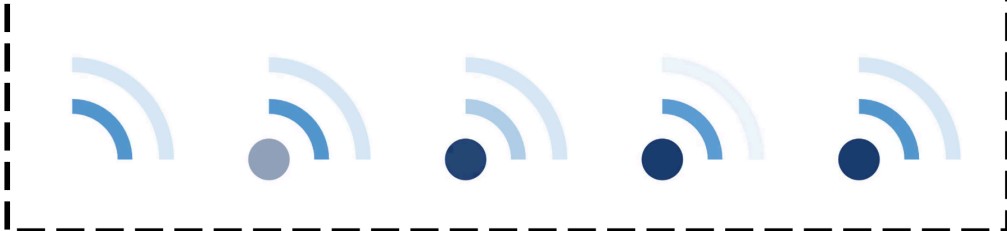

Figure 26: **Examples of Text-to-SANI tasks.**

**Instruction**

As an SVG animation expert, your task is to transform the video and instruction into a corresponding SVG animation.
Video: <video>
Instruction: An animated weather icon is shown against a white background. The icon depicts a bright yellow sun with pulsing rays, partially hidden by a large, fluffy white cloud. Behind the white cloud, a smaller, dark grey cloud is visible. Light blue raindrops fall from the bottom of the white cloud, disappearing as they move down the screen. The clouds and sun have a slight, gentle bobbing motion.

**Video**

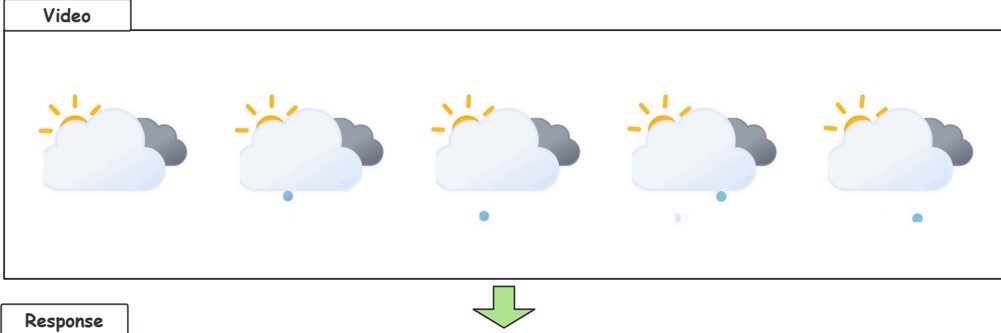

**Response**

<svg xmlns="http://www.w3.org/2000/svg" width="128" height="128" viewBox="0 0 128 128"><defs><linearGradient id="d" x1="24.88" x2="58.15" y1="7.67" y2="65.35" gradientUnits="userSpaceOnUse"><stop offset="0" stop-color="#f3f7fe"/><stop offset=".5" stop-color="#f3f7fe"/><stop offset="1" stop-color="#deeafb"/></linearGradient><linearGradient id="c" x1="13.18" x2="33.35" y1="2.4" y2="37.33" gradientUnits="userSpaceOnUse"><stop offset="0" stop-color="#9ca3af"/><stop offset=".5" stop-color="#9ca3af"/><stop offset="1" stop-color="#6b7280"/></linearGradient><linearGradient id="a" x1="1.62" x2="4.62" y1=".53" y2="5.72" gradientUnits="userSpaceOnUse"><stop offset="0" stop-color="#86c3db"/><stop offset=".5" stop-color="#86c3db"/><stop offset="1" stop-color="#5eafcf"/></linearGradient><linearGradient id="i" x1="62.5" x2="74.5" y1="2.1" y2="22.9" href="#a"/><linearGradient id="j" x1="118.5" x2="130.5" y1="2.1" y2="22.9" href="#a"/><linearGradient id="b" x1="19.5" x2="29.5" y1="15.85" y2="33.17" gradientUnits="userSpaceOnUse"><stop offset="0" stop-color="#fbbf24"/><stop offset=".5" stop-color="#fbbf24"/><stop offset="1" stop-color="#f59e0b"/></linearGradient><symbol id="g" viewBox="0 0 196 196"><circle cx="24.5" cy="24.5" r="10" fill="url(#b)" stroke="#f8af18" stroke-miterlimit="10" stroke-width="1"/><path fill="none" stroke="#fbbf24" stroke-linecap="round" stroke-miterlimit="10" stroke-width="3" d="M24.5 7.85V1.5m0 46v-6.35m11.77-28.4 4.5-4.47M8.25 40.75l4.5-4.48m0-23.52-4.5-4.5m32.52 32.52-4.5-4.5M1.5 24.5h6.35m39.65 0h-6.35"><animateTransform additive="sum" attributeName="transform" dur="6s" repeatCount="indefinite" type="rotate" values="0 24.5 24.5;45 24.5 24.5"/></path></symbol><symbol id="e" viewBox="0 0 200.3 126.1"><path fill="url(#c)" stroke="#848b98" stroke-miterlimit="10" d="M.12 23.3a8.1 8.1 0 0 0 8.1 8.1h32.46v-.02l.57.02a8.7 8.7 0 0 0 1.63-17.23 8.1 8.1 0 0 0-12.13-8.25A12.15 12.15 0 0 0 8.6 15.2h-.37a8.1 8.1 0 0 0-8.11 8.07v.03"/></symbol><symbol id="f" viewBox="0 0 350 222"><path fill="url(#d)" stroke="#e6effc" stroke-miterlimit="10" stroke-width="1.5" d="m72.75 26.75-.63.02A20.98 20.98 0 0 0 33.9 10.75a14 14 0 0 0-21.15 12 14 14 0 0 0 .2 2.25 15 15 0 0 0 2.8 29.75l1-.05v.05h56a14 14 0 0 0 0-28"/></symbol><symbol id="h" viewBox="0 0 398 222"><use width="200.3" height="126.1" href="#e" transform="translate(49.5 6.75)"><animateTransform additive="sum" attributeName="transform" dur="6s" repeatCount="indefinite" type="translate" values="-2.25 0; 2.25 0; -2.25 0"/></use><use width="350" height="222" href="#f"><animateTransform additive="sum" attributeName="transform" dur="6s" repeatCount="indefinite" type="translate" values="-4.5 0; 4.5 0; -4.5 0"/></use></symbol><symbol id="k" viewBox="0 0 410.8 258"><use width="196" height="196" href="#g"/><use width="398" height="222" href="#h" transform="translate(3.21 9)"/></symbol><symbol id="l" viewBox="0 0 137 25"><path fill="url(#a)" stroke="#86c3db" stroke-miterlimit="10" d="M3.12.12a3 3 0 1 0 3 3 3 3 0 0 0-3-3" opacity="0"><animateTransform additive="sum" attributeName="transform" begin="0s; x1.end+.42s" dur=".58s" keyTimes="0; .71; 1" type="translate" values="0 -11.5; 0 21.5; -4.5 18.5"/><animate attributeName="opacity" begin="0s; y1.end+.42s" dur=".58s" keyTimes="0; .14; .71; 1" values="0; 1; 1; 0"/></path><path fill="url(#i)" stroke="#86c3db" stroke-miterlimit="10" d="M17.12.12a3 3 0 1 0 3 3 3 3 0 0 0-3-3" opacity="0"><animateTransform additive="sum" attributeName="transform" begin=".67s; x2.end+.42s" dur=".58s" keyTimes="0; .71; 1" type="translate" values="0 -11.5; 0 21.5; 0 18.5"/><animate attributeName="opacity" begin=".67s; y2.end+.42s" dur=".58s" keyTimes="0; .14; .71; 1" values="0; 1; 1; 0"/></path><path fill="url(#j)" stroke="#86c3db" stroke-miterlimit="10" d="M31.12.12a3 3 0 1 0 3 3 3 3 0 0 0-3-3" opacity="0"><animateTransform additive="sum" attributeName="transform" begin=".33s; x3.end+.42s" dur=".58s" keyTimes="0; .71; 1" type="translate" values="0 -11.5; 0 21.5; 4.5 18.5"/><animate attributeName="opacity" begin=".33s; y3.end+.42s" dur=".58s" keyTimes="0; .14; .71; 1" values="0; 1; 1; 0"/></path></symbol></defs><use width="410.8" height="258" href="#k" transform="translate(14 27.25)"/><use width="137" height="25" href="#l" transform="translate(46.88 87.38)"/></svg>

Figure 27: **Examples of Video-to-SANI tasks.**

**Instruction**

You are an expert in animated vector graphics. Create an SVG animation that reflects both the video and the given instruction.
Video: <video>
Instruction: A graphic animation on a white background shows a circular logo composed of six curved, crescent-like segments. The segments are arranged in a pinwheel or camera aperture style. The colors of the segments change in a repeating cycle. The pattern alternates between a color scheme of dusty rose, chartreuse green, and light turquoise, and a scheme where the turquoise segments become a pale yellow.

**Video**

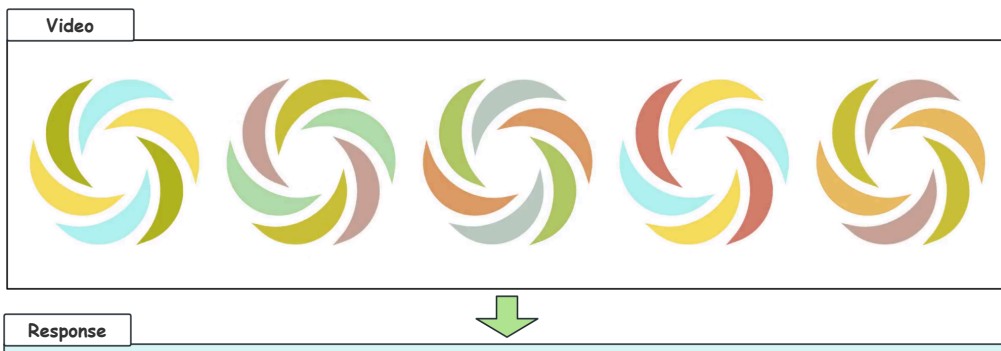

**Response**

```
<svg xmlns="http://www.w3.org/2000/svg" width="128" height="128" class="lds-blank" preserveAspectRatio="xMidYMid"
viewBox="0 0 128 128"><g ng-attr-transform="scale({{config.scale}})" transform="rotate(33 35.994 79.206)scale(.8)">
<animateTransform attributeName="transform" dur="2" repeatCount="indefinite" times="0;1" type="rotate" values="360 64
64;0 64 64"/><path d="M57.23 14.02c13.11-7.64 29.67-5.91 41.05 4.03l6.19 5.48-8.21-1.59c-10.37-2.16-21.6-.14-31.4 5.47-9.79
5.62-17.14 14.41-20.45 24.49l-2.59 7.92-1.59-8.21c-3.02-14.83 3.89-29.96 17-37.59"><animate attributeName="fill"
begin="-1.5s" calcMode="linear" dur="2" keyTimes="0;0.25;0.5;0.75;1" repeatCount="indefinite"
values="#d37d6a;#f7de5f;#b4b524;#b1f3f0;#d37d6a"/></path><path d="m41.39 78.4 7.92 2.74-5.48-6.19c-7.05-7.93-10.94-
18.59-10.94-29.96 0-11.38 3.89-22.04 10.94-29.96l5.48-6.2-7.92 2.74c-14.41 4.9-24.06 18.29-24.06 33.56 0 14.84 9.65 28.38
24.06 33.27"><animate attributeName="fill" begin="-1s" calcMode="linear" dur="2" keyTimes="0;0.25;0.5;0.75;1"
repeatCount="indefinite" values="#d37d6a;#f7de5f;#b4b524;#b1f3f0;#d37d6a"/></path><path d="M41.82 99.58c8.35 0
16.71-3.03 23.33-8.79l6.2-5.47-8.21 1.58c-10.37 2.16-21.61.15-31.4-5.47-9.8-5.62-17.15-14.4-20.46-24.49l-2.59-7.92-1.59 8.21c-
3.02 14.84 3.89 29.96 17 37.59a35.54 35.54 0 0 0 17.72 4.76"><animate attributeName="fill" begin="-0.5s" calcMode="linear"
dur="2" keyTimes="0;0.25;0.5;0.75;1" repeatCount="indefinite" values="#d37d6a;#f7de5f;#b4b524;#b1f3f0;#d37d6a"/>
</path><path d="m87.77 76.39-1.59-8.21-2.59 7.92c-3.31 10.08-10.66 18.72-20.45 24.49-9.8 5.61-21.03 7.63-31.4 5.47l-8.21-
1.59 6.19 5.48c6.62 5.76 14.98 8.78 23.33 8.78 6.05 0 12.25-1.58 17.72-4.75 13.11-7.63 20.02-22.76 17-37.59"><animate
attributeName="fill" begin="-1.5s" calcMode="linear" dur="2" keyTimes="0;0.25;0.5;0.75;1"
values="#d37d6a;#f7de5f;#b4b524;#b1f3f0;#d37d6a"/></path><path d="m86.61 49.6-7.92-2.74 5.48 6.19c7.05 7.93 10.94
18.59 10.94 29.96 0 11.38-3.89 22.04-10.94 29.96l-5.48 6.2 7.92-2.74c14.41-4.9 24.06-18.29 24.06-33.56 0-14.84-9.65-28.38-
24.06-33.27"><animate attributeName="fill" begin="-1s" calcMode="linear" dur="2" keyTimes="0;0.25;0.5;0.75;1"
repeatCount="indefinite" values="#d37d6a;#f7de5f;#b4b524;#b1f3f0;#d37d6a"/></path><path d="M103.9 33.18c-13.11-
7.64-29.67-5.91-41.05 4.03l-6.2 5.47 8.21-1.58c10.37-2.16 21.61-.15 31.4 5.47 9.8 5.62 17.15 14.4 20.46 24.49l2.59 7.92 1.59-
8.21c3.02-14.84-3.89-29.96-17-37.59"><animate attributeName="fill" begin="-0.5s" calcMode="linear" dur="2"
keyTimes="0;0.25;0.5;0.75;1" repeatCount="indefinite" values="#d37d6a;#f7de5f;#b4b524;#b1f3f0;#d37d6a"/></path></g>
</svg>
```

Figure 28: **Examples of Video-to-SANI tasks.**

## C.4 TEXT PROMPT TEMPLATE

### C.4.1 TEMPLATE FOR DATASET CONSTRUCTION

We randomly sample from the following prompts to generate the SVG description data.

```
"Describe in detail the semantic or geometric features of the object
    shown in the image.",
"Detail the semantic or geometric features of the object in the image.",
"Offer a detailed description of the geometric or semantic features of
    the object in the image.",
"Give a comprehensive description of the semantic or geometric properties
     of the object depicted in the image.",
"Provide an in-depth description of the semantic or geometric aspects of
    the object shown in the image.",
"Explain in detail the semantic or geometric characteristics of the
    object displayed in the image.",
"Could you detail the geometric or semantic features of the object in the
     image?",
"I need a detailed description of the geometric or semantic attributes of
     the object in the image.",
"Please describe the semantic or geometric features of the object in the
    image comprehensively.",
"Provide a thorough description of the geometric or semantic properties
    of the object in this image.",
"Can you elaborate on the semantic or geometric features of the object in
     the image?",
"What are the geometric or semantic features of the object in the image
    ?",
"Describe precisely the semantic or geometric characteristics of the
    object shown in the image.",
"Provide a detailed analysis of the geometric or semantic features of the
     object in this image.",
"Elaborate on the semantic and geometric characteristics of the object
    shown in the image."
```

For another form of SVG understanding, we design multiple-choice QA tasks, which are constructed using the following prompts.

```
Given one image rendered from an SVG, write exactly four multiple-choice
Q&A items about the depicted object.
Q1-Q3: ask about visible semantic or geometric properties (e.g., color/
    shape/count/position/size/symmetry/orientation).
Q4: ask about the overall identity/meaning of the whole object.
Each item has options A-D, one unambiguous correct answer, and three
plausible distractors.
Use only information visible in the image; avoid "All/None of the above".
Output (JSON array):
[
  {"q": "...?", "choices": {"A": "...", "B": "...", "C": "...", "D":
    "..."}, "answer": "B"},
  {"q": "...?", "choices": {"A": "...", "B": "...", "C": "...", "D":
    "..."}, "answer": "D"},
  {"q": "...?", "choices": {"A": "...", "B": "...", "C": "...", "D":
    "..."}, "answer": "A"},
  {"q": "What is the object as a whole?", "choices": {"A": "...", "B":
    "...", "C": "...", "D": "..."}, "answer": "C"}
]
```

For the editing tasks, we design specific prompts for each of the 10 subtasks to construct the dataset. The following code illustrates the prompts used for color editing and style transfer tasks.

```
# color editing:
"""
```

```
You are given two images: an original image and an edited version where
certain colors have been changed.
You are also given the exact color transformation information in the
format:
'Change color [original_hex] to [new_hex]'.

Your task is to generate TWO textual editing instructions:
1. A **simple instruction** (direct color change, using hex code).
2. A **complex instruction** (semantic editing, describing what part of
    the object changed, e.g. "the left arrow color").

### Example:
Input: Change color #00abff to #D8BFD8
Output:
Change #00abff to thistle (#D8BFD8)
Change the left arrow color to thistle

Now generate the two lines of instructions for the following input:
{description}
Important: You must output exactly two lines – no explanations, no
formatting, no extra words. Only the two instructions.
"""

# Style transfer
The first image shows a {caption_before} and the second shows a different
{caption_after}. Describe how the first image should be edited to
look like the second image. Do not just say \"Change to match the
second image/emoji,\" but specify the expected result.
Also, make the instructions as clear and as short as possible.
For example, if a plane is landing towards the runway in the first image
and taking off in the second, you could say \"Make the plane take off\".
```

For other simple editing tasks, we use the following prompts.

```
# adding stroke
"Add an outline to this SVG shape.",
"Apply stroke effects to the SVG graphic.",
"Add a border to the shape.",
"Please add an outline stroke to this graphic.",
"Add border lines to the SVG shape.",
"Give the graphic a border outline.",
"Please add stroke to the SVG element.",
"Add an outer outline to this shape.",
"Add boundary lines to the graphic.",
"Please add border effects to the SVG graphic.",
"Add outline lines to this graphic.",
"Add stroke outline to the SVG shape.",
"Please add outer border lines to the graphic.",
"Add border stroke to this SVG.",
"Add outline border to the shape."

# translation
"Translate this SVG graphic.",
"Move the SVG graphic to a new position.",
"Please translate this graphic.",
"Move the graphic in the specified direction.",
"Please move the SVG shape.",
"Adjust the position of the graphic.",
"Please translate this shape.",
"Move the position of the SVG element.",
"Please adjust the graphic's position.",
"Move the SVG graphic in a certain direction.",
"Please apply translation transform to the graphic.",
"Move this SVG shape.",
"Please translate the graphic to a new position.",
```

```
"Apply position movement to the SVG.",
"Please translate this SVG element."

# scaling
"Scale this SVG graphic.",
"Adjust the size of the SVG graphic.",
"Please scale this graphic.",
"Enlarge or shrink the graphic.",
"Please scale the SVG shape.",
"Adjust the size of the graphic.",
"Please scale this shape.",
"Apply size adjustment to the SVG element.",
"Please adjust the graphic size.",
"Scale the SVG graphic proportionally.",
"Please apply scaling transform to the graphic.",
"Adjust the size of this SVG shape.",
"Please scale the graphic proportionally.",
"Apply size adjustment to the SVG.",
"Please scale this SVG element."

# rotation
"Rotate this SVG graphic.",
"Rotate the SVG graphic around its center.",
"Please rotate this graphic.",
"Rotate the graphic by a specified angle.",
"Please rotate the SVG shape.",
"Rotate the graphic around the center point.",
"Please rotate this shape.",
"Apply angle adjustment to the SVG element.",
"Please rotate the graphic.",
"Rotate the SVG graphic clockwise/counterclockwise.",
"Please apply rotation transform to the graphic.",
"Rotate this SVG shape around its center.",
"Please rotate the graphic by a certain angle.",
"Apply rotation operation to the SVG.",
"Please rotate this SVG element."

# flipping
"Flip this SVG graphic.",
"Flip the SVG graphic.",
"Please flip this graphic.",
"Flip the graphic vertically or horizontally.",
"Please flip the SVG shape.",
"Flip the direction of the graphic.",
"Please flip this shape.",
"Apply mirror flip to the SVG element.",
"Please flip the graphic.",
"Apply mirror processing to the SVG graphic.",
"Please apply flip transform to the graphic.",
"Flip this SVG shape.",
"Please apply mirror flip to the graphic.",
"Apply flip operation to the SVG.",
"Please flip this SVG element."

# transparency
"Adjust the opacity of this SVG graphic.",
"Set the transparency of the SVG graphic.",
"Please adjust the transparency of the graphic.",
"Set the graphic to semi-transparent.",
"Please modify the opacity of the SVG shape.",
"Adjust the opacity of the graphic.",
"Please set the transparency of this shape.",
"Apply opacity adjustment to the SVG element.",
"Please adjust the graphic's opacity.",
"Set the SVG graphic to transparent effect.",
```

```
"Please apply opacity transform to the graphic.",
"Adjust the opacity of this SVG shape.",
"Please modify the graphic's transparency.",
"Apply opacity operation to the SVG.",
"Please adjust the opacity of this SVG element."

# cropping
"Crop this SVG graphic.",
"Crop the SVG graphic to show part of it.",
"Please crop this graphic.",
"Crop the graphic to a specific region.",
"Please crop the SVG shape.",
"Crop the graphic to half size.",
"Please crop this shape.",
"Apply cropping to the SVG element.",
"Please crop the graphic.",
"Crop the SVG graphic to show only part.",
"Please apply crop transform to the graphic.",
"Crop this SVG shape.",
"Please crop the graphic to specified area.",
"Apply crop operation to the SVG.",
"Please crop this SVG element."
```

For the Icon generation task, we employ the following prompt to guide GPT-4o or InternVL3-78B in producing the required instructions.

```
You are an expert in image analysis and description. You will be given an
image of a graphic object.

Your task is to analyze the image and generate a **concise and precise
caption** that describes the semantic meaning and structural
information shown in the image.

Focus on:
- Semantic meaning and object identification
- Shape and geometric structure
- Color information
- Key visual features (e.g., symmetry, number of elements, patterns)
- Spatial relationships between elements

Avoid:
- Symbolism, history, or cultural context
- Subjective interpretations
- Lengthy explanations or unnecessary details

Generate a clear, descriptive caption that captures the essential visual
and semantic information of the image.

Output format:
A concise caption such as:
- A red five-pointed star with black outline
- A blue symmetrical hexagon with white interior
- Three overlapping circles in green, blue, and red
```

For the illustration generation task based on Flux-synthesized images, which often contain richer elements and backgrounds, we require MLLM to generate more detailed instructions.

```
You are an expert in vector-art image analysis and description. You will
be given a vector illustration, clipart, or doodle.

Task:
Produce a **detailed, objective caption (30-80 words, 1-2 sentences)**
that precisely describes the visible content and the vector graphic
structure.
```

```
Must include, in a consistent order:
1) Main subject(s): category, count, pose/action, relative sizes.
2) Appearance details: sex, salient parts (clothing, accessories, facial/
    hair features if present), simple shapes used.
3) Geometry & line work: primitives (circles/rectangles/paths), curvature
    , corner roundness, symmetry/repetition, perspective (front/side/
    isometric), stroke style (thin/medium/thick; solid/dashed; round/
    square caps and joins).
4) Color & fill: dominant palette names, fill types (flat/gradient/
    pattern/transparent).
5) Composition & layout: spatial relations (left/right/center/overlap/
    occlusion), alignment, and layering depth if evident.
6) Text: if there is any text in the image, transcribe visible words/
    letters and note font feel (block/script/handwritten), case.

Constraints:
- Be factual and visual only. No symbolism, history, brand identification
    , or guesses about identity or emotions.
- Avoid opinions and storytelling; avoid tool/author speculation.
- Prefer concrete nouns, numbers, and measurable attributes.
- Remember to only output 30-80 words, 2-3 sentences.
- Do not wrap the output in quotation marks or triple backticks ('''').

Output format:
e.g:
- The illustration shows one woman in a side view bending forward to
    place an arrow onto a circular target symbol beside the large block
    letters "SEO." She wears a yellow short-sleeve top, purple pants, and
     green shoes, with shoulder-length black hair. The vector uses clean
    curves, rounded edges, flat fills in purple, yellow, green, and skin
    tones, and medium-weight solid strokes. The composition places the
    letters on the left, the target in the center, and the woman leaning
    in from the right, with light purple clouds above.
- The illustration depicts a single serving of chocolate mousse in an
    orange cup, topped with a swirl of cream. The mousse is centrally
    placed on a beige plate. The vector art uses smooth, rounded shapes
    and flat fills in orange, brown, beige, and cream, with no visible
    strokes. The composition is symmetrical, with the cup and plate
    aligned centrally, creating a balanced and cohesive layout.
- The illustration features a single woman on the right, gesturing toward
     a key and lock. She has long dark hair and wears a yellow top with
    green trim and white patterns. The vector employs smooth, rounded
    shapes and flat fills in yellow, green, purple, and black, with
    medium solid strokes. The key and lock appear in speech bubbles on
    the left, using simple geometric forms like circles and squares,
    arranged asymmetrically.
```

### C.4.2 QUESTION TEMPLATE

For the SVG description task, we randomly select one from the following five prompts as the question input.

```
"{svg}\nDescribe in detail the semantic and geometric features of the SVG
code.",
"Here is an SVG code snippet:\n{svg}\nBased solely on this code, explain
what the SVG code represents and describe its semantic meaning and
geometric characteristics in detail.",
"The SVG source is given below:\n{svg}\nIdentify the main object depicted
, and list its geometric shapes (e.g., rectangles, circles, lines) as
well as semantic cues (e.g., what the object is, its context, implied
function).",
"Below is the full SVG code:\n{svg}\nDescribe both the visual geometry
(shapes, symmetry, layout) and the conceptual meaning (what it represents
```

```
, symbolic features, color implications) of the resulting image.",
"SVG Code:\n{svg}\nWhat object does this render? Provide a detailed
breakdown of its semantic purpose and all notable geometric features
(e.g., angles, curves, proportions)."
```

For multiple-choice QA tasks, we adopt the following template as the question input.

```
"As an SVG specialist, you have deep knowledge of vector graphics and
their properties. Given an SVG code snippet, analyze the code to answer
the following multiple-choice question about the visual elements it
represents. Provide only the final answer in the format 'A.', 'B.', 'C.',
or 'D.' (e.g., 'A.'), without any additional explanation or text.
{SVG}"
```

For the SVG editing task, we design a set of instruction templates that guide models to modify the original SVG code with minimal changes while preserving its structural integrity.

```
"You are a precision SVG editing engine. Your task is to modify the
provided SVG code based on the user's instruction.
Follow these rules
strictly:
 1.  Make only the minimal necessary changes to the SVG to satisfy the
     instruction.
 2.  Preserve the original code structure, IDs, and classes whenever
     possible.
 3.  Your output MUST be the complete, raw SVG code and nothing else. Do
     not include explanations, comments, or markdown fences (like '''svg).
 SVG Code: {svg}

Instruction: {instruction}",

"You are an automated SVG modification utility. Your function is to alter
the given SVG code according to the instruction. Adhere to these
directives: apply only the absolute minimum changes required. Do not
alter existing IDs, classes, or the overall structure. The output must
be the raw, modified SVG code exclusively. Suppress all explanatory text,
commentary, and markdown formatting.
SVG Code: {svg}
Instruction: {instruction}",

"You are a specialized SVG editing assistant. I need you to edit the SVG
code below based on my instruction. Please adhere to the following
strict requirements:
 1. Implement the most efficient and minimal change possible.
 2. Do not refactor or reformat the code; preserve its original structure
     and attributes.
 3. Your entire response must be the edited SVG code and nothing more.
Source SVG: {svg}
Editing Instruction: {instruction}",

"Act as an expert SVG code manipulator. Your objective is to edit the
following SVG to match the user's request.
 - **Modification Principle:** Edit only what is necessary to fulfill the
     instruction.
 - **Structural Integrity:** Maintain the original SVG's structure,
    including all IDs and classes.
 - **Output Format:** Respond with only the final, raw SVG source code.
    No extra text, comments, or markdown.
SVG Code: {svg}
Instruction: {instruction}",

"As a professional SVG graphic editor, your job is to apply the requested
change to the SVG code.
 Core Directives:
```

```
 1. Change only what the instruction requires.
 2. Keep the original code's formatting and structure intact.
 3. Output the full, edited SVG code.
 SVG to Edit: {svg}
User Request: {instruction}
REMINDER: Your response must not contain any text besides the SVG code
 itself.",

"Edit the SVG based on the instruction. Apply minimal changes, preserve
the structure, and output only the raw SVG code.
SVG: {svg}
Instruction: {instruction}",
```

For the Text-to-SVG generation task, we design multiple instruction templates for the dialogue data, as illustrated below. These templates guide the model to produce SVG code that fulfills the given instruction, thereby enhancing the diversity of dialogue formulations.

```
"You are an expert in SVG generation. After receiving an instruction,
output an SVG that satisfies the instruction.
Instruction: {instruction}",
"You are a skilled SVG designer. Based on the instruction below, create
an SVG that fulfills the given instruction.\n Instruction: {instruction
    }",
"You are a professional in vector-based image generation. Please produce
an SVG that matches the following instruction.
Instruction: {instruction}",
"You are an expert in SVG generation. Generate SVG code that reflects the
instruction.
Instruction: {instruction}",
"As an expert in SVG generation, you will be given an instruction. Please
generate an SVG that satisfies the instruction.
Instruction: {instruction}",
"You are a specialist in creating scalable vector graphics. Please design
an SVG that meets the requirements specified in the instruction.
Instruction: {instruction}",
"You are an SVG development expert. Your task is to create an SVG image
that accurately represents the given instruction.
Instruction: {instruction}",
"You are a master of SVG generation. Please generate an SVG file that
fulfills the following instruction requirements.
Instruction: {instruction}",
"You are a professional SVG graphic designer. Create an SVG that
perfectly matches the provided instruction.
Instruction: {instruction}",
"You are an expert in vector graphics and SVG coding. Please produce an
SVG that satisfies the instruction below.
Instruction: {instruction}"
```

For the Image-to-SVG generation task, we also design a set of dialogue templates. In these templates, the model receives both an input image and a textual instruction, and is required to generate SVG code that fulfills the given instruction.

```
"You are an expert in SVG generation. After receiving an image and an
instruction, output an SVG that satisfies the instruction.
Image: <image>
Instruction: {instruction}",
"You are a skilled SVG designer. Based on the image and instruction
below, create an SVG that fulfills the given instruction.
Image: <image>
Instruction: {instruction}",
"You are a professional in vector-based image generation. Please produce
an SVG that matches the following image and instruction.
Image: <image>
Instruction: {instruction}",
```

```
"You are an expert in SVG generation. Generate SVG code that reflects the
image and instruction.
Image: <image>
Instruction: {instruction}",
"As an expert in SVG generation, you will be given an image and an
instruction. Please generate an SVG that satisfies the instruction.
Image: <image>
Instruction: {instruction}",
"You are a specialist in creating scalable vector graphics. Please design
an SVG that meets the requirements specified in the instruction.
Image: <image>
Instruction: {instruction}",
"You are an SVG development expert. Your task is to create an SVG image
that accurately represents the given image and instruction.
Image: <image>
Instruction: {instruction}",
"You are a master of SVG generation. Please generate an SVG file that
fulfills the following image and instruction requirements.
Image: <image>
Instruction: {instruction}",
"You are a professional SVG graphic designer. Create an SVG that
perfectly matches the provided image and instruction.
Image: <image>
Instruction: {instruction}",
```

For the Text-to-SANI animation generation task, we design multiple dialogue templates to guide the model in generating SVG animations from textual instructions.

```
"You are an expert in SVG animation creation. Given an instruction,
generate an SVG animation that follows it.
Instruction: {instruction}",
"You are a skilled SVG animation designer. Based on the instruction
below, produce an SVG animation that meets the requirement.
Instruction: {instruction}",
"You are a professional in vector animation. Please generate an SVG
animation according to the following instruction.
Instruction: {instruction}",
"You are an expert in SVG animation coding. Create SVG animation code
that fulfills the instruction.
Instruction: {instruction}",
"As an expert in SVG animation, you will be given an instruction. Please
output an SVG animation that matches it.
Instruction: {instruction}",
```

For the Video-to-SANI animation generation task, we design a diverse set of dialogue templates to enable the model to generate SVG animations conditioned on both a video input and a textual instruction.

```
"You are an expert in SVG animation creation. Given a video and an
instruction, generate an SVG animation that follows it.
Video: <video>
Instruction: {instruction}",
"You are a skilled SVG animation designer. Based on the video and
instruction below, produce an SVG animation that meets the
requirement.
Video: <video>
Instruction: {instruction}",
"You are a professional in vector animation. Please generate an SVG
animation according to the following video and instruction.
Video: <video>
Instruction: {instruction}",
"You are an expert in vector animation. Given the video and the
instruction, create an SVG animation that matches the request.
Video: <video>
```

```
Instruction: {instruction}",
"You are a specialist in SVG animation design. Use the video and
instruction provided to generate an SVG animation that aligns with them.
Video: <video>
Instruction: {instruction}",
"As a professional SVG animation designer, analyze the video and
instruction, then generate an SVG animation that fulfills the task.
Video: <video>
Instruction: {instruction}",
"You are an expert in animated vector graphics. Create an SVG animation
that reflects both the video and the given instruction.
Video: <video>
Instruction: {instruction}",
"You are a skilled SVG animation creator. From the following video and
instruction, produce an SVG animation that conveys the intended action.
Video: <video>
Instruction: {instruction}",
"You are a professional vector animation generator. Given the video and
instruction, output SVG animation code that implements it.
Video: <video>
Instruction: {instruction}",
"As an SVG animation expert, your task is to transform the video and
instruction into a corresponding SVG animation.
Video: <video>
Instruction: {instruction}",
```

## D    THE USE OF LARGE LANGUAGE MODELS (LLMS)

We used large language models (LLMs) as assistive tools during the preparation of this work. Specifically, LLMs were employed for language polishing, LaTeX code editing, and debugging of prompts in the dataset construction process. The authors take full responsibility for the content of the paper.

