# OpenReview forum: "InternSVG: Towards Unified SVG Tasks with Multimodal Large Language Models"
_ICLR.cc/2026/Conference — ICLR 2026 Poster_

### Official Review · Reviewer_rkyc · 2025-10-29

**Soundness:** 3
**Presentation:** 3
**Contribution:** 3
**Rating:** 6
**Confidence:** 4

**Summary:**

This paper aims to address the challenges of fragmented datasets and poor method transferability in the current field of SVG modeling. To this end, the authors propose a comprehensive solution named "InternSVG," which consists of three parts:
1.  **SAgoge**: A large-scale (over 16 million samples) and comprehensive multimodal SVG dataset.
2.  **SArena**: A corresponding standardized benchmark for evaluation.
3.  **InternSVG**: A unified Multimodal Large Language Model (MLLM), which is based on the mainstream ViT-LLM architecture and optimized with methods such as SVG-specific tokenization and a two-stage training strategy.

**Strengths:**

1.  **Dataset and Benchmark Contribution**: The most prominent strength of this paper is the creation of the SAgoge dataset and the SArena benchmark. This is a tremendous engineering contribution that provides an extremely valuable resource to the SVG research community. The scale, diversity (especially the inclusion of chemical structures and animations), and unified coverage of understanding, editing, and generation tasks in SAgoge fill a significant gap in the field.
2.  **Effectiveness of Unified Modeling Validated**: The paper clearly argues for and experimentally demonstrates the advantages of unified modeling. By jointly training on understanding, editing, and generation tasks within the same model, the model achieves better generalization and performance.
3.  **Comprehensive Experimental Evaluation**: The authors conducted a series of thorough experiments, comparing their method against a wide range of baselines, including traditional methods, general-purpose MLLMs, proprietary models, and specialized SVG generation models. The InternSVG model achieved state-of-the-art (SOTA) performance on their proposed SArena benchmark, which fully demonstrates the effectiveness of their method suite.

**Weaknesses:**

1.  **High Reliance on Synthetic Data**: A large portion of the illustrations and animations in the dataset are synthesized using other generative models (e.g., FLUX, Claude), and the annotations rely on automated processes (e.g., GPT-4o). This poses a potential risk: the dataset may contain systematic biases or stylistic limitations originating from these generation/annotation models, and may not fully reflect the diversity and complexity of SVGs created by human designers in the real world.
2.  **Insufficient Evaluation of Generated Code Quality**: The paper's evaluation metrics primarily focus on the post-rendering visual effects (e.g., PSNR, SSIM, FID). However, as a vector format, the quality of the SVG code itself—such as conciseness, structural soundness, and editability—is equally crucial. The paper lacks a direct quantitative evaluation of the quality of the generated code. It is possible that the model generates visually correct but redundant and hard-to-maintain SVG code.

**Questions:**

1.  Regarding synthetic data bias: Have the authors analyzed potential issues of pattern repetition or stylistic homogenization introduced by the synthetic data pipeline (e.g., using FLUX and Claude models)? Is there a noticeable gap in complexity or creativity when comparing this synthetic data to real, human-authored SVGs?
2.  Regarding SVG code quality: Besides visual evaluation, have the authors considered introducing metrics to assess the quality of the generated SVG *code* itself? For instance, code length (token count), path complexity, or the number of nodes. This would be very important for measuring the model's efficiency and the editability of the generated content.
3.  Regarding the challenges of the "style transfer" task: The paper mentions that the model's performance on the "style transfer" task is slightly inferior to the strongest baseline. Could the authors provide a specific analysis of the unique challenges this task presents to the model, which makes it difficult for the current unified training framework to handle perfectly?

---

> ### Author Response · Authors · 2025-11-21
>
> Dear Reviewer rkyc,
>
> Thank you for your thoughtful review and for your positive assessment of our work, as well as the constructive suggestions that help us further improve the paper.
>
> **For Weakness 1 and Question 1**, we acknowledge the concern regarding potential biases from synthetic data and implemented several measures to mitigate pattern repetition and stylistic homogenization.
>
> **First, for illustration and animation generation, we explicitly avoided fixed templates or a narrow set of prompts.** Instead, we adopted an open-ended, diversified instruction pipeline that drives FLUX and Claude with randomized natural-language prompts. This design encourages a broad range of compositions, styles, and semantics, and reduces the risk that the synthetic data collapses to a single dominant "model style". **Second, we applied pHash deduplication, removing samples below a distance threshold to filter out redundant outputs and preserve diversity in shape and style.**
>
> To more directly assess whether there is a noticeable gap in quality or "human-likeness" compared to real human-authored SVGs, we additionally conducted a user study on the synthesized illustration and animation data. For each participant, we randomly sampled five illustration SVGs and five animation SVGs, presenting both the SVG code and the rendered images. Participants were asked to evaluate: (i) whether the image contained any visible defects; (ii) the likelihood that the content was created by a human designer (Human-Likeness Score); and (iii) the quality and readability of the SVG code (Code Quality Score).
>
> We collected 42 valid responses from researchers (14.29%), PhD students (57.14%), and UI designers (28.57%). The results indicate that for **illustrations**, **96.19%** of samples were judged to contain no defects, with a **Human-Likeness Score of 4.20** and a **Code Quality Score of 4.31**. For **animations**, **93.81%** of samples were considered defect-free, with a **Human-Likeness Score of 4.07** and a **Code Quality Score of 4.23**. These findings demonstrate that synthesized data is generally of high quality and closely aligned with human design conventions.

---

> ### Author Response · Authors · 2025-11-21
>
> **For Weakness 2 and Question 2**, we appreciate the reviewer’s emphasis on explicitly evaluating SVG code quality rather than only post-rendering visual metrics. Therefore, in our supplementary experiments, we conducted a systematic quantitative analysis of the structural complexity of the generated SVG code.
>
> We used the SArena-Icon benchmark (6,013 real human-designed icons) as the reference dataset to ensure that the analysis reflects the structural properties expected in real design scenarios. Following the complexity metrics in Svgenius [1], we computed the number of paths via XML parsing, assessed the conciseness of command structures, and compared the average inference time across models on Text-to-SVG tasks. In addition, we further measured the average counts of SVG primitives to more comprehensively evaluate each baseline's structural expressiveness.
>
> | Model        | Path   | Rect | Circle | Polygon | Line | Ellipse | Polyline | Time  |
> | ------------ | ------ | ---- | ------ | ------- | ---- | ------- | -------- | ----- |
> | GT           | 2.98   | 0.35 | 0.29   | 0.17    | 0.10 | 0.06    | 0.02     | -     |
> | SVGDreamer   | 256.00 | 0.00 | 0.00   | 0.00    | 0.00 | 0.00    | 0.00     | 289s  |
> | VectorFusion | 64.00  | 0.00 | 0.00   | 0.00    | 0.00 | 0.00    | 0.00     | 71s   |
> | IconShop     | 6.19   | 0.00 | 0.00   | 0.00    | 0.00 | 0.00    | 0.00     | 5.7s  |
> | LLM4SVG      | 2.81   | 0.03 | 0.17   | 0.00    | 0.00 | 0.01    | 0.00     | 9.7s  |
> | OminiSVG     | 4.30   | 0.00 | 0.00   | 0.00    | 0.00 | 0.00    | 0.00     | 12.3s |
> | InternSVG    | 4.48   | 0.25 | 0.20   | 0.17    | 0.07 | 0.02    | 0.01     | 6.9s  |
>
> As shown in the table, InternSVG demonstrates strong performance in structural expressiveness, editability, and inference efficiency. In terms of expressiveness, InternSVG natively supports the full set of SVG primitives, rather than approximating shapes with large numbers of Bézier curves as in traditional optimization-based approaches (e.g., SVGDreamer [2], VectorFusion [3]). This primitive-level representation is also emphasized as crucial in recent work like StarVector [4]. InternSVG is able to express diverse geometric structures more naturally and accurately.
>
> Regarding editability, InternSVG generates an average of only 4.48 paths, substantially fewer than traditional methods such as SVGDreamer (256.00 paths) and VectorFusion (64.00 paths). These methods produce highly fragmented path structures that are difficult to edit, whereas InternSVG outputs clean and semantically coherent structures that align much better with practical editing workflows in professional design software.
>
> In terms of inference efficiency, traditional approaches rely on pixel-level optimization and require tens to hundreds of seconds (e.g., 289 s for SVGDreamer and 71 s for VectorFusion). Thanks to the explicit structural encoding in our tokenizer, InternSVG achieves an average inference time of only 6.9 seconds, significantly improving efficiency while maintaining high-quality structural outputs.

---

> ### Author Response · Authors · 2025-11-21
>
> **For Question 3,** we provide additional clarification regarding why the performance on this task is slightly weaker under the unified model framework.
>
> First, from a data perspective, style transfer inherently suffers from data scarcity. The task requires pairs of SVGs that are semantically identical but differ only in style. However, existing public SVG datasets rarely provide such fine-grained aligned pairs, nor do they offer detailed category-level organization to reliably construct them. As a result, the StyleTransformer subset contains only 60k usable paired samples—substantially fewer than those available for other tasks. This limited data scale directly constrains the achievable performance of style transfer under our unified framework.
>
> Second, the task itself is intrinsically more demanding. Style transfer requires the model to simultaneously preserve the structural and visual semantics of the original SVG while adapting it to the target style. In other words, the model must maintain content fidelity (layout, object identities, relationships) and, at the same time, perform style adaptation (colors, strokes, textures, overall artistic style). This dual constraint of "content preservation + style adaptation"[1] places stricter requirements on both SVG understanding and SVG generation capabilities than general generation tasks, where minor deviations in structure or content are often acceptable.
>
>
>
> [1]Chen, S., Dong, X., Xu, H., Wu, X., Tang, F., Zhang, H., ... & Zhuang, Y. (2025, October). Svgenius: Benchmarking llms in svg understanding, editing and generation. In *Proceedings of the 33rd ACM International Conference on Multimedia* (pp. 13289-13296).
>
> [2] Xing X, Zhou H, Wang C, et al. Svgdreamer: Text guided svg generation with diffusion model[C]//Proceedings of the IEEE/CVF Conference on Computer Vision and Pattern Recognition. 2024: 4546-4555.
>
> [3] Jain A, Xie A, Abbeel P. Vectorfusion: Text-to-svg by abstracting pixel-based diffusion models[C]//Proceedings of the IEEE/CVF Conference on Computer Vision and Pattern Recognition. 2023: 1911-1920.
>
> [4]Rodriguez, J. A., Puri, A., Agarwal, S., Laradji, I. H., Rodriguez, P., Rajeswar, S., ... & Pedersoli, M. (2025). Starvector: Generating scalable vector graphics code from images and text. In *Proceedings of the Computer Vision and Pattern Recognition Conference* (pp. 16175-16186).

---

> > ### Author Response · Authors · 2025-11-27
> >
> > Dear Reviewer rkyc,
> >
> > We would greatly appreciate it if you could let us know whether our responses have adequately addressed your concerns regarding (1) the potential biases introduced by synthetic illustration and animation data, (2) the lack of direct evaluation on SVG code quality, and (3) the challenges observed in the style transfer task. If the provided analyses and supplementary experiments have resolved these questions, your acknowledgement would be highly valuable to us. As the discussion deadline approaches, we remain happy to address any additional questions or suggestions you may have.
> >
> > Thanks!

---

> > > ### Comment · Reviewer_rkyc · 2025-11-28
> > > **Reply to rebuttal**
> > >
> > > Thank you to the author for the thoughtful and detailed response. I feel the author's reply has thoroughly addressed all my concerns. This is a very solid piece of work, and I am happy to raise the score to 8 points.

---

> > > > ### Author Response · Authors · 2025-12-01
> > > >
> > > > Thank you very much for your positive feedback and we sincerely appreciate it.

---

### Official Review · Reviewer_5t8t · 2025-10-30

**Soundness:** 2
**Presentation:** 3
**Contribution:** 3
**Rating:** 6
**Confidence:** 3

**Summary:**

This paper focuses on the tasks of Scalable Vector Graphics (SVG) understanding, generation, and editing. To address the limitations of prior works on SVG tasks, the paper proposes (1) a new large-scale and diverse SVG dataset SAgoge, (2) a new benchmark SArena with comprehensive tasks for SVG understanding, generation and editing and (3) a specialized MLLM: InternSVG for SVG tasks, built upon pre-trained VLMs with proposed special tokens and a two-stage training strategy. Experimental results demonstrate that InternSVG achieves superior performance on various SVG tasks.

**Strengths:**

* The unified modeling of SVG understanding, editing, and generation is an interesting and important problem, since there may be various practical applications, while the problem is still underexplored.
* The paper provides a comprehensive contribution to SVG tasks, with an integrated data-benchmark-model suite.
* There are thorough experiments on the newly introduced SArena benchmark, and the performance of the proposed InternSVG model outperforms both existing SVG methods and various MLLM models.

**Weaknesses:**

* The experiments in the main paper are conducted primarily on the proposed SArena benchmark. Given that SArena is developed in alignment with the training data SAgoge (probably the same data source & similar task definitions), the comparisons are not fair. More experiments on other existing benchmarks would be necessary to fully validate the effectiveness of the proposed data & methods.
* The illustration of the dataset / benchmark construction pipeline in the main paper is too brief, making it difficult to assess the methodological novelty or quality of the pipeline. And some details are not clear, e.g. how the annotations of the SVG editing task are obtained?
* Some ablations are missing. The paper proposes special tokens and an embedding initialization strategy for InternSVG training. But there is no ablation study about the impact of these designs on the task performances.

**Questions:**

For the SVG understanding task, have the authors considered or conducted experiments comparing the SVG-based understanding with the rendered image/video based understanding?

---

> ### Author Response · Authors · 2025-11-21
>
> Dear Reviewer 5t8t,
>
> Thank you for your constructive feedback and insightful comments. We are grateful for your recognition of the unified modeling approach and the comprehensive data-benchmark-model suite we have developed. We appreciate your thorough assessment and would like to address each of your concerns below.
>
> **For Weakness 1**, we would like to clarify that InternSVG 8B has already been evaluated on **SGP-Bench**[1] during the submission stage, and the results are reported in Appendix B.3. SGP-Bench is an SVG code understanding benchmark containing 4,340 multiple-choice questions targeting semantic and structural reasoning over SVG code. InternSVG 8B achieves the best score of **72.3**, indicating that training on SAgoge substantially improves the model’s ability to understand SVG structures and semantics on a dataset that is fully disjoint from SArena. To further validate the effectiveness  and generalization of SAgoge and our modeling framework, we additionally evaluated InternSVG 8B on **SVG-Stack**[2] and **UniSVG**[3] benchmarks. Specifically, **SVG-Stack** focuses on the *Image-to-SVG* reconstruction task, while **UniSVG** covers *Text-to-SVG*, *Image-to-SVG*, and *SVG understanding* tasks. On **SVG-Stack**, InternSVG 8B achieves the **best performance on Dino, LPIPS, and MSE**, while using **significantly fewer output tokens** compared to StarVector (**1.2k vs. 3.7k–5.3k**), and obtains comparable **SSIM** scores.
>
> | Method        | Dino ↑    | LPIPS ↓   | SSIM ↑    | MSE ↓     | Tokens |
> | ------------- | --------- | --------- | --------- | --------- | ------ |
> | GPT-4-V       | 0.852     | 0.317     | 0.711     | 0.195     | 443    |
> | StarVector 8B | 0.966     | 0.058     | **0.947** | 0.026     | 5.3k   |
> | InternSVG     | **0.968** | **0.052** | 0.916     | **0.023** | 1.2k   |
>
> On **UniSVG**, InternSVG 8B achieves **SOTA performance** with a **Final Score of 0.826**, substantially outperforming all baseline methods. In particular, it obtains a score of **0.939 on the ISVGEN metric** (the overall score for the Image-to-SVG task), surpassing the second-best model by **0.166**. It also achieves the **best TSVGEN score of 0.741** for the Text-to-SVG task. These results on three external benchmarks, SGP-Bench, SVG-Stack, and UniSVG, clearly demonstrate that the gains of InternSVG 8B are not confined to SArena or any single data source. Instead, they show that the SAgoge dataset and the InternSVG framework provide robust and broadly generalizable improvements across diverse SVG understanding and generation tasks.
>
> | Model Name        | Final Score | ISVGEN SSIM | ISVGEN LPIPS | ISVGEN CLIP Score | ISVGEN Score | TSVGEN SSIM | TSVGEN LPIPS | TSVGEN CLIP Score | TSVGEN Score | SVGUN Easy-Acc | SVGUN Hard-Acc | Bert      | SBert     |
> | ----------------- | ----------- | ----------- | ------------ | ----------------- | ------------ | ----------- | ------------ | ----------------- | ------------ | -------------- | -------------- | --------- | --------- |
> | LLaMA-3.2-3B      | 0.567       | 0.563       | 0.674        | 0.690             | 0.592        | 0.491       | 0.616        | 0.772             | 0.638        | 0.347          | 0.201          | -0.291    | 0.455     |
> | Qwen2.5-VL-7B     | 0.614       | 0.564       | 0.614        | 0.738             | 0.633        | 0.538       | 0.619        | 0.764             | 0.642        | 0.543          | 0.571          | 0.082     | 0.596     |
> | GPT 4V            | 0.650       | 0.557       | 0.582        | 0.740             | 0.638        | 0.620       | 0.531        | 0.816             | 0.712        | 0.893          | 0.448          | 0.211     | 0.520     |
> | Claude 3.7        | 0.722       | 0.622       | 0.473        | 0.855             | 0.743        | **0.622**   | 0.503        | 0.852             | 0.735        | 0.863          | 0.624          | 0.322     | 0.709     |
> | LLaMA-3.2-3B-SFT  | 0.732       | 0.722       | 0.378        | 0.843             | 0.775        | 0.617       | 0.511        | 0.812             | 0.709        | **0.990**      | 0.560          | 0.511     | 0.735     |
> | Qwen2.5-VL-7B SFT | 0.752       | 0.725       | 0.368        | 0.836             | 0.773        | 0.621       | **0.479**    | 0.853             | 0.740        | 0.983          | 0.604          | **0.574** | **0.827** |
> | InternSVG 8B      | **0.826**   | **0.890**   | **0.100**    | **0.970**         | **0.939**    | 0.620       | 0.490        | **0.860**         | **0.741**    | 0.943          | **0.625**      | 0.564     | 0.821     |

---

> ### Author Response · Authors · 2025-11-21
>
> **For Weakness 2**, we sincerely appreciate this valuable suggestion. Due to space limitations in the main paper, the data construction pipeline may not have been sufficiently clear. To address this, we have provided detailed descriptions of the dataset, benchmark design, synthetic data generation procedures, and the full annotation prompts in Appendix C to ensure clarity and reproducibility. Our data pipeline starts from collecting raw SVG code either via  web crawling or synthetic generation. We then normalize all coordinates to a 128×128 viewBox and simplify the SVG code using SVGO. For understanding tasks, we rasterize each SVG and feed the rendered image to MLLMs such as GPT-4o or InternVL3-78B to obtain fine-grained geometric and semantic descriptions (the exact prompts are listed  in Appendix C.4.1). For generation tasks, SVGs in the Icon and Illustration categories are annotated by providing rasterized images to an MLLM for captioning. For chemical structures, we directly use the IUPAC names or common chemical names retrieved from PubChem as textual instructions. For animation data, each SVG animation is converted into an MP4 video and annotate it using Gemini 2.0 Flash.
>
> For the editing task, the dataset is constructed as *(instruction, original SVG, edited SVG)* triples, the data generation process can be categorized into three types: simple editing, color editing, and style transfer.
>
> Simple editing includes operations such as cropping, flipping, opacity adjustment, adding stroke, rotation, scaling, and translation. We first generate transformation instructions using hand-crafted templates with randomized parameters, and then apply the corresponding operations by inserting elements such as `transform` or `clipPath` into the SVG code. To ensure correctness, we use Qwen2.5-VL-72B to compare the rendered images before and after editing, evaluating whether the transformation is reasonable, and then generate refined textual instructions based on prompt templates. During test set construction, human annotators also manually filter and validate the generated instructions.
>
> For color editing, we manually predefined 147 commonly used colors, and perform random color replacement at the SVG code level using regular expressions. After rasterization, we employ an MLLM to generate both simple color-editing instructions and semantic-level color-editing descriptions. Simple instructions follow predefined templates, while semantic instructions are generated by comparing the visual differences before and after the edit. Qwen2.5-VL-72B is used to annotate the training set, while GPT-4o is used for test-set instruction generation followed by human verification.
>
> For style transfer, we collect SVGs with identical category labels from datasets such as openmoji and colorsvg, rasterize each SVG, and compute similarity scores between images using CLIP. Image pairs with similarity scores above 0.8 are selected as input data. We then generate captions using Qwen2.5-VL-72B and prompt an MLLM to produce an editing instruction conditioned on the two images and their captions. The training set is annotated using Qwen2.5-VL-72B, whereas the test set is annotated with GPT-4o, followed by thorough manual inspection.
>
> All implementation details and exact prompt templates are documented in Appendix C to facilitate reproducibility.

---

> ### Author Response · Authors · 2025-11-21
>
> **For Weakness 3**, as described in Sec. 4.1, introducing SVG-specialized tokens effectively reduces token length, and the proposed subword-based embedding initialization  accelerates convergence and allows the model to reach strong performance with substantially less training data. To make their impact more explicit, we have added ablation studies evaluating both components. Concretely, we sampled only 50% of the Icon, Illustration, and Chemistry training data used in stage 2 (approximately 1.95M samples) and trained three model variants:
>
> **(1) Model Raw, without special tokens; (2) Model T, with special tokens but random embedding initialization; and (3) Model T+E, with both special tokens and subword-based embedding initialization.**
>
> The results on SArena-Icon and SArena-Illustration are presented in the table below. As shown, **Model T+E consistently achieves the highest success rate and the best overall performance, confirming the effectiveness of combining SVG-specialized tokens with subword-based initialization.** In addition, **models that incorporate special tokens (Model T and Model T+E) exhibit substantially higher success rates on illustration generation, which involves longer SVG sequences.** This is consistent with the token-compression effect illustrated in Figure 3(b): introducing SVG-specific special tokens shortens the sequence length, which in turn leads to more stable training and improved task success rates.
>
> > Model performance on SArena-Icon understanding and generation tasks.
>
> | Model | Un   | Text2SVG Success Rate | FID ↓  | FID-C ↓ | CLIP Score Text-Image ↑ | CLIP Score Image-Image ↑ | Img2SVG Success Rate | DINO_Score ↑ | SSIM ↑ | LPIPS ↓ | PSNR ↑ |
> | ----- | ---- | --------------------- | ------ | ------- | ----------------------- | ------------------------ | -------------------- | ------------ | ------ | ------- | ------ |
> | Raw   | 79.3 | 92.20%                | 15.788 | 5.112   | 22.767                  | 76.884                   | 84.30%               | 0.809        | 0.493  | 0.381   | 6.571  |
> | T     | 79.7 | 97.62%                | 11.922 | 4.578   | 23.275                  | 77.374                   | 89.59%               | 0.919        | 0.741  | 0.166   | 14.342 |
> | T + E | 80.8 | 98.42%                | 11.599 | 4.500   | 23.348                  | 77.541                   | 95.64%               | 0.937        | 0.795  | 0.139   | 15.837 |
>
> > Model performance on SArena-Illustration generation tasks.
>
> | Model | Text2SVG Success Rate | FID ↓  | FID-C ↓ | CLIP Score Text-Image ↑ | CLIP Score Image-Image ↑ | Img2SVG Success Rate | DINO_Score ↑ | SSIM ↑ | LPIPS ↓ | PSNR ↑ |
> | ----- | --------------------- | ------ | ------- | ----------------------- | ------------------------ | -------------------- | ------------ | ------ | ------- | ------ |
> | Raw   | 57.22%                | 68.301 | 26.108  | 18.870                  | 66.222                   | 50.08%               | 0.735        | 0.331  | 0.412   | 6.053  |
> | T     | 69.02%                | 53.080 | 22.358  | 19.174                  | 66.469                   | 57.37%               | 0.807        | 0.448  | 0.307   | 9.424  |
> | T + E | 78.81%                | 42.817 | 17.438  | 19.194                  | 67.379                   | 75.31%               | 0.862        | 0.578  | 0.257   | 11.810 |

---

> ### Author Response · Authors · 2025-11-21
>
> For **Question 1**, we would first like to clarify that *SVG-based understanding* and *rendered image/video–based understanding* correspond to fundamentally different problem settings.
>
> **SVG understanding can in principle be handled by a pure LLM operating on textual SVG code**, whereas understanding rendered images or videos requires **VLMs**. Thus, the two settings differ both in model architecture and in input modality, making a direct comparison non-trivial. More importantly, our work focuses on **code-level SVG understanding as the foundation for SVG editing and generation**. SVG is a special type of visual data encoded entirely in text, which allows the model to infer visual semantics directly from the underlying code. Compared with understanding rendered images or videos, performing understanding directly on SVG code relies only on text input, **enabling the model to imagine the underlying semantics and geometric structures without requiring any image or video as auxiliary input.** This code-level understanding provides a more fine-grained grasp of the structural properties of vector graphics. As shown in Table 5, our extensive ablation studies demonstrate that improving the model’s SVG code understanding capability in turn leads to better performance on SVG editing and generation tasks. Moreover, prior work such as **SGP-Bench** [1] has shown through comprehensive experiments that training models to understand SVG code not only enhances the visual imagination ability of LLMs but also further strengthens their reasoning capabilities.
>
> [1] Qiu Z, Liu W, Feng H, et al. Can Large Language Models Understand Symbolic Graphics Programs?[C]//The Thirteenth International Conference on Learning Representations.
>
> [2] Rodriguez J A, Puri A, Agarwal S, et al. Starvector: Generating scalable vector graphics code from images and text[C]//Proceedings of the Computer Vision and Pattern Recognition Conference. 2025: 16175-16186.
>
> [3] Li J, Yu J, Wei C, et al. Unisvg: A unified dataset for vector graphic understanding and generation with multimodal large language models[C]//Proceedings of the 33rd ACM International Conference on Multimedia. 2025: 13156-13163.

---

> > ### Author Response · Authors · 2025-11-27
> >
> > Dear Reviewer 5t8t,
> >
> > We would greatly appreciate it if you could let us know whether our responses have adequately addressed your concerns regarding (1) lack of evaluations on other existing benchmarks, (2) the clarity of the dataset and benchmark construction pipeline, (3) the ablation analyses on SVG-specialized tokens and embedding initialization, and (4) the distinction between SVG-based understanding and rendered image/video–based understanding. If our clarifications and additional experimental results have resolved these issues, your positive feedback would be very valuable to us. As the discussion deadline approaches, we remain happy to address any additional questions or suggestions you may have.
> >
> > Thanks!

---

### Official Review · Reviewer_gzZQ · 2025-10-31

**Soundness:** 3
**Presentation:** 3
**Contribution:** 3
**Rating:** 6
**Confidence:** 3

**Summary:**

This paper presents the InternSVG family, a unified suite that integrates dataset, benchmark, and model for SVG tasks. It introduces SAgoge, a large-scale multimodal SVG dataset with 16 million samples; SArena, a unified benchmark covering understanding, editing, and generation; and InternSVG, a MLLM designed for unified SVG modeling. Extensive experiments show that InternSVG achieves state-of-the-art results across multiple SVG-related tasks.

**Strengths:**

1. The work establishes a coherent and large-scale suite for SVG research, combining SAgoge and SArena to support consistent training and evaluation across understanding, editing, and generation tasks.
2. The model employs SVG-specific tokenization and a subword-based initialization strategy, together with a two-stage training scheme that reduces sequence length, stabilizes optimization, and speeds up convergence.
3. Comprehensive evaluations demonstrate that InternSVG achieves the best performance on all SArena sub-tasks, surpassing strong proprietary models such as GPT-4o and Claude-4-Sonnet.
4. The paper is well organized and easy to follow.

**Weaknesses:**

1. Since SAgoge’s annotations are generated using InternVL and QwenVL, which are also included as baselines, it remains unclear whether this setup introduces a bias favoring models from the same family during evaluation.
2. InternSVG is trained on large, SVG-specific data, whereas other models are used without such domain fine-tuning, making it unclear whether the gains arise from architectural advances or from data advantages.

**Questions:**

See Weakness

---

> ### Author Response · Authors · 2025-11-21
>
> Dear Reviewer gzZQ,
>
> Thank you for your thoughtful review and recognition of our contributions. We would like to address the concerns you raised regarding potential evaluation bias and the source of performance gains.
>
> **For Weakness 1**, we fully understand the concern about potential bias, and would like to clarify that the training dataset SAgoge and SArena benchmark adopt fundamentally different annotation strategies:
>
> - For SAgoge dataset, we used InternVL3 and Qwen2.5-VL for annotating Icon and Illustration data, as relying on proprietary models for 3.4M samples would incur prohibitive costs.
> - For SArena benchmark, we carefully considered evaluation fairness and deliberately selected the proprietary model, GPT-4o, as the annotator due to its exceptional image captioning capabilities. All annotations were further verified through manual inspection to ensure quality. This design decouples the evaluation process from any specific open-source model family and ensures a fair comparison across InternVL, QwenVL, and other open-source models, effectively mitigating potential bias arising from model family alignment.
>
> Moreover, InternSVG outperforms both open-source models (LLaMA-4, InternVL3, Qwen2.5-VL) and proprietary models (GPT-4o, Claude-Sonnet-4) across most SArena metrics, further substantiating that our effectiveness does not stem from evaluation bias.
>
> **For Weakness 2**, we acknowledge that InternSVG is trained on large-scale SVG data, while other models have not undergone such domain-specific fine-tuning. However, we believe this difference is not a confounder, but instead highlights the core contribution of our work. Existing open-source datasets generally lack high-quality SVG training data, despite SVG has critical applications in web UI design and scientific/engineering graphics. Therefore, we constructed SAgoge and introduced SVG-specialized tokens with a unified modeling paradigm, enabling effective mastery of SVG tasks.
>
> In practice, **data and architecture are tightly coupled**. Our SVG-specialized tokens, subword-based embedding initialization, unified modeling scheme, and two-stage training strategy are all designed to better leverage the strengths of SAgoge, rather than relying on raw scale alone. To further validate the effectiveness of our proposed methods, we have already demonstrated in Tables 5 and 6 the efficacy of unified modeling and the two-stage training strategy.
>
> We further conducted additional experiments to verify the effectiveness of SVG-specialized tokens and subword-based embedding initialization. We sampled 50% of the Icon, Illustration, and Chemistry training data used in stage 2 (~1.95M samples) and evaluated on icon understanding, icon generation, and illustration generation tasks. The tables below show evaluation results: Model Raw (baseline without special tokens), Model T (special tokens with random initialization), and Model T+E (special tokens with subword-based initialization). Results show that Model T+E achieves substantially superior performance in both task success rates and evaluation metrics, validating the effectiveness of our designed special tokens and subword-based embedding initialization.
>
> | Model |  Un  | Text2SVG Success Rate | FID ↓  | FID-C ↓ | CLIP Score Text-Image ↑ | CLIP Score Image-Image ↑ | Img2SVG Success Rate | DINO_Score ↑ | SSIM ↑ | LPIPS ↓ | PSNR ↑ |
> | :---: | :--: | :-------------------: | :----: | :-----: | :---------------------: | :----------------------: | :------------------: | :----------: | :----: | :-----: | :----: |
> |  Raw  | 79.3 |        92.20%         | 15.788 |  5.112  |         22.767          |          76.884          |        84.30%        |    0.809     | 0.493  |  0.381  | 6.571  |
> |   T   | 79.7 |        97.62%         | 11.922 |  4.578  |         23.275          |          77.374          |        89.59%        |    0.919     | 0.741  |  0.166  | 14.342 |
> | T + E | 80.8 |        98.42%         | 11.599 |  4.500  |         23.348          |          77.541          |        95.64%        |    0.937     | 0.795  |  0.139  | 15.837 |
>
> | Model | Text2SVG Success Rate | FID ↓  | FID-C ↓ | CLIP Score Text-Image ↑ | CLIP Score Image-Image ↑ | Img2SVG Success Rate | DINO_Score ↑ | SSIM ↑ | LPIPS ↓ | PSNR ↑ |
> | :---: | :-------------------: | :----: | :-----: | :---------------------: | :----------------------: | :------------------: | :----------: | :----: | :-----: | :----: |
> |  Raw  |        57.22%         | 68.301 | 26.108  |         18.870          |          66.222          |        50.08%        |    0.735     | 0.331  |  0.412  | 6.053  |
> |   T   |        69.02%         | 53.080 | 22.358  |         19.174          |          66.469          |        57.37%        |    0.807     | 0.448  |  0.307  | 9.424  |
> | T + E |        78.81%         | 42.817 | 17.438  |         19.194          |          67.379          |        75.31%        |    0.862     | 0.578  |  0.257  | 11.810 |

---

> > ### Author Response · Authors · 2025-11-27
> >
> > Dear Reviewer gzZQ,
> >
> > We would appreciate it if you could let us know whether our previous responses have adequately addressed your concerns regarding (1) the potential evaluation bias introduced by SAgoge’s annotation sources and (2) the distinction between data advantages and architectural contributions in InternSVG’s performance. If our clarifications and additional experimental results have resolved these issues, your positive feedback would be very valuable to us. As the discussion deadline approaches, we remain happy to address any additional questions or suggestions you may have.
> >
> > Thanks!

---

> > ### Comment · Reviewer_gzZQ · 2025-11-27
> >
> > Thanks for the clear response. The rebuttal clearly resolves my concern. Considering the method's insight and the solid experimental results, I will raise my score and I recommend to accept this paper.

---

> > > ### Author Response · Authors · 2025-11-27
> > >
> > > Thank you very much for your positive feedback and we sincerely appreciate it.

---

### Official Review · Reviewer_42aJ · 2025-11-01

**Soundness:** 3
**Presentation:** 2
**Contribution:** 2
**Rating:** 4
**Confidence:** 4

**Summary:**

This paper presents InternSVG, an integrated data–benchmark–model suite designed for unified modeling of Scalable Vector Graphics (SVG) tasks — encompassing understanding, editing, and generation. The work consists of three main components:
1. SAgoge — a large-scale (16M samples) multimodal dataset for SVG tasks covering icons, illustrations, chemical diagrams, and animations.
2. SArena — a standardized benchmark for evaluating understanding, editing, and generation performance across SVG domains.
3. InternSVG — a unified multimodal large language model (MLLM) built on a ViT–MLP–LLM backbone, augmented with SVG-specific tokenization (including tag, attribute, and numeric tokens) and subword-based embedding initialization. It is trained in a two-stage curriculum from simple to complex SVGs.

**Strengths:**

1. The work takes a holistic and unified approach to SVG understanding, editing, and generation — previously treated as isolated problems.
2. The paper is clearly written and well-organized, with informative figures illustrating datasets, architecture, and tokenization.
3. The open dataset and benchmark will likely become an important reference for future research on vector graphics and multimodal LLMs.

**Weaknesses:**

1. While the paper demonstrates strong empirical results, it provides limited theoretical or interpretive analysis of why unified modeling helps cross-task transfer. For example, a visualization or probing analysis of SVG token embeddings could strengthen understanding of learned representations.
2. The annotation pipeline leverages GPT-4o, Gemini, and other proprietary models for labeling, which may raise concerns about the bias. It would be helpful to involve human evaluation on a small subset of the synthetic data.
3. While quantitative metrics are comprehensive, qualitative results (e.g., in Figure 4) could be expanded to better illustrate failure cases or tradeoffs between semantic accuracy and geometric precision.
4. It would be helpful if the cross-domain transfer (e.g., training on icons → testing on scientific diagrams) could be further analyzed to highlight the generalization strength.
5. The authors mention replacing unrenderable SVGs with black images for evaluation. It would be helpful if the authors could quantify how often InternSVG produces invalid SVGs compared to other baselines. This would clarify the model’s robustness and syntactic correctness.

**Questions:**

1. Could you provide more analysis or visualization to help understand how unified modeling leads to cross-task improvements? For example, have you examined whether representations learned from SVG understanding tasks transfer effectively to generation tasks (e.g., through probing, clustering, or layer-wise similarity analysis)? Such insights could strengthen the paper’s conceptual contribution beyond empirical results.
2. While quantitative metrics are comprehensive (FID, SSIM, CLIP, etc.), have you conducted any human evaluation on the perceptual or semantic quality of generated SVGs? If not, could you comment on whether the model-generated outputs align with human preference or creative intent?

---

> ### Author Response · Authors · 2025-11-21
>
> Dear Reviewer 42aj,
>
> Thank you for your careful reading and valuable suggestions. We appreciate your feedback and will address each of your points in the revised version.
>
> **For Weakness 1 and Question 1**, we sincerely appreciate your valuable suggestions. We fully agree that, beyond reporting the empirical gains, it is important to provide a more interpretative analysis of why unified modeling facilitates cross-task transfer from the perspective of representation learning.
>
> First, as shown in Table 5, our quantitative ablations already indicate that unified modeling exhibits significant improvements over task-specialized models across all task metrics, which clearly demonstrates that unified modeling brings substantial cross-task transfer benefits. To further explain this phenomenon from the representation learning perspective, we have added a t-SNE feature visualization experiment. Concretely, we consider the three models used in the ablation study: the understanding-specialized model, the generation-specialized model, and the unified model. We sample 100 understanding examples and 100 generation examples from our dataset. The understanding-specialized model is evaluated on understanding inputs, the generation-specialized model  on generation inputs, while the unified model is evaluated on both types of inputs separately. For each input, we extract the last-layer hidden representation and project it into 2D space using t-SNE.
>
> The visualization reveals a clear contrast between the specialized  and unified settings:
>
> (1) For the two specialized models, the understanding features (blue) and generation features (orange) occupy two well-separated regions, indicating that the internal representations of the two task types are largely independent and not aligned in a shared semantic space.
>
> (2)  For the unified model, the understanding (green) and generation (red) features form two neighboring clusters in the embedding space. This indicates that unified modeling encourages shared and aligned representations across the two task types, forming a unified and transferable semantic space rather than two isolated representation structures.
>
> This t-SNE analysis provides representation-level evidence for our core claim: unified modeling not only improves performance, but also promotes **deep knowledge sharing and transfer through representation alignment** between understanding and generation tasks.
>
> Moreover, recent unified frameworks such as Janus [1] in the natural-image multimodal setting have reported similar observations: a shared Transformer backbone jointly serving understanding and generation tasks can learn structurally consistent representations across these tasks, and in turn achieve performance that matches or surpasses task-specialized models. Our findings on SVG tasks are consistent with this line of evidence and further support the benefits of unified modeling for cross-task transfer.
>
> **For Weakness 2**, we conducted a user study during the rebuttal stage to assess the quality of our synthetic annotations. We sampled 100 model-generated instructions from our dataset (40 icons, 40 illustrations, and 20 animations) and designed two types of evaluation questions. The first was a binary **consistency check**, where participants were asked to judge whether the model-generated instruction was semantically consistent with the rendered image or video. The second was a **pairwise preference test**, where participants were shown a human-authored instruction and a model-generated instruction for the same image (with sources hidden), and asked to choose among "A is better", "B is better", "Both are good", or "Both are bad". We collected 42 valid responses from participants with academic or professional backgrounds relevant to our study, including 14.29% researchers, 57.14% PhD students, and 28.57% UI designers. The cohort primarily comes from computer-related fields and also includes professional UI designers. Each questionnaire randomly selected 12 judgment questions and 12 comparison questions from the pool. The results show that in the semantic consistency evaluation, 95.04% of samples were rated as "semantically consistent with the image/video" (97.62% for icons, 92.38% for illustrations, and 95.24% for animations). In the comparative assessment of annotation quality, participants preferred the model-generated instructions in 45.43% of the cases, while human-authored instructions were preferred in only 15.67% of cases, and 36.90% of samples were considered equally good. Overall, **more than 82% of the samples indicate that model-generated annotations are comparable to human annotations in quality**. These findings validate the reliability of our synthetic annotations and suggest that potential labeling bias does not negatively affect model training or generalization.

---

> ### Author Response · Authors · 2025-11-21
>
> **For Weakness 3**, we agree that qualitative analysis is important for understanding the trade-offs between semantic accuracy and geometric precision. Due to space limitations, Sec 5.3 in the main paper includes only Figure 4 to showcase the SVG generation capability of InternSVG. To further compare with baselines, we have substantially expanded the qualitative analysis in **Appendix B.5**, where Figures 5–15 present representative outputs of InternSVG across different tasks, together with side-by-side comparisons against other baselines.
>
> For the Text-to-SVG tasks on icons and illustrations, although traditional approaches such as VectorFusion [2] and SVGDreamer [3] can produce visually appealing SVGs, they often suffer from poor instruction following and frequently introduce many irrelevant elements. Existing general-purpose models tend to produce structurally unstable or semantically incomplete SVGs and struggle to achieve both visual quality and semantic alignment. LLM-based SVG generation methods such as LLM4SVG [4] and OmniSVG [5] also frequently produce outputs that deviate substantially from the given instructions, revealing limited generalization ability. In contrast, InternSVG achieves strong instruction fidelity and consistent fine-grained SVG generation, producing stable, high-quality results that closely match the prompts. For the Image-to-SVG task, InternSVG delivers higher visual fidelity and a higher task success rate than general models and LLM-based SVG generation methods such as StarVector [6], LLM4SVG, and OmniSVG. At the same time, compared with traditional methods like DiffVG [7] and LIVE [8], InternSVG generates fewer redundant elements, exhibits fewer structural artifacts, and achieves better detail preservation, as illustrated in rows 2 and 5 of Figure 6.
>
> **For Weakness 4**, we appreciate this valuable suggestion. To rigorously quantify the cross-domain generalization strength of our model, we conducted a zero-shot cross-domain transfer experiment during the rebuttal phase.  Specifically, we trained a variant of the InternSVG 8B model only on the Icon Image-to-SVG task data in stage 2, and then directly evaluated it on the SArena-Chemistry Image-to-SVG task, without using any Chemistry data during training. The results show that this icon-only InternSVG 8B variant still achieves substantial improvements over the original InternVL3-8B baseline across all metrics. Concretely, the DINO Score increased from **0.865 to 0.930 (7.5%↑)**, SSIM increased from **0.783 to 0.824 (5.2%↑)**, LPIPS decreased from **0.203 to 0.156 (23.2%↓)**, and PSNR increased from **13.84 to 16.296 (17.8%↑)**. The comprehensive improvements across all visual quality metrics strongly validate that our method and data possess exceptional generalization capabilities in cross-domain transfer scenarios.
>
> | Model        | Success Rate | DINO_Score ↑ | SSIM ↑ | LPIPS ↓ | PSNR ↑ |
> | ------------ | ------------ | ------------ | ------ | ------- | ------ |
> | InternVL3-8B | 96.27%       | 0.865        | 0.783  | 0.203   | 13.840 |
> | Ours w/ icon | 98.61%       | 0.930        | 0.824  | 0.156   | 16.296 |
>
> **For Weakness 5**, we sincerely appreciate your suggestion. We will update the paper to report full success rates in Tables 11–14 for all SArena generation tasks. For example, although StarVector achieves competitive quantitative results on the SArena-Icon Image-to-SVG task, its actual success rate is only 72.51%. Therefore, reporting metrics computed only on successfully rendered SVGs can be misleading and does not fully reflect a model’s true robustness or syntactic reliability. In contrast, InternSVG-8B consistently achieves over 90% success rate across all tasks, demonstrating substantially stronger syntactic correctness and reliability.

---

> ### Author Response · Authors · 2025-11-21
>
> **For Question 2**, in response to the reviewer's concerns regarding whether the perceptual and semantic quality of the SVG generated by our model aligns with human preferences, we conducted a systematic human evaluation. For each participant, we randomly sampled 5 icons, 5 illustrations, 3 chemical structures, and 3 animations from the Text-to-SVG/Text-to-SANI and Image-to-SVG/Video-to-SANI evaluation results for scoring. In each task group, we presented the participants with the input text description or image, along with the SVG outputs generated by six models (GPT-4o, Gemini-2.5-Flash, Claude-Sonnet-4, Qwen2.5-VL-72B, OmniSVG, InternSVG-8B) in randomized order. Participants rated each output on a scale of 1–5 across multiple dimensions, including:
>
> - Semantic Match / Visual Similarity: Whether the output accurately fulfills the input description or faithfully reconstructs the image content;
> - Visual Quality: Whether the graphic structure is clear, concise, and stable, and whether there are obvious defects;
> - Human Design Alignment: Whether the SVG conforms to common human design intuitions (simplicity, structural clarity, compositional balance, color harmony, etc.);
> - Usability: Whether the generated SVG can be directly used in practical design and editing workflows.
>
> The user study results demonstrate that InternSVG 8B significantly outperforms other baseline models in both adherence to user input instructions and image reconstruction fidelity. The generated SVGs exhibit higher quality, clearer and more stable graphic structures, fewer defects, and better alignment with human design preferences and usability. Overall, our quantitative evaluation metrics are consistent with the user study results. For example, in Image-to-SVG for icons, the user evaluation results show InternSVG-8B (4.64), Claude-Sonnet-4 (4.00), OmniSVG-3B (3.90), Gemini-2.5-Flash (3.73), GPT-4o (3.72), and Qwen2.5-VL-72B (2.42), presenting a trend that is largely consistent with our quantitative metric results.
>
> | Model            | Visual Similarity | Visual Quality | Human Design Alignment | Usability | Average Score |
> | ---------------- | ----------------- | -------------- | ---------------------- | --------- | ------------- |
> | InternSVG 8B     | 4.85              | 4.55           | 4.66                   | 4.48      | 4.64          |
> | Claude Sonnet 4  | 4.23              | 3.89           | 4.06                   | 3.82      | 4.00          |
> | OmniSVG 3B       | 4.16              | 3.77           | 3.93                   | 3.74      | 3.90          |
> | Gemini 2.5 Flash | 3.99              | 3.83           | 3.59                   | 3.51      | 3.73          |
> | GPT-4o           | 4.06              | 3.76           | 3.57                   | 3.49      | 3.72          |
> | Qwen2.5-VL-72B   | 2.65              | 2.35           | 2.48                   | 2.20      | 2.42          |
>
> [1] Wu C, Chen X, Wu Z, et al. Janus: Decoupling visual encoding for unified multimodal understanding and generation[C]//Proceedings of the Computer Vision and Pattern Recognition Conference. 2025: 12966-12977.
>
> [2] Jain A, Xie A, Abbeel P. Vectorfusion: Text-to-svg by abstracting pixel-based diffusion models[C]//Proceedings of the IEEE/CVF Conference on Computer Vision and Pattern Recognition. 2023: 1911-1920.
>
> [3] Xing X, Zhou H, Wang C, et al. Svgdreamer: Text guided svg generation with diffusion model[C]//Proceedings of the IEEE/CVF Conference on Computer Vision and Pattern Recognition. 2024: 4546-4555.
>
> [4] Xing X, Hu J, Liang G, et al. Empowering llms to understand and generate complex vector graphics[C]//Proceedings of the Computer Vision and Pattern Recognition Conference. 2025: 19487-19497.
>
> [5] Yang Y, Cheng W, Chen S, et al. Omnisvg: A unified scalable vector graphics generation model[C]//The Thirty-ninth Annual Conference on Neural Information Processing Systems. 2025.
>
> [6] Rodriguez J A, Puri A, Agarwal S, et al. Starvector: Generating scalable vector graphics code from images and text[C]//Proceedings of the Computer Vision and Pattern Recognition Conference. 2025: 16175-16186.
>
> [7] Li T M, Lukáč M, Gharbi M, et al. Differentiable vector graphics rasterization for editing and learning[J]. ACM Transactions on Graphics (TOG), 2020, 39(6): 1-15.
>
> [8] Ma X, Zhou Y, Xu X, et al. Towards layer-wise image vectorization[C]//Proceedings of the IEEE/CVF Conference on Computer Vision and Pattern Recognition. 2022: 16314-16323.

---

> > ### Comment · Reviewer_42aJ · 2025-11-24
> > **Reply to rebuttal**
> >
> > Thanks authors for the reply. It addressed my concerns and I would love to increase my rating.

---

> > > ### Author Response · Authors · 2025-11-27
> > >
> > > Thank you very much for your positive feedback and we sincerely appreciate it.

---

### Author Response · Authors · 2025-11-25
**Changes in the revised version of the paper**

Dear reviewers,

We sincerely appreciate your insightful and constructive feedback on our paper. Across the rebuttal, we made additions responding to all major reviewer concerns. We have now uploaded the revised manuscript, and all newly added content has been highlighted in **magenta** for your convenience.

In **Sec. 3.1.2**, we added more detailed descriptions of the data annotation pipeline, covering the construction processes for understanding, generation, and editing data.

In **Sec. 5.2**, **Appendix B.4** and **B.5**, we included the performance of InternSVG on SVG-Stack and UniSVG.

In **Sec. 5.4**, we added ablation studies evaluating the effectiveness of SVG-specific special tokens and subword-based embedding initialization.

In **Appendix B.6**, we provided a comparative analysis of the code quality of generated SVGs.

In **Appendix B.7**, we added visualization of cross-task representation alignment.

In **Appendix B.9**, we included a cross-domain generalization analysis.

In **Appendix B.11**, we provided detailed descriptions of the user study, including both the evaluation of synthetic data quality and the human evaluation of SVG generation quality.

---

### Author Response · Authors · 2025-12-01
**Summary to Area Chairs**

Dear Area Chairs,

We want to further clarify our work and to highlight the significant efforts made during the rebuttal period.

## Summary of responses from reviewers

We fully understand and respect the ICLR Program Chairs' decision to roll back scores to their initial state to maintain the integrity of the conference. However, we respectfully wish to draw your attention to the fact that **before the system reset, our extensive rebuttal efforts had successfully addressed the reviewers’ concerns and led to a significant score increase from an initial 6, 6, 6, 4 to 8, 8, 6, 6.** This improvement in the scores reflects the reviewers’ genuine recognition of the enhanced quality of our paper after revision.

- **Reviewer 42aJ** (commented on 24 Nov 2025, 18:03 UTC, before the leak incident): Score improved from **4** to **6**. The reviewer noted that "Thanks authors for the reply. It addressed my concerns and I would love to increase my rating."
- **Reviewer gzZQ** (commented on 27 Nov 2025, 09:02 UTC, before the leak incident): Score improved from **6** to **8**. The reviewer noted that "Thanks for the clear response. The rebuttal clearly resolves my concern. Considering the method's insight and the solid experimental results, I will raise my score and I recommend to accept this paper."
- **Reviewer rkyc** (commented on 28 Nov 2025, 09:35 UTC, after the leak incident): Score improved from **6** to **8**. The reviewer noted that "Thank you to the author for the thoughtful and detailed response. I feel the author's reply has thoroughly addressed all my concerns. This is a very solid piece of work, and I am happy to raise the score to 8 points."

## Summary of the Core Contributions of the Paper

InternSVG is designed to address the challenges of fragmented datasets, task-specific modeling, and limited generalization in existing SVG research. To this end, we focus on unified SVG modeling and introduce the **InternSVG family**, which consists of the **SAgoge dataset**, the **SArena benchmark**, and the **InternSVG 8B model**. Our core contributions are as follows:

- **SAgoge Dataset:** We construct the largest and most comprehensive multimodal SVG dataset to date, containing over **16 million** training samples. The dataset spans icons, illustrations, chemical structures, and animations, and supports unified training for SVG understanding, editing, and generation tasks.
- **SArena Benchmark:** We develop SArena, a companion benchmark designed to provide standardized and systematic evaluation of SVG understanding, editing, and generation. It consists of four sub-benchmarks and fills the gaps in existing fragmented evaluation protocols.
- **InternSVG 8B:** We present a unified MLLM for SVG understanding, editing, and generation. By introducing SVG-specific special tokens, subword-based embedding initialization, and a two-stage training strategy, InternSVG 8B is capable of modeling a wide range of SVGs, from short static graphics to long-sequence illustrations and complex animations.

- **Extensive empirical validation:** By conducting extensive experiments, we demonstrate the advantages of unified modeling. The results on SArena and prior benchmarks show that InternSVG 8B consistently outperforms traditional approaches as well as general-purpose open-source and proprietary models.

## Summary of our work during Rebuttal

During the rebuttal phase, we further incorporated the following enhancements:

1. Detailed descriptions of the data annotation pipeline, including the construction processes for understanding, generation, and editing data.
2. Evaluations of InternSVG 8B on SVG-Stack and UniSVG
3. Ablation studies assessing the effectiveness of SVG-specific special tokens and subword-based embedding initialization.
4. A comparative analysis of the structural quality of generated SVG code.
5. Visualization of cross-task representation alignment.
6. A cross-domain generalization analysis.
7. We conducted a user study, including the evaluation of synthetic data quality and the human assessment of SVG generation quality.

Thank you for your time and dedication to the review process.

Yours sincerely,

Authors

---

### Meta-Review · Program_Chairs · 2026-01-06

**Summary:**

The concerns of the reviewers can be summarized as below:
1. Limited interpretive analysis / qualitative results. (42aJ, 5t8t)
2. Model robustness in terms of unrenderable SVGs (42aJ)
3. Evaluation bias introduced by InternVL and QwenVL due to leveraging them for data annotation. Evaluation fairness (gzZQ, 5t8t, rkyc)
4. Unclear of the performance improvement (gzZQ)
5. insufficient evaluation of the generated code quality (rkyc)

**Reviewer Concerns:**

All the reviewers (except 5t8t) replied that their concerns have been addressed by the rebuttal and agreed to improve their ratings.
For the concerns raised by 5t8t, the rebuttal provided a comprehensive response. I agree with the authors that the concern shall be addressed.

**Reviewer Scores:**

Except the 5t8t, all the reviewers are agreed to improve their ratings.

---

### Decision · Program_Chairs · 2026-01-26

Accept (Poster)